# Follow the Perturbed Leader: Optimism and Fast Parallel Algorithms for Smooth Minimax Games

**Arun Sai Suggala**
Carnegie Mellon University
asuggala@cs.cmu.edu

**Praneeth Netrapalli**
Microsoft Research, India
praneeth@microsoft.com

## Abstract

We consider the problem of online learning and its application to solving minimax games. For the online learning problem, Follow the Perturbed Leader (FTPL) is a widely studied algorithm which enjoys the optimal $O\left(T^{1/2}\right)$ *worst case* regret guarantee for both convex and nonconvex losses. In this work, we show that when the sequence of loss functions is *predictable*, a simple modification of FTPL which incorporates optimism can achieve better regret guarantees, while retaining the optimal worst case regret guarantee for unpredictable sequences. A key challenge in obtaining these tighter regret bounds is the stochasticity and optimism in the algorithm, which requires different analysis techniques than those commonly used in the analysis of FTPL. The key ingredient we utilize in our analysis is the dual view of perturbation as regularization. While our algorithm has several applications, we consider the specific application of minimax games. For solving smooth convex-concave games, our algorithm only requires access to a linear optimization oracle. For Lipschitz and smooth nonconvex-nonconcave games, our algorithm requires access to an optimization oracle which computes the perturbed best response. In both these settings, our algorithm solves the game up to an accuracy of $O\left(T^{-1/2}\right)$ using $T$ calls to the optimization oracle. An important feature of our algorithm is that it is highly parallelizable and requires only $O(T^{1/2})$ iterations, with each iteration making $O\left(T^{1/2}\right)$ parallel calls to the optimization oracle.

## 1 Introduction

In this work, we consider the problem of online learning, where in each iteration, the learner chooses an action and observes a loss function. The goal of the learner is to choose a sequence of actions which minimizes the cumulative loss suffered over the course of learning. The paradigm of online learning has many theoretical and practical applications and has been widely studied in a number of fields, including game theory and machine learning. One of the popular applications of online learning is in solving minimax games arising in various contexts such as boosting [1], robust optimization [2], Generative Adversarial Networks [3].

In recent years, a number of efficient algorithms have been developed for regret minimization. These algorithms fall into two broad categories, namely, Follow the Regularized Leader (FTRL) [4] and FTPL [5] style algorithms. When the sequence of loss functions encountered by the learner are convex, both these algorithms are known to achieve the optimal $O\left(T^{1/2}\right)$ worst case regret [6, 7]. While these algorithms have similar regret guarantees, they differ in computational aspects. Each iteration of FTRL involves optimization of a non-linear convex function over the action space (also called the projection step). In contrast, each step of FTPL involves solving a linear optimization problem, which can be implemented efficiently for many problems of interest [8, 9, 10]. For example, if the action space is an $\ell_p$ ball for some $p \notin \{1, 2, \infty\}$, then projecting onto this set is much more computationally expensive than performing linear optimization over this set. As another example, consider the

scenario where the action space is the set of all positive semidefinite matrices. Then projecting onto this set requires performing expensive singular value decompositions. Whereas, linear optimization only requires computation of the leading eigenvector. This crucial difference between FTRL and FTPL makes the latter algorithm more attractive in practice. Even in the more general nonconvex setting, where the loss functions encountered by the learner can potentially be nonconvex, FTPL algorithms are attractive. In this setting, FTPL requires access to an offline optimization oracle which computes the perturbed best response, and achieves $O\left(T^{1/2}\right)$ worst case regret [11]. Furthermore, these optimization oracles can be efficiently implemented for many problems by leveraging the rich body of work on global optimization [12].

Despite its importance and popularity, FTPL has been mostly studied for the worst case setting, where the loss functions are assumed to be adversarially chosen. In a number of applications of online learning, the loss functions are actually benign and predictable [13]. In such scenarios, FTPL can not utilize the predictability of losses to achieve tighter regret bounds. While [11, 13] study variants of FTPL which can make use of predictability, these works either consider restricted settings or provide sub-optimal regret guarantees (see Section 2 for more details). This is unlike FTRL, where optimistic variants that can utilize the predictability of loss functions have been well understood [13, 14] and have been shown to provide faster convergence rates in applications such as minimax games. In this work, we aim to bridge this gap and study a variant of FTPL called Optimistic FTPL (OFTPL), which can achieve better regret bounds, while retaining the optimal worst case regret guarantee for unpredictable sequences. The main challenge in obtaining these tighter regret bounds is handling the stochasticity and optimism in the algorithm, which requires different analysis techniques to those commonly used in the analysis of FTPL. In this work, we rely on the dual view of perturbation as regularization to derive regret bounds of OFTPL.

To demonstrate the usefulness of OFTPL, we consider the problem of solving minimax games. A widely used approach for solving such games relies on online learning algorithms [6]. In this approach, both the minimization and the maximization players play a repeated game against each other and rely on online learning algorithms to choose their actions in each round of the game. In our algorithm for solving games, we let both the players use OFTPL to choose their actions. For solving smooth convex-concave games, our algorithm only requires access to a linear optimization oracle. For Lipschitz and smooth nonconvex-nonconcave games, our algorithm requires access to an optimization oracle which computes the perturbed best response. In both these settings, our algorithm solves the game up to an accuracy of $O\left(T^{-1/2}\right)$ using $T$ calls to the optimization oracle. While there are prior algorithms that achieve these convergence rates [11, 15], an important feature of our algorithm is that it is highly parallelizable and requires only $O(T^{1/2})$ iterations, with each iteration making $O\left(T^{1/2}\right)$ parallel calls to the optimization oracle. We note that such parallelizable algorithms are especially useful in large-scale machine learning applications such as training of GANs, adversarial training, which often involve huge datasets such as ImageNet [16].

## 2 Preliminaries and Background Material

**Online Learning.**   The online learning framework can be seen as a repeated game between a learner and an adversary. In this framework, in each round $t$, the learner makes a prediction $\mathbf{x}_t \in \mathcal{X} \subseteq \mathbb{R}^d$ for some compact set $\mathcal{X}$, and the adversary simultaneously chooses a loss function $f_t : \mathcal{X} \to \mathbb{R}$ and observe each others actions. The goal of the learner is to choose a sequence of actions $\{\mathbf{x}_t\}_{t=1}^T$ so that the following notion of regret is minimized: $\sum_{t=1}^T f_t(\mathbf{x}_t) - \inf_{\mathbf{x} \in \mathcal{X}} \sum_{t=1}^T f_t(\mathbf{x})$.

When the domain $\mathcal{X}$ and loss functions $f_t$ are convex, a number of efficient algorithms for regret minimization have been studied. Some of these include deterministic algorithms such as Online Mirror Descent, Follow the Regularized Leader (FTRL) [4, 7], and stochastic algorithms such as Follow the Perturbed Leader (FTPL) [5]. In FTRL, one predicts $\mathbf{x}_t$ as $\operatorname{argmin}_{\mathbf{x} \in \mathcal{X}} \sum_{i=1}^{t-1} \langle \nabla_i, \mathbf{x} \rangle + R(\mathbf{x})$, for some strongly convex regularizer $R$, where $\nabla_i = \nabla f_i(\mathbf{x}_i)$. FTRL is known to achieve the optimal $O(T^{1/2})$ worst case regret in the convex setting [4]. In FTPL, one predicts $\mathbf{x}_t$ as $m^{-1} \sum_{j=1}^m \mathbf{x}_{t,j}$, where $\mathbf{x}_{t,j}$ is a minimizer of the following linear optimization problem: $\operatorname{argmin}_{\mathbf{x} \in \mathcal{X}} \left\langle \sum_{i=1}^{t-1} \nabla_i - \sigma_{t,j}, \mathbf{x} \right\rangle$. Here, $\{\sigma_{t,j}\}_{j=1}^m$ are independent random perturbations drawn from some appropriate probability distribution such as exponential distribution or uniform distribution in a hyper-cube. Various choices of perturbation distribution gives rise to various FTPL algorithms. When the loss functions are

linear, Kalai and Vempala [5] show that FTPL achieves $O\left(T^{1/2}\right)$ expected regret, irrespective of the choice of $m$. When the loss functions are convex, Hazan [7] showed that the deterministic version of FTPL (*i.e.,* as $m \to \infty$) achieves $O\left(T^{1/2}\right)$ regret. While projection free methods for online convex learning have been studied since the early work of [17], surprisingly, regret bounds of FTPL for finite $m$ have only been recently studied [10]. Hazan and Minasyan [10] show that for Lipschitz and convex functions, FTPL achieves $O\left(T^{1/2} + m^{-1/2}T\right)$ expected regret, and for smooth convex functions, the algorithm achieves $O\left(T^{1/2} + m^{-1}T\right)$ expected regret.

When either the domain $\mathcal{X}$ or the loss functions $f_t$ are non-convex, no deterministic algorithm can achieve $o(T)$ regret [6, 11]. In such cases, one has to rely on randomized algorithms to achieve sub-linear regret. In randomized algorithms, in each round $t$, the learner samples the prediction $\mathbf{x}_t$ from a distribution $P_t \in \mathcal{P}$, where $\mathcal{P}$ is the set of all probability distributions supported on $\mathcal{X}$. The goal of the learner is to choose a sequence of distributions $\{P_t\}_{t=1}^T$ to minimize the expected regret $\sum_{t=1}^T \mathbb{E}_{\mathbf{x} \sim P_t}\left[f_t(\mathbf{x})\right] - \inf_{\mathbf{x} \in \mathcal{X}} \sum_{t=1}^T f_t(\mathbf{x})$. A popular technique to minimize the expected regret is to consider a linearized problem in the space of probability distributions with losses $\tilde{f}_t(P) = \mathbb{E}_{\mathbf{x} \sim P}\left[f_t(\mathbf{x})\right]$ and perform FTRL in this space. In such a technique, $P_t$ is computed as: $\operatorname{argmin}_{P \in \mathcal{P}} \sum_{i=1}^{t-1} \tilde{f}_i(P) + R(P)$, for some strongly convex regularizer $R(P)$. When $R(P)$ is the negative entropy of $P$, the algorithm is called entropic mirror descent or continuous exponential weights. This algorithm achieves $O\left(T^{1/2}\right)$ expected regret for bounded loss functions $f_t$. Another technique to minimize expected regret is to rely on FTPL [11, 18]. Here, the learner generates the random prediction $\mathbf{x}_t$ by first sampling a random perturbation $\sigma$ and then computing the perturbed best response, which is defined as $\operatorname{argmin}_{\mathbf{x} \in \mathcal{X}} \sum_{i=1}^{t-1} f_i(\mathbf{x}) - \langle \sigma, \mathbf{x} \rangle$. In a recent work, Agarwal et al. [18] show that this algorithm achieves $O\left(T^{2/3}\right)$ expected regret, whenever the sequence of loss functions are Lipschitz. This was later improved to $O\left(T^{1/2}\right)$ by Suggala and Netrapalli [11]. We now briefly discuss the computational aspects of FTRL and FTPL. Each iteration of FTRL (with entropic regularizer) requires sampling from a non-logconcave distribution. In contrast, FTPL requires solving a nonconvex optimization problem to compute the perturbed best response. Of these, computing the perturbed best response seems significantly easier since standard algorithms such as gradient descent seem to be able to find approximate global optima reasonably fast, even for complicated tasks such as training deep neural networks.

**Online Learning with Optimism.** When the sequence of loss functions are convex and predictable, Rakhlin and Sridharan [13, 14] study optimistic variants of FTRL which can exploit the predictability to obtain better regret bounds. Let $g_t$ be our guess of $\nabla_t$ at the beginning of round $t$. Given $g_t$, we predict $\mathbf{x}_t$ in Optimistic FTRL (OFTRL) as $\operatorname{argmin}_{\mathbf{x} \in \mathcal{X}} \left\langle \sum_{i=1}^{t-1} \nabla_i + g_t, \mathbf{x} \right\rangle + R(\mathbf{x})$. Note that when $g_t = 0$, OFTRL is equivalent to FTRL. [13, 14] show that the regret bounds of OFTRL only depend on $(g_t - \nabla_t)$. Moreover, these works show that OFTRL provides faster convergence rates for solving smooth convex-concave games. In contrast to FTRL, the optimistic variants of FTPL have been less well understood. [13] studies OFTPL for linear loss functions. But they consider restrictive settings and their algorithms require the knowledge of sizes of deviations $(g_t - \nabla_t)$. [11] studies OFTPL for the more general nonconvex setting. The algorithm predicts $\mathbf{x}_t$ as $\operatorname{argmin}_{\mathbf{x} \in \mathcal{X}} \sum_{i=1}^{t-1} f_i(\mathbf{x}) + g_t(\mathbf{x}) - \langle \sigma, \mathbf{x} \rangle$, where $g_t$ is our guess of $f_t$. However, the regret bounds of [11] are sub-optimal and weaker than the bounds we obtain in our work (see Theorem 4.2). Moreover, [11] does not provide any consequences of their results to minimax games. We note that their sub-optimal regret bounds translate to sub-optimal rates of convergence for solving smooth minimax games.

**Minimax Games.** Consider the following problem, which we refer to as minimax game: $\min_{\mathbf{x} \in \mathcal{X}} \max_{\mathbf{y} \in \mathcal{Y}} f(\mathbf{x}, \mathbf{y})$. In these games, we are often interested in finding a Nash Equilibrium (NE). A pair $(P, Q)$, where $P$ is a probability distribution over $\mathcal{X}$ and $Q$ is a probability distribution over $\mathcal{Y}$, is called a NE if: $\sup_{\mathbf{y} \in \mathcal{Y}} \mathbb{E}_{\mathbf{x} \sim P}\left[f(\mathbf{x}, \mathbf{y})\right] \leqslant \mathbb{E}_{\mathbf{x} \sim P, \mathbf{y} \sim Q}\left[f(\mathbf{x}, \mathbf{y})\right] \leqslant \inf_{\mathbf{x} \in \mathcal{X}} \mathbb{E}_{\mathbf{y} \sim Q}\left[f(\mathbf{x}, \mathbf{y})\right].$ A standard technique for finding a NE of the game is to rely on no-regret algorithms [6, 7]. Here, both $\mathbf{x}$ and $\mathbf{y}$ players play a repeated game against each other and use online learning algorithms to choose their actions. The average of the iterates generated via this repeated game can be shown to converge to a NE.

**Projection Free Learning.** Projection free learning algorithms are attractive as they only involve solving linear optimization problems. Two broad classes of projection free techniques have been considered for online convex learning and minimax games, namely, Frank-Wolfe (FW) methods

and FTPL based methods. Garber and Hazan [8] consider the problem of online learning when the action space $\mathcal{X}$ is a *polytope*. They provide a FW method which achieves $O\left(T^{1/2}\right)$ regret using $T$ calls to the linear optimization oracle. Hazan and Kale [17] provide a FW technique which achieves $O\left(T^{3/4}\right)$ regret for general online convex learning with Lipschitz losses and uses $T$ calls to the linear optimization oracle. In a recent work, Hazan and Minasyan [10] show that FTPL achieves $O\left(T^{2/3}\right)$ regret for online convex learning with smooth losses, using $T$ calls to the linear optimization oracle. This translates to $O\left(T^{-1/3}\right)$ rate of convergence for solving smooth convex-concave games. Note that, in contrast, our algorithm achieves $O\left(T^{-1/2}\right)$ convergence rate in the same setting. Gidel et al. [9] study FW methods for solving convex-concave games. When the constraint sets $\mathcal{X}, \mathcal{Y}$ are *strongly convex*, the authors show geometric convergence of their algorithms. In a recent work, He and Harchaoui [15] propose a FW technique for solving smooth convex-concave games which converges at a rate of $O\left(T^{-1/2}\right)$ using $T$ calls to the linear optimization oracle. We note that our simple OFTPL based algorithm achieves these rates, with the added advantage of parallelizability. That being said, He and Harchaoui [15] achieve dimension free convergence rates in the Euclidean setting, where the smoothness is measured w.r.t $\|\cdot\|_2$ norm. In contrast, the rates of convergence of our algorithm depend on the dimension.

**Notation.** $\|\cdot\|$ is a norm on some vector space, which is typically $\mathbb{R}^d$ in our work. $\|\cdot\|_*$ is the dual norm of $\|\cdot\|$, which is defined as $\|\mathbf{x}\|_* = \sup\{\langle \mathbf{u}, \mathbf{x}\rangle : \mathbf{u} \in \mathbb{R}^d, \|\mathbf{u}\| \leqslant 1\}$. We use $\Psi_1, \Psi_2$ to denote norm compatibility constants of $\|\cdot\|$, which are defined as $\Psi_1 = \sup_{\mathbf{x}\neq 0} \|\mathbf{x}\|/\|\mathbf{x}\|_2$, $\Psi_2 = \sup_{\mathbf{x}\neq 0} \|\mathbf{x}\|_2/\|\mathbf{x}\|$.

We use the notation $f_{1:t}$ to denote $\sum_{i=1}^{t} f_i$ and $\nabla_i$ to denote $\nabla f_i(\mathbf{x}_i)$. In some cases, when clear from context, we overload the notation $f_{1:t}$ and use it to denote the set $\{f_1, f_2 \dots f_t\}$. For any convex function $f$, $\partial f(\mathbf{x})$ is the set of all subgradients of $f$ at $\mathbf{x}$. For any function $f : \mathcal{X} \times \mathcal{Y} \to \mathbb{R}$, $f(\cdot, \mathbf{y}), f(\mathbf{x}, \cdot)$ denote the functions $\mathbf{x} \to f(\mathbf{x}, \mathbf{y}), \mathbf{y} \to f(\mathbf{x}, \mathbf{y})$. For any function $f : \mathcal{X} \to \mathbb{R}$ and any probability distribution $P$, we let $f(P)$ denote $\mathbb{E}_{\mathbf{x}\sim P}\left[f(\mathbf{x})\right]$. Similarly, for any function $f : \mathcal{X} \times \mathcal{Y} \to \mathbb{R}$ and any two distributions $P, Q$, we let $f(P, Q)$ denote $\mathbb{E}_{\mathbf{x}\sim P, \mathbf{y}\sim Q}\left[f(\mathbf{x}, \mathbf{y})\right]$. For any set of distributions $\{P_j\}_{j=1}^{m}$, $\frac{1}{m}\sum_{j=1}^{m} P_j$ is the mixture distribution which gives equal weights to its components. We use $\text{Exp}(\eta)$ to denote the exponential distribution, whose CDF is given by $P(Z \leqslant s) = 1 - \exp(-s/\eta)$.

## 3  Dual view of Perturbation as Regularization

In this section, we present a key result which shows that when the sequence of loss functions are convex, every FTPL algorithm is an FTRL algorithm. Our analysis of OFTPL relies on this dual view to obtain tight regret bounds. This duality between FTPL and FTRL was originally studied by Hofbauer and Sandholm [19], where the authors show that any FTPL algorithm, with perturbation distribution admitting a strictly positive density on $\mathbb{R}^d$, is an FTRL algorithm w.r.t some convex regularizer. However, many popular perturbation distributions such as exponential and uniform distributions don't have a strictly positive density. In a recent work, Abernethy et al. [20] point out that the duality between FTPL and FTRL holds for very general perturbation distributions. However, the authors do not provide a formal theorem showing this result. Here, we provide a proposition formalizing the claim of [20].

**Proposition 3.1.** *Consider the problem of online convex learning, where the sequence of loss functions $\{f_t\}_{t=1}^{T}$ encountered by the learner are convex. Consider the deterministic version of FTPL algorithm, where the learner predicts $\mathbf{x}_t$ as $\mathbb{E}_\sigma\left[\operatorname{argmin}_{\mathbf{x}\in\mathcal{X}} \langle \nabla_{1:t-1} - \sigma, \mathbf{x}\rangle\right]$. Suppose the perturbation distribution is absolutely continuous w.r.t the Lebesgue measure. Then there exists a convex regularizer $R : \mathbb{R}^d \to \mathbb{R} \cup \{\infty\}$, with domain $\text{dom}(R) \subseteq \mathcal{X}$, such that $\mathbf{x}_t = \operatorname{argmin}_{\mathbf{x}\in\mathcal{X}} \langle \nabla_{1:t-1}, \mathbf{x}\rangle + R(\mathbf{x})$. Moreover, $-\nabla_{1:t-1} \in \partial R(\mathbf{x}_t)$, and $\mathbf{x}_t = \partial R^{-1}\left(-\nabla_{1:t-1}\right)$, where $\partial R^{-1}$ is the inverse of $\partial R$ in the sense of multivalued mappings.*

**Algorithm 1** Convex OFTPL
---
1: **Input:** Perturbation Distribution $P_{\text{PRTB}}$, number of samples $m$, number of iterations $T$
2: Denote $\nabla_0 = 0$
3: **for** $t = 1 \ldots T$ **do**
4:      Let $g_t$ be the guess for $\nabla_t$
5:      **for** $j = 1 \ldots m$ **do**
6:          Sample $\sigma_{t,j} \sim P_{\text{PRTB}}$
7:          $\mathbf{x}_{t,j} \in \operatorname{argmin}_{\mathbf{x} \in \mathcal{X}} \langle \nabla_{0:t-1} + g_t - \sigma_{t,j}, \mathbf{x} \rangle$
8:      **end for**
9:      Let $\mathbf{x}_t = \frac{1}{m} \sum_{j=1}^{m} \mathbf{x}_{t,j}$
10:     Play $\mathbf{x}_t$ and observe loss function $f_t$
11: **end for**
---

# 4 Online Learning with OFTPL

## 4.1 Online Convex Learning

In this section, we present the OFTPL algorithm for online convex learning and derive an upper bound on its regret. The algorithm we consider is similar to the OFTRL algorithm (see Algorithm 1). Let $g_t[f_1 \ldots f_{t-1}]$ be our guess for $\nabla_t$ at the beginning of round $t$, with $g_1 = 0$. To simplify the notation, in the sequel, we suppress the dependence of $g_t$ on $\{f_i\}_{i=1}^{t-1}$. Given $g_t$, we predict $\mathbf{x}_t$ in OFTPL as follows. We sample independent perturbations $\{\sigma_{t,j}\}_{j=1}^{m}$ from the perturbation distribution $P_{\text{PRTB}}$ and compute $\mathbf{x}_t$ as $m^{-1} \sum_{j=1}^{m} \mathbf{x}_{t,j}$, where $\mathbf{x}_{t,j}$ is a minimizer of the following linear optimization problem

$$\mathbf{x}_{t,j} \in \operatorname*{argmin}_{\mathbf{x} \in \mathcal{X}} \langle \nabla_{1:t-1} + g_t - \sigma_{t,j}, \mathbf{x} \rangle.$$

We now present our main theorem which bounds the regret of OFTPL. A key quantity the regret depends on is the *stability* of predictions of the deterministic version of OFTPL. Intuitively, an algorithm is stable if its predictions in two consecutive iterations differ by a small quantity. To capture this notion, we first define function $\nabla \Phi : \mathbb{R}^d \to \mathbb{R}^d$ as: $\nabla \Phi(g) = \mathbb{E}_\sigma \left[ \operatorname{argmin}_{\mathbf{x} \in \mathcal{X}} \langle g - \sigma, \mathbf{x} \rangle \right]$. Observe that $\nabla \Phi(\nabla_{1:t-1} + g_t)$ is the prediction of the deterministic version of OFTPL. We say the predictions of OFTPL are stable, if $\nabla \Phi$ is a Lipschitz function.

**Definition 4.1** (Stability). The predictions of OFTPL are said to be $\beta$-stable w.r.t some norm $\| \cdot \|$, if

$$\forall g_1, g_2 \in \mathbb{R}^d \quad \| \nabla \Phi(g_1) - \nabla \Phi(g_2) \|_* \leqslant \beta \| g_1 - g_2 \|.$$

**Theorem 4.1.** *Suppose the perturbation distribution $P_{PRTB}$ is absolutely continuous w.r.t Lebesgue measure. Let $D$ be the diameter of $\mathcal{X}$ w.r.t $\| \cdot \|$, which is defined as $D = \sup_{\mathbf{x}_1, \mathbf{x}_2 \in \mathcal{X}} \| \mathbf{x}_1 - \mathbf{x}_2 \|$. Let $\eta = \mathbb{E}_\sigma \left[ \| \sigma \|_* \right]$, and suppose the predictions of OFTPL are $C\eta^{-1}$-stable w.r.t $\| \cdot \|_*$, where $C$ is a constant that depends on the set $\mathcal{X}$. Finally, suppose the sequence of loss functions $\{f_t\}_{t=1}^{T}$ are convex, Holder smooth and satisfy*

$$\forall \mathbf{x}_1, \mathbf{x}_2 \in \mathcal{X} \quad \| \nabla f_t(\mathbf{x}_1) - \nabla f_t(\mathbf{x}_2) \|_* \leqslant L \| \mathbf{x}_1 - \mathbf{x}_2 \|^\alpha,$$

*for some constant $\alpha \in [0, 1]$. Then the expected regret of Algorithm 1 satisfies*

$$\sup_{\mathbf{x} \in \mathcal{X}} \mathbb{E} \left[ \sum_{t=1}^{T} f_t(\mathbf{x}_t) - f_t(\mathbf{x}) \right] \leqslant \eta D + \sum_{t=1}^{T} \frac{C}{2\eta} \mathbb{E} \left[ \| \nabla_t - g_t \|_*^2 \right] - \sum_{t=1}^{T} \frac{\eta}{2C} \mathbb{E} \left[ \| \mathbf{x}_t^\infty - \tilde{\mathbf{x}}_{t-1}^\infty \|^2 \right]$$

$$+ LT \left( \frac{\Psi_1 \Psi_2 D}{\sqrt{m}} \right)^{1+\alpha}.$$

*where $\mathbf{x}_t^\infty = \mathbb{E} \left[ \mathbf{x}_t | g_t, f_{1:t-1}, \mathbf{x}_{1:t-1} \right]$ and $\tilde{\mathbf{x}}_{t-1}^\infty = \mathbb{E} \left[ \tilde{\mathbf{x}}_{t-1} | f_{1:t-1}, \mathbf{x}_{1:t-1} \right]$ and $\tilde{\mathbf{x}}_{t-1}$ denotes the prediction in the $t^{th}$ iteration of Algorithm 1, if guess $g_t = 0$ was used. Here, $\Psi_1, \Psi_2$ denote the norm compatibility constants of $\| \cdot \|$.*

***Proof Sketch***. For any $\mathbf{x} \in \mathcal{X}$, we have

$$\sum_{t=1}^{T} \mathbb{E} \left[ f_t(\mathbf{x}_t) - f_t(\mathbf{x}) \right] \overset{(a)}{\leqslant} \sum_{t=1}^{T} \mathbb{E} \left[ \langle \mathbf{x}_t - \mathbf{x}, \nabla_t \rangle \right] = \sum_{t=1}^{T} \mathbb{E} \left[ \langle \mathbf{x}_t - \mathbf{x}_t^\infty, \nabla_t \rangle \right] + \mathbb{E} \left[ \langle \mathbf{x}_t^\infty - \mathbf{x}, \nabla_t \rangle \right],$$

---

**Algorithm 2** Nonconvex OFTPL

1: **Input:** Perturbation Distribution $P_{\text{PRTB}}$, number of samples $m$, number of iterations $T$
2: Denote $f_0 = 0$
3: **for** $t = 1 \dots T$ **do**
4:     Let $g_t$ be the guess for $f_t$
5:     **for** $j = 1 \dots m$ **do**
6:         Sample $\sigma_{t,j} \sim P_{\text{PRTB}}$
7:         $\mathbf{x}_{t,j} \in \operatorname{argmin}_{\mathbf{x} \in \mathcal{X}} f_{0:t-1}(\mathbf{x}) + g_t(\mathbf{x}) - \sigma_{t,j}(\mathbf{x})$
8:     **end for**
9:     Let $P_t$ be the empirical distribution over $\{\mathbf{x}_{t,1}, \mathbf{x}_{t,2} \dots \mathbf{x}_{t,m}\}$
10:    Play $\mathbf{x}_t$, a random sample generated from $P_t$
11:    Observe loss function $f_t$
12: **end for**

---

where $(a)$ follows from convexity of $f_t$. The first term in the RHS involves the difference between the iterates of stochastic and deterministic versions of OFTPL. To bound this term, we use the following two facts: (a) conditioned on the past randomness, $\mathbf{x}_t - \mathbf{x}_t^\infty$ is the average of $m$ i.i.d, bounded, mean 0 random variables and so its variance is $O(D^2/m)$, (b) $f_t$ is Holder smooth. Using these two facts, we get $\mathbb{E}\left[\langle \mathbf{x}_t - \mathbf{x}_t^\infty, \nabla_t \rangle\right] = O\left(D/\sqrt{m}\right)^{1+\alpha}$. The second term in the RHS is related to the regret of deterministic OFTPL. To bound this term, we rely on the duality between deterministic OFTPL and OFTRL (Proposition 3.1), and use a similar proof technique as the one used to derive regret bounds of OFTRL. One key distinction between OFTRL and OFTPL is that in OFTRL it is typically assumed that the regularizer $R$ is differentiable. However, the regularizer corresponding to OFTPL need not be differentiable. As a result, the traditional Bregmann divergence used in the analysis of OFTRL is not well defined. So, our analysis instead relies on "pseudo" Bregmann divergence, which is obtained by replacing the gradient of $R$ in Bregmann divergence with an appropriately chosen sub-gradient. $\quad\square$

Regret bounds that hold with high probability can be found in Appendix G. The above Theorem shows that the regret of OFTPL only depends on $\|\nabla_t - g_t\|_*$, which quantifies the accuracy of our guess $g_t$. In contrast, the regret of FTPL depends on $\|\nabla_t\|_*$ [7]. This shows that for predictable sequences, with an appropriate choice of $g_t$, OFTPL can achieve better regret guarantees than FTPL. As we demonstrate in Section 5, this helps us design faster algorithms for solving minimax games.

Note that the above result is very general and holds for any absolutely continuous perturbation distribution. The key challenge in instantiating this result for any particular perturbation distribution is in showing the stability of predictions. Several past works have studied the stability of FTPL for various perturbation distributions such as uniform, exponential, Gumbel distributions [5, 7, 10]. Consequently, the above result can be used to derive tight regret bounds for all these perturbation distributions. As one particular instantiation of Theorem 4.1, we consider the special case of $g_t = 0$ and derive regret bounds for FTPL, when the perturbation distribution is the uniform distribution over a ball centered at the origin.

**Corollary 4.1** (FTPL). *Suppose the perturbation distribution is equal to the uniform distribution over $\{\mathbf{x} : \|\mathbf{x}\|_2 \leqslant (1 + d^{-1})\eta\}$. Let $D$ be the diameter of $\mathcal{X}$ w.r.t $\|\cdot\|_2$. Then $\mathbb{E}_\sigma \left[\|\sigma\|_2\right] = \eta$, and the predictions of OFTPL are $dD\eta^{-1}$-stable w.r.t $\|\cdot\|_2$. Suppose, the sequence of loss functions $\{f_t\}_{t=1}^T$ are $G$-Lipschitz and satisfy $\sup_{\mathbf{x} \in \mathcal{X}} \|\nabla f_t(\mathbf{x})\|_2 \leqslant G$. Moreover, suppose $f_t$ satisfies the Holder smooth condition in Theorem 4.1 w.r.t $\|\cdot\|_2$ norm. Then the expected regret of Algorithm 1 with guess $g_t = 0$, satisfies*

$$\sup_{\mathbf{x} \in \mathcal{X}} \mathbb{E}\left[\sum_{t=1}^T f_t(\mathbf{x}_t) - f_t(\mathbf{x})\right] \leqslant \eta D + \frac{dDG^2 T}{2\eta} + LT\left(\frac{D}{\sqrt{m}}\right)^{1+\alpha}.$$

This recovers the regret bounds of FTPL for general convex loss functions derived by [10].

## 4.2 Online Nonconvex Learning

We now study OFTPL in the nonconvex setting. In this setting, we assume the sequence of loss functions belong to some function class $\mathcal{F}$ containing real-valued measurable functions on $\mathcal{X}$. Some

popular choices for $\mathcal{F}$ include the set of Lipschitz functions, the set of bounded functions. The OFTPL algorithm in this setting is described in Algorithm 2. Similar to the convex case, we first sample random perturbation functions $\{\sigma_{t,j}\}_{j=1}^{m}$ from some distribution $P_{\text{PRTB}}$. Some examples of perturbation functions that have been considered in the past include $\sigma_{t,j}(\mathbf{x}) = \langle \bar{\sigma}_{t,j}, \mathbf{x} \rangle$, for some random vector $\bar{\sigma}_{t,j}$ sampled from exponential or uniform distributions [11, 18]. Another popular choice for $\sigma_{t,j}$ is the Gumbel process, which results in the continuous exponential weights algorithm [21]. Letting, $g_t$ be our guess of loss function $f_t$ at the beginning of round $t$, the learner first computes $\mathbf{x}_{t,j}$ as $\operatorname{argmin}_{\mathbf{x} \in \mathcal{X}} \sum_{i=1}^{t-1} f_i(\mathbf{x}) + g_t(\mathbf{x}) - \sigma_{t,j}(\mathbf{x})$. We assume access to an optimization oracle which computes a minimizer of this problem. We often refer to this oracle as the *perturbed best response* oracle. Let $P_t$ denote the empirical distribution of $\{\mathbf{x}_{t,j}\}_{j=1}^{m}$. The learner then plays an $\mathbf{x}_t$ which is sampled from $P_t$. Algorithm 2 describes this procedure. We note that for the online learning problem, $m = 1$ suffices, as the expected loss suffered by the learner in each round is independent of $m$; that is $\mathbb{E}\left[f_t(\mathbf{x}_t)\right] = \mathbb{E}\left[f_t(\mathbf{x}_{t,1})\right]$. However, the choice of $m$ affects the rate of convergence when Algorithm 2 is used for solving nonconvex nonconcave minimax games.

Before we present the regret bounds, we introduce the *dual space* associated with $\mathcal{F}$. Let $\|\cdot\|_{\mathcal{F}}$ be a seminorm associated with $\mathcal{F}$. For example, when $\mathcal{F}$ is the set of Lipschitz functions, $\|\cdot\|_{\mathcal{F}}$ is the Lipschitz seminorm. Various choices of $(\mathcal{F}, \|\cdot\|_{\mathcal{F}})$ induce various distance metrics on $\mathcal{P}$, the set of all probability distributions on $\mathcal{X}$. We let $\gamma_{\mathcal{F}}$ denote the Integral Probability Metric (IPM) induced by $(\mathcal{F}, \|\cdot\|_{\mathcal{F}})$, which is defined as

$$\gamma_{\mathcal{F}}(P, Q) = \sup_{f \in \mathcal{F}, \|f\|_{\mathcal{F}} \leqslant 1} \left| \mathbb{E}_{\mathbf{x} \sim P}\left[f(\mathbf{x})\right] - \mathbb{E}_{\mathbf{x} \sim Q}\left[f(\mathbf{x})\right] \right|.$$

We often refer to $(\mathcal{P}, \gamma_{\mathcal{F}})$ as the dual space of $(\mathcal{F}, \|\cdot\|_{\mathcal{F}})$. When $\mathcal{F}$ is the set of Lipschitz functions and when $\|\cdot\|_{\mathcal{F}}$ is the Lipschitz seminorm, $\gamma_{\mathcal{F}}$ is the Wasserstein distance. Table 1 in Appendix E.1 presents examples of $\gamma_{\mathcal{F}}$ induced by some popular function spaces. Similar to the convex case, the regret bounds in the nonconvex setting depend on the stability of predictions of OFTPL.

**Definition 4.2** (Stability). Suppose the perturbation function $\sigma(\mathbf{x})$ is sampled from $P_{\text{PRTB}}$. For any $f \in \mathcal{F}$, define random variable $\mathbf{x}_f(\sigma)$ as $\operatorname{argmin}_{\mathbf{x} \in \mathcal{X}} f(\mathbf{x}) - \sigma(\mathbf{x})$. Let $\nabla \Phi(f)$ denote the distribution of $\mathbf{x}_f(\sigma)$. The predictions of OFTPL are said to be $\beta$-stable w.r.t $\|\cdot\|_{\mathcal{F}}$ if

$$\forall f, g \in \mathcal{F} \quad \gamma_{\mathcal{F}}(\nabla \Phi(f), \nabla \Phi(g)) \leqslant \beta \|f - g\|_{\mathcal{F}}.$$

**Theorem 4.2.** *Suppose the sequence of loss functions $\{f_t\}_{t=1}^{T}$ belong to $(\mathcal{F}, \|\cdot\|_{\mathcal{F}})$. Suppose the perturbation distribution $P_{PRTB}$ is such that $\operatorname{argmin}_{\mathbf{x} \in \mathcal{X}} f(\mathbf{x}) - \sigma(\mathbf{x})$ has a unique minimizer with probability one, for any $f \in \mathcal{F}$. Let $\mathcal{P}$ be the set of probability distributions over $\mathcal{X}$. Define the diameter of $\mathcal{P}$ as $D = \sup_{P_1, P_2 \in \mathcal{P}} \gamma_{\mathcal{F}}(P_1, P_2)$. Let $\eta = \mathbb{E}\left[\|\sigma\|_{\mathcal{F}}\right]$. Suppose the predictions of OFTPL are $C\eta^{-1}$-stable w.r.t $\|\cdot\|_{\mathcal{F}}$, for some constant $C$ that depends on $\mathcal{X}$. Then the expected regret of Algorithm 2 satisfies*

$$\sup_{\mathbf{x} \in \mathcal{X}} \mathbb{E}\left[\sum_{t=1}^{T} f_t(\mathbf{x}_t) - f_t(\mathbf{x})\right] \leqslant \eta D + \sum_{t=1}^{T} \frac{C}{2\eta} \mathbb{E}\left[\|f_t - g_t\|_{\mathcal{F}}^2\right] - \sum_{t=1}^{T} \frac{\eta}{2C} \mathbb{E}\left[\gamma_{\mathcal{F}}(P_t^{\infty}, \tilde{P}_{t-1}^{\infty})^2\right],$$

*where $P_t^{\infty} = \mathbb{E}\left[P_t | g_t, f_{1:t-1}, P_{1:t-1}\right]$, $\tilde{P}_t^{\infty} = \mathbb{E}\left[\tilde{P}_{t-1} | f_{1:t-1}, P_{1:t-1}\right]$ and $\tilde{P}_{t-1}$ is the empirical distribution computed in the $t^{th}$ iteration of Algorithm 2, if guess $g_t = 0$ was used.*

***Proof Sketch***. The proof uses similar arguments as in the proof of Theorem 4.1. For any $P \in \mathcal{P}$, we have

$$\mathbb{E}\left[f_t(\mathbf{x}_t) - f_t(P)\right] = \mathbb{E}\left[f_t(P_t) - f_t(P_t^{\infty})\right] + \mathbb{E}\left[f_t(P_t^{\infty}) - f_t(P)\right] \stackrel{(a)}{=} \mathbb{E}\left[f_t(P_t^{\infty}) - f_t(P)\right],$$

where $(a)$ follows from the fact that $\mathbb{E}\left[f_t(P_t) - f_t(P_t^{\infty}) | g_t, f_{1:t-1}, P_{1:t-1}\right] = 0$. The RHS is related to the regret of deterministic OFTPL. In the convex case, to bound this term, we relied on duality between OFTRL and OFTPL. However, in the nonconvex case, we can not take this route as there are no known analogs of Fenchel duality for infinite dimensional function spaces. As a result, more careful analysis is needed to obtain the regret bounds. Our analysis mimics the arguments made in the convex case, albeit without explicitly relying on duality theory. □

As in the convex case, the key challenge in instantiating the above result for any particular perturbation distribution is in showing the stability of predictions. In a recent work, [11] consider

linear perturbation functions $\sigma(\mathbf{x}) = \langle \bar{\sigma}, \mathbf{x} \rangle$, for $\bar{\sigma}$ sampled from exponential distribution, and show stability of FTPL. We now instantiate the above Theorem for this setting.

**Corollary 4.2.** *Consider the setting of Theorem 4.2. Let $\mathcal{F}$ be the set of Lipschitz functions and $\|\cdot\|_{\mathcal{F}}$ be the Lipschitz seminorm, which is defined as $\|f\|_{\mathcal{F}} = \sup_{\mathbf{x} \neq \mathbf{y} \text{ in } \mathcal{X}} |f(\mathbf{x}) - f(\mathbf{y})|/\|\mathbf{x} - \mathbf{y}\|_1$. Suppose the perturbation function is such that $\sigma(\mathbf{x}) = \langle \bar{\sigma}, \mathbf{x} \rangle$, where $\bar{\sigma} \in \mathbb{R}^d$ is a random vector whose entries are sampled independently from $Exp(\eta)$. Then $\mathbb{E}_{\sigma}[\|\sigma\|_{\mathcal{F}}] = \eta \log d$, and the predictions of OFTPL are $O\left(d^2 D \eta^{-1}\right)$-stable w.r.t $\|\cdot\|_{\mathcal{F}}$. Moreover, the expected regret of Algorithm 2 is upper bounded by $O\left(\eta D \log d + \sum_{t=1}^{T} \frac{d^2 D}{\eta} \mathbb{E}\left[\|f_t - g_t\|_{\mathcal{F}}^2\right] - \sum_{t=1}^{T} \frac{\eta}{d^2 D} \mathbb{E}\left[\gamma_{\mathcal{F}}(P_t^{\infty}, \tilde{P}_{t-1}^{\infty})^2\right]\right).$*

We note that the above regret bounds are tighter than the regret bounds of [11], where the authors show that the regret of OFTPL is bounded by $O\left(\eta D \log d + \sum_{t=1}^{T} \frac{d^2 D}{\eta} \mathbb{E}\left[\|f_t - g_t\|_{\mathcal{F}}^2\right]\right)$. These tigher bounds help us design faster algorithms for solving minimax games in the nonconvex setting (see Section 5 for a more detailed discussion).

# 5 Minimax Games

We now consider the problem of solving minimax games of the following form

$$\min_{\mathbf{x} \in \mathcal{X}} \max_{\mathbf{y} \in \mathcal{Y}} f(\mathbf{x}, \mathbf{y}). \tag{1}$$

Nash equilibria of such games can be computed by playing two online learning algorithms against each other [6, 7]. In this work, we study the algorithm where both the players employ OFTPL to decide their actions in each round. For convex-concave games, both the players use the OFTPL algorithm described in Algorithm 1 (see Algorithm 3 in Appendix D). The following theorem derives the rate of convergence of this algorithm to a Nash equilibirum (NE).

**Theorem 5.1.** *Consider the minimax game in Equation (1). Suppose both the domains $\mathcal{X}, \mathcal{Y}$ are compact subsets of $\mathbb{R}^d$, with diameter $D = \max\{\sup_{\mathbf{x}_1, \mathbf{x}_2 \in \mathcal{X}} \|\mathbf{x}_1 - \mathbf{x}_2\|_2, \sup_{\mathbf{y}_1, \mathbf{y}_2 \in \mathcal{Y}} \|\mathbf{y}_1 - \mathbf{y}_2\|_2\}$. Suppose $f$ is convex in $\mathbf{x}$, concave in $\mathbf{y}$ and is smooth w.r.t $\|\cdot\|_2$*

$$\|\nabla_{\mathbf{x}} f(\mathbf{x}, \mathbf{y}) - \nabla_{\mathbf{x}} f(\mathbf{x}', \mathbf{y}')\|_2 + \|\nabla_{\mathbf{y}} f(\mathbf{x}, \mathbf{y}) - \nabla_{\mathbf{y}} f(\mathbf{x}', \mathbf{y}')\|_2 \leqslant L\|\mathbf{x} - \mathbf{x}'\|_2 + L\|\mathbf{y} - \mathbf{y}'\|_2.$$

*Suppose Algorithm 3 is used to solve the minimax game. Suppose the perturbation distributions used by both the players are the same and equal to the uniform distribution over $\{\mathbf{x} : \|\mathbf{x}\|_2 \leqslant (1 + d^{-1})\eta\}$. Suppose the guesses used by $\mathbf{x}, \mathbf{y}$ players in the $t^{th}$ iteration are $\nabla_{\mathbf{x}} f(\tilde{\mathbf{x}}_{t-1}, \tilde{\mathbf{y}}_{t-1}), \nabla_{\mathbf{y}} f(\tilde{\mathbf{x}}_{t-1}, \tilde{\mathbf{y}}_{t-1})$, where $\tilde{\mathbf{x}}_{t-1}, \tilde{\mathbf{y}}_{t-1}$ denote the predictions of $\mathbf{x}, \mathbf{y}$ players in the $t^{th}$ iteration, if guess $g_t = 0$ was used. If Algorithm 3 is run with $\eta = 6dD(L+1), m = T$, then the iterates $\{(\mathbf{x}_t, \mathbf{y}_t)\}_{t=1}^{T}$ satisfy*

$$\sup_{\mathbf{x} \in \mathcal{X}, \mathbf{y} \in \mathcal{Y}} \mathbb{E}\left[f\left(\frac{1}{T}\sum_{t=1}^{T} \mathbf{x}_t, \mathbf{y}\right) - f\left(\mathbf{x}, \frac{1}{T}\sum_{t=1}^{T} \mathbf{y}_t\right)\right] = O\left(\frac{dD^2(L+1)}{T}\right).$$

Rates of convergence which hold with high probability can be found in Appendix G. We note that Theorem 5.1 can be extended to more general noise distributions and settings where gradients of $f$ are Holder smooth w.r.t non-Euclidean norms, and $\mathcal{X}, \mathcal{Y}$ lie in spaces of different dimensions (see Theorem D.1 in Appendix). We now discuss the above result.

- Theorem 5.1 shows that for smooth convex-concave games, Algorithm 3 converges to a NE at $O\left(T^{-1}\right)$ rate using $4T^2$ calls to the linear optimization oracle. Moreover, the algorithm runs in $T$ iterations, with each iteration making $4T$ parallel calls to the optimization oracle. In contrast, FTPL makes $2T^3$ calls to the linear optimization oracle to achieve $O\left(T^{-1}\right)$ rates of convergence and runs for $T^2$ iterations, with each iteration making $2T$ parallel calls to the optimization oracle. This can be obtained by setting $m = \sqrt{T}, \alpha = 1$, and $\eta = O\left(\sqrt{T}\right)$ in Corollary 4.1.

- The Frank-Wolfe technique of He and Harchaoui [15] achieves the same convergence rates as our algorithm; that is, it achieves $O\left(T^{-1}\right)$ rates using $T^2$ calls to the linear optimization oracle. However, unlike [15], our algorithm is parallelizable and can be run in $T$ iterations.

- He and Harchaoui [15] achieve dimension free convergence rates in the Euclidean setting, where the smoothness is measured w.r.t $\|\cdot\|_2$ norm. In contrast, the rates of convergence of our algorithm depend on the dimension. We believe the dimension dependence in the rates can be removed by appropriately choosing the perturbation distributions based on domains $\mathcal{X}, \mathcal{Y}$ (see Appendix F).

- Note that OFTRL also achieves $O\left(T^{-1}\right)$ rates of convergence after $T$ iterations. However, each iteration of OFTRL involves optimization of a non-linear convex function over the domains $\mathcal{X}, \mathcal{Y}$, which can be quite expensive in practice.

We now consider the more general nonconvex-nonconcave games. In this case, both the players use the nonconvex OFTPL algorithm described in Algorithm 2 to choose their actions. Instead of generating a single sample from the empirical distribution $P_t$ computed in $t^{th}$ iteration of Algorithm 2, the players now play the entire distribution $P_t$ (see Algorithm 4 in Appendix E). Letting $\{P_t\}_{t=1}^T, \{Q_t\}_{t=1}^T$, be the sequence of iterates generated by the $\mathbf{x}$ and $\mathbf{y}$ players, the following theorem shows that $\left(\frac{1}{T}\sum_{t=1}^T P_t, \frac{1}{T}\sum_{t=1}^T Q_t\right)$ converges to a NE.

**Theorem 5.2.** *Consider the minimax game in Equation (1). Suppose the domains $\mathcal{X}, \mathcal{Y}$ are compact subsets of $\mathbb{R}^d$ with diameter $D = \max\{\sup_{\mathbf{x}_1,\mathbf{x}_2\in\mathcal{X}} \|\mathbf{x}_1 - \mathbf{x}_2\|_1, \sup_{\mathbf{y}_1,\mathbf{y}_2\in\mathcal{Y}} \|\mathbf{y}_1 - \mathbf{y}_2\|_1\}$. Suppose $f$ is Lipschitz w.r.t $\|\cdot\|_1$ and satisfies*

$$\max\left\{\sup_{\mathbf{x}\in\mathcal{X},\mathbf{y}\in\mathcal{Y}} \|\nabla_{\mathbf{x}}f(\mathbf{x},\mathbf{y})\|_\infty, \sup_{\mathbf{x}\in\mathcal{X},\mathbf{y}\in\mathcal{Y}} \|\nabla_{\mathbf{y}}f(\mathbf{x},\mathbf{y})\|_\infty\right\} \leqslant G.$$

*Moreover, suppose $f$ satisfies the following smoothness property*

$$\|\nabla_{\mathbf{x}}f(\mathbf{x},\mathbf{y}) - \nabla_{\mathbf{x}}f(\mathbf{x}',\mathbf{y}')\|_\infty + \|\nabla_{\mathbf{y}}f(\mathbf{x},\mathbf{y}) - \nabla_{\mathbf{y}}f(\mathbf{x}',\mathbf{y}')\|_\infty \leqslant L\|\mathbf{x} - \mathbf{x}'\|_1 + L\|\mathbf{y} - \mathbf{y}'\|_1.$$

*Suppose both $\mathbf{x}$ and $\mathbf{y}$ players use Algorithm 4 to solve the game with linear perturbation functions $\sigma(\mathbf{z}) = \langle\bar{\sigma}, \mathbf{z}\rangle$, where $\bar{\sigma} \in \mathbb{R}^d$ is such that each of its entries is sampled independently from $Exp(\eta)$. Suppose the guesses used by $\mathbf{x}$ and $\mathbf{y}$ players in the $t^{th}$ iteration are $f(\cdot, \tilde{Q}_{t-1}), f(\tilde{P}_{t-1}, \cdot)$, where $\tilde{P}_{t-1}, \tilde{Q}_{t-1}$ denote the predictions of $\mathbf{x}, \mathbf{y}$ players in the $t^{th}$ iteration, if guess $g_t = 0$ was used. If Algorithm 4 is run with $\eta = 10d^2 D(L+1), m = T$, then the iterates $\{(P_t, Q_t)\}_{t=1}^T$ satisfy*

$$\sup_{\mathbf{x}\in\mathcal{X},\mathbf{y}\in\mathcal{Y}} \mathbb{E}\left[f\left(\frac{1}{T}\sum_{t=1}^T P_t, \mathbf{y}\right) - f\left(\mathbf{x}, \frac{1}{T}\sum_{t=1}^T Q_t\right)\right] = O\left(\frac{d^2 D^2(L+1)\log d}{T}\right)$$

$$+ O\left(\min\left\{D^2 L, \frac{d^2 G^2\log T}{LT}\right\}\right).$$

More general versions of the Theorem, which consider other function classes and general perturbation distributions, can be found in Appendix E. We now discuss the above result.

- Theorem 5.2 shows that Algorithm 4 converges to a NE at $\tilde{O}\left(T^{-1}\right)$ rate using $T^2$ calls to the perturbed best response oracle. This matches the rates of convergence of FTPL [11]. However, the key advantage of our algorithm is that it is highly parallelizable and runs in $T$ iterations, in contrast to FTPL, which runs in $T^2$ iterations.

- As previously stated, [11] also study OFTPL for non-convex losses and upper bound its regret as $O\left(\eta D\log d + \sum_{t=1}^T \frac{d^2 D}{\eta}\mathbb{E}\left[\|f_t - g_t\|_{\mathcal{F}}^2\right]\right)$. However, by relying on this regret bound, we can only obtain $O\left(T^{-3/4}\right)$ convergence rates, even if we set $m = \infty$ and run the algorithm for $T$ iterations (see Appendix E.4).

# 6 Conclusion

We studied an optimistic variant of FTPL which achieves better regret guarantees when the sequence of loss functions is predictable. As one specific application of our algorithm, we considered the problem of solving minimax games. For solving convex-concave games, our algorithm requires access to a linear optimization oracle and for nonconvex-nonconcave games our algorithm requires access to a more powerful perturbed best response oracle. In both these settings, our algorithm achieves $O\left(T^{-1/2}\right)$ convergence rates using $T$ calls to the oracles. Moreover, our algorithm runs in $O\left(T^{1/2}\right)$ iterations, with each iteration making $O\left(T^{1/2}\right)$ parallel calls to the optimization oracle. We believe our improved algorithms for solving minimax games are useful in a number of modern machine learning applications such as training of GANs, adversarial training, which involve solving nonconvex-nonconcave minimax games and often deal with huge datasets.

## Broader Impact

Many problems in machine learning and statistics have a game theoretic component to them. Two popular modern applications that illustrate this are adversarial training and density estimation using IPMs. These applications often involve solving large-scale minimax games. In many cases these games are nonconvex-nonconcave, which makes it even more harder to find a NE. Existing approaches for solving these games have mostly relied on algorithms from online convex learning. However, such algorithms are not guaranteed to converge to a NE of nonconvex-nonconcave games. As a result, there is a need for faster algorithms for provably solving large-scale nonconvex-nonconcave games. Our work takes a first step towards this goal by proposing fast parallelizable algorithms which provably converge to a NE in both convex and nonconvex settings.

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
