[Supplementary Material]

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

# Contents

# A  Dual view of Perturbations as Regularization

## A.1  Proof of Theorem 3.1

We first define a convex function $\Psi : \mathbb{R}^d \to \mathbb{R}$ as

$$\Psi(f) = \mathbb{E}_\sigma\left[\sup_{\mathbf{x}\in\mathcal{X}}\langle f + \sigma, \mathbf{x}\rangle\right] = \mathbb{E}_\sigma\left[\sup_{\mathbf{x}\in\mathcal{X}}\langle f + \sigma, \mathbf{x}\rangle\right],$$

where perturbation $\sigma$ follows probability distribution $P_{\text{PRTB}}$ which is absolutely continuous w.r.t the Lebesgue measure. For our choice of $P_{\text{PRTB}}$, we now show that $\Psi$ is differentiable. Consider the function $\psi(g) = \sup_{\mathbf{x}\in\mathcal{X}}\langle g, \mathbf{x}\rangle$. Since $\psi(g)$ is a proper convex function, we know that it is differentiable almost everywhere, except on a set of Lebesgue measure $0$ [see Theorem 25.5 of 22]. Moreover, it is easy to verify that $\text{argmax}_{\mathbf{x}\in\mathcal{X}}\langle g, \mathbf{x}\rangle \in \partial\psi(g)$. These two observations, together with the fact that $P_{\text{PRTB}}$ is absolutely continuous, show that the $\sup$ expression inside the expectation of $\Psi$ has a unique maximizer with probability one.

Since the sup expression inside the expectation has a unique maximizer with probability $1$, we can swap the expectation and gradient to obtain [see Proposition 2.2 of 23]

$$\nabla\Psi(f) = \mathbb{E}_\sigma\left[\text{argmax}_{\mathbf{x}\in\mathcal{X}}\langle f + \sigma, \mathbf{x}\rangle\right]. \tag{2}$$

Note that $\nabla\Psi$ is related to the prediction of deterministic version of FTPL. Specifically, $\nabla\Psi(-\nabla_{1:t-1})$ is the prediction of deterministic FTPL in the $t^{th}$ iteration. We now show that $\nabla\Psi(f) = \text{argmin}_{\mathbf{x}\in\mathcal{X}}\langle -f, \mathbf{x}\rangle + R(\mathbf{x})$, for some convex function $R$.

Since all differentiable functions are closed, $\Psi(f)$ is a proper, closed and differentiable convex function over $\mathbb{R}^d$. Let $R(\mathbf{x})$ denote the Fenchel conjugate of $\Psi(f)$

$$R(\mathbf{x}) = \sup_{f\in\text{dom}(\Psi)}\langle \mathbf{x}, f\rangle - \Psi(f),$$

where $\text{dom}(\Psi)$ denotes the domain of $\Psi$. Following Theorem H.1 (see Appendix H), $\Psi(f)$ is the Fenchel conjugate of $R(\mathbf{x})$

$$\Psi(f) = \sup_{\mathbf{x}\in\text{dom}(R)}\langle f, \mathbf{x}\rangle - R(\mathbf{x}).$$

Furthermore, from Theorem H.2 we have

$$\nabla\Psi(f) = \text{argmax}_{\mathbf{x}\in\text{dom}(R)}\langle f, \mathbf{x}\rangle - R(\mathbf{x}).$$

We now show that the domain of $R$ is a subset of $\mathcal{X}$. This, together with the previous two equations, would then immediately imply

$$\Psi(f) = \sup_{\mathbf{x}\in\mathcal{X}}\langle f, \mathbf{x}\rangle - R(\mathbf{x}), \tag{3}$$

$$\nabla\Psi(f) = \text{argmax}_{\mathbf{x}\in\mathcal{X}}\langle f, \mathbf{x}\rangle - R(\mathbf{x}). \tag{4}$$

From Theorem H.4, we know that the domain of $R$ satisfies

$$\text{ri}(\text{dom}(R)) \subseteq \text{range}\nabla\Psi \subseteq \text{dom}(R),$$

where $\text{ri}(A)$ denotes the relative interior of a set $A$. Moreover, from the definition of $\nabla\Psi(f)$ in Equation (2), we have $\text{range}\nabla\Psi \subseteq \mathcal{X}$. Combining these two properties, we can show that one of the following statements is true

$$\text{ri}(\text{dom}(R)) \subseteq \text{range}\nabla\Psi \subseteq \mathcal{X} \subseteq \text{dom}(R),$$
$$\text{ri}(\text{dom}(R)) \subseteq \text{range}\nabla\Psi \subseteq \text{dom}(R) \subseteq \mathcal{X}.$$

Suppose the first statement is true. Since $\mathcal{X}$ is a compact set, it is easy to see that $\mathcal{X} = \text{dom}(R)$. If the second statement is true, then $\text{dom}(R) \subseteq \mathcal{X}$. Together, these two statements imply $\text{dom}(R) \subseteq \mathcal{X}$.

**Connecting back to FTPL.** We now connect the above results to FTPL. From Equation (2), we know that the prediction at iteration $t$ of deterministic FTPL is equal to $\nabla\Psi(-\nabla_{1:t-1})$. From Equation (4), $\nabla\Psi(-\nabla_{1:t-1})$ is defined as

$$\mathbf{x}_t = \nabla\Psi(-\nabla_{1:t-1}) = \underset{\mathbf{x}\in\mathcal{X}}{\operatorname{argmax}}\,\langle -\nabla_{1:t-1}, \mathbf{x}\rangle - R(\mathbf{x}).$$

This shows that

$$\mathbf{x}_t = \underset{\mathbf{x}\in\mathcal{X}}{\operatorname{argmin}}\,\langle \nabla_{1:t-1}, \mathbf{x}\rangle + R(\mathbf{x}).$$

So the prediction of FTPL can also be obtained using FTRL for some convex regularizer $R(\mathbf{x})$. Finally, to show that $-\nabla_{1:t-1} \in \partial R(\mathbf{x}_t), \mathbf{x}_t = \partial R^{-1}\left(-\nabla_{1:t-1}\right)$, we rely on Theorem H.3. Since $\mathbf{x}_t = \nabla\Psi(-\nabla_{1:t-1})$, from Theorem H.3, we have

$$-\nabla_{1:t-1} \in \partial R(\mathbf{x}_t), \quad \mathbf{x}_t = \nabla\Psi(-\nabla_{1:t-1}) = \partial R^{-1}\left(-\nabla_{1:t-1}\right),$$

where $\partial R^{-1}$ is the inverse of $\partial R$ in the sense of multivalued mappings. Note that, even though $\partial R$ can be a multivalued mapping, its inverse $\partial R^{-1} = \nabla\Psi$ is a singlevalued mapping (this follows form differentiability of $\Psi$). This finishes the proof of the Theorem.

# B Online Convex Learning

## B.1 Proof of Theorem 4.1

Before presenting the proof of the Theorem, we introduce some notation.

### B.1.1 Notation

We define functions $\Phi : \mathbb{R}^d \to \mathbb{R}$, $R : \mathbb{R}^d \to \mathbb{R}$ as follows

$$\Phi(f) = \mathbb{E}_\sigma\left[\inf_{\mathbf{x}\in\mathcal{X}}\langle f-\sigma, \mathbf{x}\rangle\right], \quad R(\mathbf{x}) = \sup_{f\in\mathbb{R}^d}\langle f, \mathbf{x}\rangle + \Phi(-f).$$

Note that $\Phi$ is related to the function $\Psi$ defined in the proof of Proposition 3.1. To be precise, $\Psi(f) = -\Phi(-f)$. Moreover, $R(\mathbf{x})$ is the Fenchel conjugate of $\Psi$. For our choice of perturbation distribution, $\Psi$ is differentiable (see proof of Proposition 3.1). This implies $\Phi$ is also differentiable with gradient $\nabla\Phi$ defined as

$$\nabla\Phi(f) = \mathbb{E}_\sigma\left[\underset{\mathbf{x}\in\mathcal{X}}{\operatorname{argmin}}\langle f-\sigma, \mathbf{x}\rangle\right].$$

Note that $\nabla\Phi$ is the prediction of deterministic version of FTPL. In Proposition 3.1 we showed that

$$\nabla\Phi(f) = \underset{\mathbf{x}\in\mathcal{X}}{\operatorname{argmin}}\langle f, \mathbf{x}\rangle + R(\mathbf{x}).$$

### B.1.2 Main Argument

Since $\mathbf{x}_t^\infty$ is the prediction of deterministic version of FTPL, following FTPL-FTRL duality proved in Proposition 3.1, $\mathbf{x}_t^\infty$ can equivalently be written as

$$\mathbf{x}_t^\infty = \nabla\Phi\left(\nabla_{1:t-1} + g_t\right) = \underset{\mathbf{x}\in\mathcal{X}}{\operatorname{argmin}}\langle \nabla_{1:t-1} + g_t, \mathbf{x}\rangle + R(\mathbf{x}).$$

Similarly, $\tilde{\mathbf{x}}_t^\infty$ can be written as

$$\tilde{\mathbf{x}}_t^\infty = \nabla\Phi\left(\nabla_{1:t}\right) = \underset{\mathbf{x}\in\mathcal{X}}{\operatorname{argmin}}\langle \nabla_{1:t}, \mathbf{x}\rangle + R(\mathbf{x}).$$

We use the notation $\nabla_{1:0} = 0$. So $\tilde{\mathbf{x}}_0^\infty, \mathbf{x}_1^\infty$ are equal to $\operatorname{argmin}_{\mathbf{x}\in\mathcal{X}} R(\mathbf{x})$. From the first order optimality conditions, we have

$$-\nabla_{1:t-1} - g_t \in \partial R\left(\mathbf{x}_t^\infty\right), \quad -\nabla_{1:t} \in \partial R\left(\tilde{\mathbf{x}}_t^\infty\right).$$

Define functions $B(\cdot, \mathbf{x}_t^\infty), B(\cdot, \tilde{\mathbf{x}}_t^\infty)$ for any $t \in [T]$ as

$$B(\mathbf{x}, \mathbf{x}_t^\infty) = R(\mathbf{x}) - R(\mathbf{x}_t^\infty) + \langle \nabla_{1:t-1} + g_t, \mathbf{x} - \mathbf{x}_t^\infty\rangle,$$
$$B(\mathbf{x}, \tilde{\mathbf{x}}_t^\infty) = R(\mathbf{x}) - R(\tilde{\mathbf{x}}_t^\infty) + \langle \nabla_{1:t}, \mathbf{x} - \tilde{\mathbf{x}}_t^\infty\rangle.$$

From the stability of predictions of OFTPL we know that: $\|\nabla\Phi(g_1)-\nabla\Phi(g_2)\|\leqslant C\eta^{-1}\|g_1-g_2\|_*$. Following our connection between $\Psi, \Phi$, this implies $\|\nabla\Psi(g_1)-\nabla\Psi(g_2)\|\leqslant C\eta^{-1}\|g_1-g_2\|_*$. This implies the following smoothness condition on $\Psi$ [see Lemma 15 of 24]

$$\Psi(g_2)\leqslant\Psi(g_1)+\langle\nabla\Psi(g_1),g_2-g_1\rangle+\frac{C\eta^{-1}}{2}\|g_1-g_2\|_*^2.$$

Since $\Psi$ is $C\eta^{-1}$-smooth w.r.t $\|\cdot\|_*$, following duality between strong convexity and strong smoothness properties (see Theorem H.5), we can infer that $R$ is $C^{-1}\eta$- strongly convex w.r.t $\|\cdot\|$ norm and satisfies

$$B(\mathbf{x},\mathbf{x}_t^\infty)\geqslant\frac{\eta}{2C}\|\mathbf{x}-\mathbf{x}_t^\infty\|^2,\quad B(\mathbf{x},\tilde{\mathbf{x}}_t^\infty)\geqslant\frac{\eta}{2C}\|\mathbf{x}-\tilde{\mathbf{x}}_t^\infty\|^2.$$

We now go ahead and bound the regret of the learner. For any $\mathbf{x}\in\mathcal{X}$, we have

$$
\begin{aligned}
f_t(\mathbf{x}_t)-f_t(\mathbf{x})&\overset{(a)}{\leqslant}\langle\mathbf{x}_t-\mathbf{x},\nabla_t\rangle=\langle\mathbf{x}_t-\mathbf{x}_t^\infty,\nabla_t\rangle+\langle\mathbf{x}_t^\infty-\mathbf{x},\nabla_t\rangle\\
&=\langle\mathbf{x}_t-\mathbf{x}_t^\infty,\nabla_t\rangle+\langle\mathbf{x}_t^\infty-\tilde{\mathbf{x}}_t^\infty,\nabla_t-g_t\rangle+\langle\mathbf{x}_t^\infty-\tilde{\mathbf{x}}_t^\infty,g_t\rangle\\
&\quad+\langle\tilde{\mathbf{x}}_t^\infty-\mathbf{x},\nabla_t\rangle\\
&\leqslant\langle\mathbf{x}_t-\mathbf{x}_t^\infty,\nabla_t\rangle+\|\mathbf{x}_t^\infty-\tilde{\mathbf{x}}_t^\infty\|\|\nabla_t-g_t\|_*+\langle\mathbf{x}_t^\infty-\tilde{\mathbf{x}}_t^\infty,g_t\rangle\\
&\quad+\langle\tilde{\mathbf{x}}_t^\infty-\mathbf{x},\nabla_t\rangle,
\end{aligned}
$$

where $(a)$ follows from convexity of $f$. Next, a simple calculation shows that

$$
\begin{aligned}
\langle\mathbf{x}_t^\infty-\tilde{\mathbf{x}}_t^\infty,g_t\rangle&=B(\tilde{\mathbf{x}}_t^\infty,\tilde{\mathbf{x}}_{t-1}^\infty)-B(\tilde{\mathbf{x}}_t^\infty,\mathbf{x}_t^\infty)-B(\mathbf{x}_t^\infty,\tilde{\mathbf{x}}_{t-1}^\infty)\\
\langle\tilde{\mathbf{x}}_t^\infty-\mathbf{x},\nabla_t\rangle&=B(\mathbf{x},\tilde{\mathbf{x}}_{t-1}^\infty)-B(\mathbf{x},\tilde{\mathbf{x}}_t^\infty)-B(\tilde{\mathbf{x}}_t^\infty,\tilde{\mathbf{x}}_{t-1}^\infty).
\end{aligned}
$$

Substituting this in the previous inequality gives us

$$
\begin{aligned}
f_t(\mathbf{x}_t)-f_t(\mathbf{x})&\leqslant\langle\mathbf{x}_t-\mathbf{x}_t^\infty,\nabla_t\rangle+\|\mathbf{x}_t^\infty-\tilde{\mathbf{x}}_t^\infty\|\|\nabla_t-g_t\|_*\\
&\quad+B(\tilde{\mathbf{x}}_t^\infty,\tilde{\mathbf{x}}_{t-1}^\infty)-B(\tilde{\mathbf{x}}_t^\infty,\mathbf{x}_t^\infty)-B(\mathbf{x}_t^\infty,\tilde{\mathbf{x}}_{t-1}^\infty)\\
&\quad+B(\mathbf{x},\tilde{\mathbf{x}}_{t-1}^\infty)-B(\mathbf{x},\tilde{\mathbf{x}}_t^\infty)-B(\tilde{\mathbf{x}}_t^\infty,\tilde{\mathbf{x}}_{t-1}^\infty)\\
&=\langle\mathbf{x}_t-\mathbf{x}_t^\infty,\nabla_t\rangle+\|\mathbf{x}_t^\infty-\tilde{\mathbf{x}}_t^\infty\|\|\nabla_t-g_t\|_*\\
&\quad+B(\mathbf{x},\tilde{\mathbf{x}}_{t-1}^\infty)-B(\mathbf{x},\tilde{\mathbf{x}}_t^\infty)-B(\tilde{\mathbf{x}}_t^\infty,\mathbf{x}_t^\infty)-B(\mathbf{x}_t^\infty,\tilde{\mathbf{x}}_{t-1}^\infty)\\
&\overset{(a)}{\leqslant}\langle\mathbf{x}_t-\mathbf{x}_t^\infty,\nabla_t\rangle+\|\mathbf{x}_t^\infty-\tilde{\mathbf{x}}_t^\infty\|\|\nabla_t-g_t\|_*\\
&\quad+B(\mathbf{x},\tilde{\mathbf{x}}_{t-1}^\infty)-B(\mathbf{x},\tilde{\mathbf{x}}_t^\infty)-\frac{\eta\|\tilde{\mathbf{x}}_t^\infty-\mathbf{x}_t^\infty\|^2}{2C}-\frac{\eta\|\mathbf{x}_t^\infty-\tilde{\mathbf{x}}_{t-1}^\infty\|^2}{2C},
\end{aligned}
$$

where $(a)$ follows from strongly convexity of $R$. Summing over $t=1,\dots T$, gives us

$$
\begin{aligned}
\sum_{t=1}^T f_t(\mathbf{x}_t)-f_t(\mathbf{x})&\leqslant\sum_{t=1}^T\langle\mathbf{x}_t-\mathbf{x}_t^\infty,\nabla_t\rangle+\underbrace{B(\mathbf{x},\tilde{\mathbf{x}}_0^\infty)-B(\mathbf{x},\tilde{\mathbf{x}}_T^\infty)}_{S_1}\\
&\quad+\sum_{t=1}^T\|\mathbf{x}_t^\infty-\tilde{\mathbf{x}}_t^\infty\|\|\nabla_t-g_t\|_*\\
&\quad-\frac{\eta}{2C}\sum_{t=1}^T\left(\|\tilde{\mathbf{x}}_t^\infty-\mathbf{x}_t^\infty\|^2+\|\mathbf{x}_t^\infty-\tilde{\mathbf{x}}_{t-1}^\infty\|^2\right).
\end{aligned}
$$

**Bounding $S_1$.** We now bound $B(\mathbf{x},\tilde{\mathbf{x}}_0^\infty)-B(\mathbf{x},\tilde{\mathbf{x}}_T^\infty)$. From the definition of $B$, we have

$$B(\mathbf{x},\tilde{\mathbf{x}}_0^\infty)-B(\mathbf{x},\tilde{\mathbf{x}}_T^\infty)=R(\tilde{\mathbf{x}}_T^\infty)-\langle\nabla_{1:T},\mathbf{x}-\tilde{\mathbf{x}}_T^\infty\rangle-R(\tilde{\mathbf{x}}_0^\infty)+\langle\nabla_{1:0},\mathbf{x}-\tilde{\mathbf{x}}_T^\infty\rangle.$$

Note that $\nabla_{1:0}=0$. This gives us

$$B(\mathbf{x},\tilde{\mathbf{x}}_0^\infty)-B(\mathbf{x},\tilde{\mathbf{x}}_T^\infty)=R(\tilde{\mathbf{x}}_T^\infty)-\langle\nabla_{1:T},\mathbf{x}-\tilde{\mathbf{x}}_T^\infty\rangle-R(\tilde{\mathbf{x}}_0^\infty).$$

We now use duality to convert the RHS of the above equation, which is currently in terms of $R$, into a quantity which depends on $\Phi$. From Proposition 3.1 we have

$$\Phi(g)=-\Psi(-g)=\inf_{\mathbf{x}\in\mathcal{X}}\langle g,\mathbf{x}\rangle+R(\mathbf{x}).$$

Since $\tilde{\mathbf{x}}_T^\infty$ is the minimizer of $\langle\nabla_{1:T},\mathbf{x}\rangle+R(\mathbf{x})$, we have $\Phi(\nabla_{1:T})=\langle\nabla_{1:T},\tilde{\mathbf{x}}_T^\infty\rangle+R(\tilde{\mathbf{x}}_T^\infty)$. Similarly, $\Phi(0)=R(\tilde{\mathbf{x}}_0^\infty)$. Substituting these in the previous equation gives us

$$
\begin{aligned}
B(\mathbf{x},\tilde{\mathbf{x}}_0^\infty)-B(\mathbf{x},\tilde{\mathbf{x}}_T^\infty) &= \Phi(\nabla_{1:T})-\langle\nabla_{1:T},\mathbf{x}\rangle-\Phi(0) \\
&= \mathbb{E}_\sigma\left[\inf_{\mathbf{x}'\in\mathcal{X}}\left\langle\nabla_{1:T}-\sigma,\mathbf{x}'\right\rangle\right]-\langle\nabla_{1:T},\mathbf{x}\rangle-\mathbb{E}_\sigma\left[\inf_{\mathbf{x}'\in\mathcal{X}}\left\langle-\sigma,\mathbf{x}'\right\rangle\right] \\
&\leqslant \mathbb{E}_\sigma\left[\langle\nabla_{1:T}-\sigma,\mathbf{x}\rangle\right]-\langle\nabla_{1:T},\mathbf{x}\rangle-\mathbb{E}_\sigma\left[\inf_{\mathbf{x}'\in\mathcal{X}}\left\langle-\sigma,\mathbf{x}'\right\rangle\right] \\
&= \mathbb{E}_\sigma\left[\inf_{\mathbf{x}'\in\mathcal{X}}\left\langle\sigma,\mathbf{x}'\right\rangle\right]-\mathbb{E}_\sigma\left[\langle\sigma,\mathbf{x}\rangle\right] \\
&\leqslant D\mathbb{E}_\sigma\left[\|\sigma\|_*\right]=\eta D
\end{aligned}
$$

**Bounding Regret.** Substituting this in our regret bound and taking expectation on both sides gives us

$$
\begin{aligned}
\mathbb{E}\left[\sum_{t=1}^T f_t(\mathbf{x}_t)-f_t(\mathbf{x})\right] &\leqslant \sum_{t=1}^T\mathbb{E}\left[\langle\mathbf{x}_t-\mathbf{x}_t^\infty,\nabla_t\rangle\right]+\eta D+\sum_{t=1}^T\mathbb{E}\left[\|\mathbf{x}_t^\infty-\tilde{\mathbf{x}}_t^\infty\|\|\nabla_t-g_t\|_*\right] \\
&\quad -\frac{\eta}{2C}\sum_{t=1}^T\left(\mathbb{E}\left[\|\tilde{\mathbf{x}}_{t-1}^\infty-\mathbf{x}_t^\infty\|^2\right]+\mathbb{E}\left[\|\mathbf{x}_t^\infty-\tilde{\mathbf{x}}_{t-1}^\infty\|^2\right]\right) \\
&\leqslant \sum_{t=1}^T\mathbb{E}\left[\langle\mathbf{x}_t-\mathbf{x}_t^\infty,\nabla_t\rangle\right]+\eta D+\sum_{t=1}^T\frac{C}{2\eta}\mathbb{E}\left[\|\nabla_t-g_t\|_*^2\right] \\
&\quad -\frac{\eta}{2C}\sum_{t=1}^T\mathbb{E}\left[\|\mathbf{x}_t^\infty-\tilde{\mathbf{x}}_{t-1}^\infty\|^2\right]
\end{aligned}
$$

To finish the proof, we make use of the Holder's smoothness assumption on $f_t$ to bound the first term in the RHS above. From Holder's smoothness assumption, we have

$$
\langle\mathbf{x}_t-\mathbf{x}_t^\infty,\nabla_t-\nabla f_t(\mathbf{x}_t^\infty)\rangle\leqslant L\|\mathbf{x}_t-\mathbf{x}_t^\infty\|^{1+\alpha}.
$$

Using this, we get

$$
\begin{aligned}
\mathbb{E}\left[\langle\mathbf{x}_t-\mathbf{x}_t^\infty,\nabla_t\rangle|g_t,\mathbf{x}_{1:t-1},f_{1:t}\right] &\leqslant \mathbb{E}\left[\langle\mathbf{x}_t-\mathbf{x}_t^\infty,\nabla f_t(\mathbf{x}_t^\infty)\rangle+L\|\mathbf{x}_t-\mathbf{x}_t^\infty\|^{1+\alpha}|g_t,\mathbf{x}_{1:t-1},f_{1:t}\right] \\
&\overset{(a)}{=} L\mathbb{E}\left[\|\mathbf{x}_t-\mathbf{x}_t^\infty\|^{1+\alpha}|g_t,\mathbf{x}_{1:t-1},f_{1:t}\right] \\
&\overset{(b)}{\leqslant} \Psi_1^{1+\alpha}L\mathbb{E}\left[\|\mathbf{x}_t-\mathbf{x}_t^\infty\|_2^{1+\alpha}|g_t,\mathbf{x}_{1:t-1},f_{1:t}\right] \\
&\overset{(c)}{\leqslant} \Psi_1^{1+\alpha}L\mathbb{E}\left[\|\mathbf{x}_t-\mathbf{x}_t^\infty\|_2^2|g_t,\mathbf{x}_{1:t-1},f_{1:t}\right]^{(1+\alpha)/2} \\
&\overset{(d)}{\leqslant} L\left(\frac{\Psi_1\Psi_2 D}{\sqrt{m}}\right)^{1+\alpha},
\end{aligned}
$$

where $(a)$ follows from the fact that $\mathbb{E}\left[\langle\mathbf{x}_t-\mathbf{x}_t^\infty,\nabla f_t(\mathbf{x}_t^\infty)\rangle|g_t,\mathbf{x}_{1:t-1},f_{1:t}\right]=0$, $(b)$ follows from the definition of norm compatibility constant $\Psi_1$, $(c)$ follows from Holders inequality and $(d)$ uses the fact that conditioned on $\{g_t,\mathbf{x}_{1:t-1},f_{1:t}\}$, $\mathbf{x}_t-\mathbf{x}_t^\infty$ is the average of $m$ i.i.d bounded mean 0 random variables, the variance of which scales as $O(D^2/m)$. Substituting this in the above regret bound gives us the required result.

## B.2 Proof of Corollary 4.1

We first bound $\mathbb{E}_\sigma\left[\|\sigma\|_2\right]$. Relying on spherical symmetry of the perturbation distribution and the fact that the density of $P_{\text{PRTB}}$ on the spherical shell of radius $r$ is proportional to $r^{d-1}$, we get

$$
\mathbb{E}_\sigma\left[\|\sigma\|_2\right]=\frac{\int_{r=0}^{(1+d^{-1})\eta}r\times r^{d-1}dr}{\int_{r=0}^{(1+d^{-1})\eta}r^{d-1}dr}=\eta.
$$

We now bound the stability of predictions of OFTPL. Our technique for bounding the stability uses similar arguments as Hazan and Minasyan [10] (see Lemma 4.2 of [10]). Recall, to bound stability, we need to show that $\Phi(g) = \mathbb{E}_\sigma \left[\inf_{\mathbf{x} \in \mathcal{X}} \langle g - \sigma, \mathbf{x} \rangle\right]$ is smooth. Let $\phi_0(g) = \inf_{\mathbf{x} \in \mathcal{X}} \langle g, \mathbf{x} - \mathbf{x}_{00} \rangle$, where $\mathbf{x}_{00}$ is an arbitrary point in $\mathcal{X}$. We can rewrite $\Phi(g)$ as

$$\Phi(g) = \mathbb{E}_\sigma \left[\phi_0(g - \sigma)\right] + \langle g, \mathbf{x}_{00} \rangle.$$

Since the second term in the RHS above is linear in $g$, any upper bound on the smoothness of $\mathbb{E}_\sigma \left[\phi_0(g - \sigma)\right]$ is also a bound on the smoothness of $\Phi(g)$. So we focus on bounding the smoothness of $\mathbb{E}_\sigma \left[\phi_0(g - \sigma)\right]$.

First note that $\phi_0(g)$ is $D$ Lipschitz and satisfies the following for any $g_1, g_2 \in \mathbb{R}^d$

$$\begin{aligned}
\phi_0(g_1) - \phi_0(g_2) &= \inf_{\mathbf{x} \in \mathcal{X}} \langle -g_2, \mathbf{x} - \mathbf{x}_{00} \rangle - \inf_{\mathbf{x} \in \mathcal{X}} \langle -g_1, \mathbf{x} - \mathbf{x}_{00} \rangle \\
&\leqslant \sup_{\mathbf{x} \in \mathcal{X}} \langle g_1 - g_2, \mathbf{x} - \mathbf{x}_{00} \rangle \\
&\leqslant D \|g_1 - g_2\|_2.
\end{aligned}$$

Letting $\Phi_0(g) = \mathbb{E}_\sigma \left[\phi_0(g - \sigma)\right]$, Lemma 4.2 of Hazan and Minasyan [10] shows that $\Phi_0(g)$ is smooth and satisfies

$$\|\nabla \Phi_0(g_1) - \nabla \Phi_0(g_2)\|_2 \leqslant dD\eta^{-1} \|g_1 - g_2\|_2.$$

This shows that the predictions of OFTPL are $dD\eta^{-1}$ stable. The rest of the proof involves substituting $C = dD$ in the regret bound of Theorem 4.1 and setting $g_t = 0$ and using the fact that $\|\nabla_t\|_2 \leqslant G$.

# C  Online Nonconvex Learning

## C.1  Proof of Theorem 4.2

Before we present the proof of the Theorem, we introduce some notation and present some useful intermediate results. We note that unlike the convex case, there are no know Fenchel duality theorems for infinite dimensional setting. So more careful arguments are need to obtain tight regret bounds. Our proof mimics the proof of Theorem 4.1.

### C.1.1  Notation

Let $\mathcal{P}$ be the set of all probability measures on $\mathcal{X}$. We define functions $\Phi : \mathcal{F} \to \mathbb{R}$, $R : \mathcal{P} \to \mathbb{R}$ as follows

$$\begin{aligned}
\Phi(f) &= \mathbb{E}_\sigma \left[\inf_{P \in \mathcal{P}} \mathbb{E}_{\mathbf{x} \sim P} \left[f(\mathbf{x}) - \sigma(\mathbf{x})\right]\right], \\
R(P) &= \sup_{f \in \mathcal{F}} -\mathbb{E}_{\mathbf{x} \sim P} \left[f(\mathbf{x})\right] + \Phi(f).
\end{aligned}$$

Also, note that the function $\nabla \Phi : \mathcal{F} \to \mathcal{P}$ defined in Section 4.2 can be written as

$$\nabla \Phi(f) = \mathbb{E}_\sigma \left[\operatorname*{argmin}_{P \in \mathcal{P}} \mathbb{E}_{\mathbf{x} \sim P} \left[f(\mathbf{x}) - \sigma(\mathbf{x})\right]\right].$$

Note that, $\nabla \Phi(f)$ is well defined because from our assumption on the perturbation distribution, the minimization problem inside the expectation has a unique minimizer with probability one. To simplify the notation, in the sequel, we use the shorthand notation $\langle P, f \rangle$ to denote $\mathbb{E}_{\mathbf{x} \sim P} \left[f(\mathbf{x})\right]$, for any $P \in \mathcal{P}$ and $f \in \mathcal{F}$. Similarly, for any $P_1, P_2 \in \mathcal{P}$ and $f \in \mathcal{F}$, we use the notation $\langle P_1 - P_2, f \rangle$ to denote $\mathbb{E}_{\mathbf{x} \sim P_1} \left[f(\mathbf{x})\right] - \mathbb{E}_{\mathbf{x} \sim P_2} \left[f(\mathbf{x})\right]$.

### C.1.2  Intermediate Results

**Lemma C.1.** *For any $g \in \mathcal{F}$, $R(\nabla \Phi(g)) = -\langle \nabla \Phi(g), g \rangle + \Phi(g)$.*

*Proof.* Define $P_{g,\sigma}$ as

$$P_{g,\sigma} = \operatorname*{argmin}_{P \in \mathcal{P}} \mathbb{E}_{\mathbf{x} \sim P} \left[g(\mathbf{x}) - \sigma(\mathbf{x})\right].$$

Note that $\nabla\Phi(g) = \mathbb{E}_\sigma[P_{g,\sigma}]$. For any $g, h \in \mathcal{F}$, we have

$$\Phi(h) = \mathbb{E}_\sigma\left[\inf_{P\in\mathcal{P}} \langle P, h - \sigma\rangle\right]$$
$$\leqslant \mathbb{E}_\sigma\left[\langle P_{g,\sigma}, h - \sigma\rangle\right]$$
$$= \mathbb{E}_\sigma\left[\langle P_{g,\sigma}, g - \sigma\rangle\right] + \mathbb{E}_\sigma\left[\langle P_{g,\sigma}, h - g\rangle\right]$$
$$= \Phi(g) + \langle\nabla\Phi(g), h - g\rangle.$$

This shows that for any $g, h \in \mathcal{F}$

$$\Phi(h) - \langle\nabla\Phi(g), h\rangle \leqslant \Phi(g) - \langle\nabla\Phi(g), g\rangle. \tag{5}$$

Taking supremum over $h$ of the LHS quantity gives us

$$R(\nabla\Phi(g)) = \sup_{h\in\mathcal{F}} \Phi(h) - \langle\nabla\Phi(g), h\rangle = \Phi(g) - \langle\nabla\Phi(g), g\rangle.$$

$\square$

**Lemma C.2** (Strong Smoothness). *The function $-\Phi$ is convex and strongly smooth and satisfies the following inequality for any $g_1, g_2 \in \mathcal{F}$*

$$-\Phi(g_2) \leqslant -\Phi(g_1) - \langle\nabla\Phi(g_1), g_2 - g_1\rangle + \frac{C}{2\eta}\|g_2 - g_1\|_\mathcal{F}^2.$$

*Proof.* Let $g_1, g_2 \in \mathcal{F}$ and $\alpha \in [0, 1]$. Then

$$\Phi(\alpha g_1 + (1-\alpha)g_2) = \mathbb{E}_\sigma\left[\inf_{P\in\mathcal{P}} \langle P, \alpha g_1 + (1-\alpha)g_2 - \sigma\rangle\right]$$
$$\geqslant \alpha\mathbb{E}_\sigma\left[\inf_{P\in\mathcal{P}} \langle P, g_1 - \sigma\rangle\right] + (1-\alpha)\mathbb{E}_\sigma\left[\inf_{P\in\mathcal{P}} \langle P, g_2 - \sigma\rangle\right]$$
$$= \alpha\Phi(g_1) + (1-\alpha)\Phi(g_2).$$

This shows that $-\Phi$ is convex. To show smoothness, we rely on the following stability property

$$\forall g_1, g_2 \in \mathcal{F} \quad \gamma_\mathcal{F}(\nabla\Phi(g_1), \nabla\Phi(g_2)) \leqslant \frac{C}{\eta}\|g_1 - g_2\|_\mathcal{F}.$$

Let $T$ be an arbitrary positive integer and for $t \in \{0, 1, \ldots T\}$, define $\alpha_t = t/T$. Let $h = g_2 - g_1$. We have

$$\Phi(g_1) - \Phi(g_2) = \Phi(g_1 + \alpha_0 h) - \Phi(g_1 + \alpha_T h)$$
$$= \sum_{t=0}^{T-1} (\Phi(g_1 + \alpha_t h) - \Phi(g_1 + \alpha_{t+1}h))$$

Since $-\Phi$ is convex and satisfies Equation (5), we have

$$\Phi(g_1) - \Phi(g_2) = \sum_{t=0}^{T-1} (\Phi(g_1 + \alpha_t h) - \Phi(g_1 + \alpha_{t+1}h))$$
$$\leqslant -\sum_{t=0}^{T-1} \frac{1}{T} \langle\nabla\Phi(g_1 + \alpha_{t+1}h), h\rangle$$

Using stability, we get

$$\Phi(g_1) - \Phi(g_2) \leqslant - \sum_{t=0}^{T-1} \frac{1}{T} \left\langle \nabla \Phi (g_1 + \alpha_{t+1} h), h \right\rangle$$

$$= \sum_{t=0}^{T-1} \frac{1}{T} \left( \left\langle \nabla \Phi (g_1) - \nabla \Phi (g_1 + \alpha_{t+1} h), h \right\rangle - \left\langle \nabla \Phi (g_1), h \right\rangle \right)$$

$$\overset{(a)}{\leqslant} - \left\langle \nabla \Phi (g_1), h \right\rangle + \sum_{t=0}^{T-1} \frac{1}{T} \gamma_{\mathcal{F}}(\nabla \Phi (g_1), \nabla \Phi (g_1 + \alpha_{t+1} h)) \|h\|_{\mathcal{F}}$$

$$\overset{(b)}{\leqslant} - \left\langle \nabla \Phi (g_1), h \right\rangle + \sum_{t=0}^{T-1} \frac{C}{T\eta} \|\alpha_{t+1} h\|_{\mathcal{F}} \|h\|_{\mathcal{F}}$$

$$= - \left\langle \nabla \Phi (g_1), h \right\rangle + \sum_{t=0}^{T-1} \frac{C \alpha_{t+1}}{T\eta} \|h\|_{\mathcal{F}}^2$$

$$= - \left\langle \nabla \Phi (g_1), h \right\rangle + \frac{C}{\eta} \frac{T+1}{2T} \|h\|_{\mathcal{F}}^2,$$

where $(a)$ follows from the definition of $\gamma_{\mathcal{F}}$ and $(b)$ follows from the stability assumption. Taking $T \to \infty$, we get

$$-\Phi(g_2) \leqslant -\Phi(g_1) - \left\langle \nabla \Phi (g_1), g_2 - g_1 \right\rangle + \frac{C}{2\eta} \|g_2 - g_1\|_{\mathcal{F}}^2.$$

$\square$

**Lemma C.3** (Strong Convexity). *For any $P \in \mathcal{P}$ and $g \in \mathcal{F}$, $R$ satisfies the following inequality*

$$R(P) \geqslant R(\nabla \Phi (g)) + \left\langle \nabla \Phi (g) - P, g \right\rangle + \frac{\eta}{2C} \gamma_{\mathcal{F}}(P, \nabla \Phi (g))^2.$$

*Proof.* From Lemma C.2 we know that the following holds for any $g, h \in \mathcal{F}$

$$\Phi(g) \geqslant \underbrace{\Phi(h) + \left\langle \nabla \Phi (h), g - h \right\rangle - \frac{C}{2\eta} \|g - h\|_{\mathcal{F}}^2}_{\Phi_{\mathrm{lb}, h}(g)}.$$

Define $R_{\mathrm{lb}, h}(P)$ as

$$R_{\mathrm{lb}}(P) = \sup_{g \in \mathcal{F}} - \left\langle P, g \right\rangle + \Phi_{\mathrm{lb}, h}(g).$$

Since $\Phi(g) \geqslant \Phi_{\mathrm{lb}, h}(g)$ for all $g \in \mathcal{F}$, $R(P) \geqslant R_{\mathrm{lb}, h}(P)$ for all $P$. We now derive an expression for $R_{\mathrm{lb}, h}(P)$. Note that from Lemma C.1 we have $R(\nabla \Phi (h)) = - \left\langle \nabla \Phi (h), h \right\rangle + \Phi(h)$. Using this, we get

$$R_{\mathrm{lb}, h}(P) = \sup_{g \in \mathcal{F}} - \left\langle P, g \right\rangle + \Phi_{\mathrm{lb}, h}(g)$$

$$\overset{(a)}{=} \sup_{g \in \mathcal{F}} \left( - \left\langle P, g \right\rangle + \Phi(h) + \left\langle \nabla \Phi (h), g - h \right\rangle - \frac{C}{2\eta} \|g - h\|_{\mathcal{F}}^2 \right)$$

$$\overset{(b)}{=} R(\nabla \Phi (h)) + \sup_{g \in \mathcal{F}} \left( \left\langle \nabla \Phi (h) - P, g \right\rangle - \frac{C}{2\eta} \|g - h\|_{\mathcal{F}}^2 \right),$$

where $(a)$ follows from the definition of $\Phi_{\mathrm{lb}, h}(g)$ and $(b)$ follows from Lemma C.1. We now do a change of variables in the supremum of the above expression. Substituting $g' = g - h$, we get

$$R_{\mathrm{lb}, h}(P) = R(\nabla \Phi (h)) + \left\langle \nabla \Phi (h) - P, h \right\rangle + \sup_{g' \in \mathcal{F}} \left( \left\langle \nabla \Phi (h) - P, g' \right\rangle - \frac{C}{2\eta} \|g'\|_{\mathcal{F}}^2 \right).$$

We now show that

$$\sup_{g' \in \mathcal{F}} \left( \left\langle \nabla \Phi (h) - P, g' \right\rangle - \frac{C}{2\eta} \|g'\|_{\mathcal{F}}^2 \right) \geqslant \frac{\eta}{2C} \gamma_{\mathcal{F}}(P, \nabla \Phi (h))^2.$$

To this end, we choose a $g'' \in \mathcal{F}$ such that

$$\|g''\|_{\mathcal{F}} = \frac{\eta}{C}\gamma_{\mathcal{F}}(P, \nabla\Phi(h)), \quad \langle\nabla\Phi(h) - P, g''\rangle = \frac{\eta}{C}\gamma_{\mathcal{F}}(P, \nabla\Phi(h))^2. \tag{6}$$

If such a $g''$ can be found, we have

$$\sup_{g'\in\mathcal{F}}\left(\langle\nabla\Phi(h) - P, g'\rangle - \frac{C}{2\eta}\|g'\|_{\mathcal{F}}^2\right) \geqslant \langle\nabla\Phi(h) - P, g''\rangle - \frac{C}{2\eta}\|g''\|_{\mathcal{F}}^2$$

$$= \frac{\eta}{2C}\gamma_{\mathcal{F}}(P, \nabla\Phi(h))^2.$$

This would then imply the main claim of the Lemma.

$$R(P) \geqslant R_{\mathrm{lb},h}(P) \geqslant R(\nabla\Phi(h)) + \langle\nabla\Phi(h) - P, h\rangle + \frac{\eta}{2C}\gamma_{\mathcal{F}}(P, \nabla\Phi(h))^2.$$

**Finding $g''$.** We now construct a $g''$ which satisfies Equation (6). From the definition of $\gamma_{\mathcal{F}}$ we know that

$$\gamma_{\mathcal{F}}(P, \nabla\Phi(h)) = \sup_{\|g'\|_{\mathcal{F}}\leqslant 1}|\langle\nabla\Phi(h) - P, g'\rangle|$$

Suppose the supremum is achieved at $g^*$. Define $g''$ as $\frac{\eta s}{C}\gamma_{\mathcal{F}}(P, \nabla\Phi(h))g^*$, where $s = \mathrm{sign}(\langle\nabla\Phi(h) - P, g^*\rangle)$. It can be easily verified that $g''$ satifies Equation (6).

If the supremum is never achieved, the same argument as above can still be made using a sequence of functions $\{g_n\}_{n=1}^{\infty}$ such that

$$\|g_n\|_{\mathcal{F}} \leqslant 1, \quad \lim_{n\to\infty}|\langle\nabla\Phi(h) - P, g_n\rangle| = \gamma_{\mathcal{F}}(P, \nabla\Phi(h)).$$

Define $g_n''$ as $\frac{\eta s_n}{C}\gamma_{\mathcal{F}}(P, \nabla\Phi(h))g_n$, where $s_n = \mathrm{sign}(\langle\nabla\Phi(h) - P, g_n\rangle)$. Since $\lim_{n\to\infty}\|g_n\|_{\mathcal{F}} = 1$, we have $\lim_{n\to\infty}\|g_n''\|_{\mathcal{F}} = \frac{\eta}{C}\gamma_{\mathcal{F}}(P, \nabla\Phi(h))$. Moreover,

$$\lim_{n\to\infty}\langle\nabla\Phi(h) - P, g_n''\rangle = \lim_{n\to\infty}\frac{\eta}{C}\gamma_{\mathcal{F}}(P, \nabla\Phi(h))\left|\langle\nabla\Phi(h) - P, g_n\rangle\right| = \frac{\eta}{C}\gamma_{\mathcal{F}}(P, \nabla\Phi(h))^2.$$

This shows that

$$\sup_{g'\in\mathcal{F}}\left(\langle\nabla\Phi(h) - P, g'\rangle - \frac{C}{2\eta}\|g'\|_{\mathcal{F}}^2\right) \geqslant \lim_{n\to\infty}\langle\nabla\Phi(h) - P, g_n''\rangle - \frac{C}{2\eta}\|g_n''\|_{\mathcal{F}}^2$$

$$= \frac{\eta}{2C}\gamma_{\mathcal{F}}(P, \nabla\Phi(h))^2.$$

This finishes the proof of the Lemma. □

### C.1.3 Main Argument

We are now ready to prove Theorem 4.2. Our proof relies on Lemma C.3 and uses similar arguments as used in the proof of Theorem 4.1. We first rewrite $P_t, \tilde{P}_t$ as

$$P_t = \frac{1}{m}\sum_{j=1}^{m}\operatorname*{argmin}_{P\in\mathcal{P}}\mathbb{E}_{\mathbf{x}\sim P}\left[\sum_{i=1}^{t-1}f_i(\mathbf{x}) + g_t(\mathbf{x}) - \sigma_{t,j}(\mathbf{x})\right],$$

$$\tilde{P}_t = \frac{1}{m}\sum_{j=1}^{m}\operatorname*{argmin}_{P\in\mathcal{P}}\mathbb{E}_{\mathbf{x}\sim P}\left[\sum_{i=1}^{t}f_i(\mathbf{x}) - \sigma_{t,j}'(\mathbf{x})\right].$$

Note that

$$P_t^{\infty} = \mathbb{E}\left[P_t|g_t, f_{1:t-1}, P_{1:t-1}\right] = \nabla\Phi(f_{1:t-1} + g_t),$$

$$\tilde{P}_t^{\infty} = \mathbb{E}\left[\tilde{P}_t|f_{1:t-1}, P_{1:t-1}\right] = \nabla\Phi(f_{1:t}),$$

with $P_1^{\infty} = \tilde{P}_0^{\infty} = \nabla\Phi(0)$. Define functions $B(\cdot, P_t^{\infty}), B(\cdot, \tilde{P}_t^{\infty})$ as

$$B(P, P_t^{\infty}) = R(P) - R(P_t^{\infty}) + \langle P - P_t^{\infty}, f_{1:t-1} + g_t\rangle,$$

$$B(P, \tilde{P}_t^{\infty}) = R(P) - R(\tilde{P}_t^{\infty}) + \langle P - \tilde{P}_t^{\infty}, f_{1:t}\rangle.$$

From Lemma C.3, we have

$$B(P, P_t^\infty) \geqslant \frac{\eta}{2C} \gamma_{\mathcal{F}}(P, P_t^\infty)^2, \quad B(P, \tilde{P}_t^\infty) \geqslant \frac{\eta}{2C} \gamma_{\mathcal{F}}(P, \tilde{P}_t^\infty)^2.$$

For any $P \in \mathcal{P}$, we have

$$
\begin{aligned}
\mathbb{E}\left[f_t(\mathbf{x}_t) - f_t(P)\right] &= \mathbb{E}\left[f_t(P_t) - f_t(P)\right] \\
&= \mathbb{E}\left[\langle P_t - P, f_t \rangle\right] \\
&= \mathbb{E}\left[\langle P_t - P_t^\infty, f_t \rangle\right] + \mathbb{E}\left[\langle P_t^\infty - P, f_t \rangle\right] \\
&= \mathbb{E}\left[\langle P_t - P_t^\infty, f_t \rangle\right] + \mathbb{E}\left[\left\langle P_t^\infty - \tilde{P}_t^\infty, f_t - g_t \right\rangle\right] \\
&\quad + \mathbb{E}\left[\left\langle P_t^\infty - \tilde{P}_t^\infty, g_t \right\rangle\right] + \mathbb{E}\left[\left\langle \tilde{P}_t^\infty - P, f_t \right\rangle\right] \\
&\stackrel{(a)}{\leqslant} \mathbb{E}\left[\gamma_{\mathcal{F}}(P_t^\infty, \tilde{P}_t^\infty)\|f_t - g_t\|_{\mathcal{F}}\right] + \mathbb{E}\left[\left\langle P_t^\infty - \tilde{P}_t^\infty, g_t \right\rangle\right] \\
&\quad + \mathbb{E}\left[\left\langle \tilde{P}_t^\infty - P, f_t \right\rangle\right],
\end{aligned}
$$

where $(a)$ follows from the fact that $\mathbb{E}\left[\langle P_t - P_t^\infty, f_t \rangle | g_t, f_{1:t-1}, P_{1:t-1}\right] = 0$ and as a result $\mathbb{E}\left[\langle P_t - P_t^\infty, f_t \rangle\right] = 0$. Next, a simple calculation shows that

$$\left\langle P_t^\infty - \tilde{P}_t^\infty, g_t \right\rangle = B(\tilde{P}_t^\infty, \tilde{P}_{t-1}^\infty) - B(\tilde{P}_t^\infty, P_t^\infty) - B(P_t^\infty, \tilde{P}_{t-1}^\infty)$$

$$\left\langle \tilde{P}_t^\infty - P, f_t \right\rangle = B(P, \tilde{P}_{t-1}^\infty) - B(P, \tilde{P}_t^\infty) - B(\tilde{P}_t^\infty, \tilde{P}_{t-1}^\infty).$$

Substituting this in the previous regret bound gives us

$$
\begin{aligned}
\mathbb{E}\left[f_t(\mathbf{x}_t) - f_t(P)\right] &\leqslant \mathbb{E}\left[\gamma_{\mathcal{F}}(P_t^\infty, \tilde{P}_t^\infty)\|f_t - g_t\|_{\mathcal{F}}\right] + \mathbb{E}\left[B(\tilde{P}_t^\infty, \tilde{P}_{t-1}^\infty) - B(\tilde{P}_t^\infty, P_t^\infty) - B(P_t^\infty, \tilde{P}_{t-1}^\infty)\right] \\
&\quad + \mathbb{E}\left[B(P, \tilde{P}_{t-1}^\infty) - B(P, \tilde{P}_t^\infty) - B(\tilde{P}_t^\infty, \tilde{P}_{t-1}^\infty)\right] \\
&= \mathbb{E}\left[\gamma_{\mathcal{F}}(P_t^\infty, \tilde{P}_t^\infty)\|f_t - g_t\|_{\mathcal{F}}\right] \\
&\quad + \mathbb{E}\left[B(P, \tilde{P}_{t-1}^\infty) - B(P, \tilde{P}_t^\infty) - B(\tilde{P}_t^\infty, P_t^\infty) - B(P_t^\infty, \tilde{P}_{t-1}^\infty)\right] \\
&\stackrel{(a)}{\leqslant} \mathbb{E}\left[\gamma_{\mathcal{F}}(P_t^\infty, \tilde{P}_t^\infty)\|f_t - g_t\|_{\mathcal{F}}\right] \\
&\quad + \mathbb{E}\left[B(P, \tilde{P}_{t-1}^\infty) - B(P, \tilde{P}_t^\infty)\right] - \mathbb{E}\left[\frac{\eta}{2C}\gamma_{\mathcal{F}}(\tilde{P}_t^\infty, P_t^\infty)^2 + \frac{\eta}{2C}\gamma_{\mathcal{F}}(P_t^\infty, \tilde{P}_{t-1}^\infty)^2\right] \\
&\stackrel{(b)}{\leqslant} \frac{C}{2\eta}\mathbb{E}\left[\|f_t - g_t\|_{\mathcal{F}}^2\right] + \mathbb{E}\left[B(P, \tilde{P}_{t-1}^\infty) - B(P, \tilde{P}_t^\infty)\right] - \mathbb{E}\left[\frac{\eta}{2C}\gamma_{\mathcal{F}}(P_t^\infty, \tilde{P}_{t-1}^\infty)^2\right]
\end{aligned}
$$

where $(a)$ follows from Lemma C.3, and $(b)$ uses the fact that $|xy| \leqslant \frac{1}{2c}|x|^2 + \frac{c}{2}|y|^2$, for any $x, y$, $c > 0$. Summing over $t = 1, \ldots T$ gives us

$$
\begin{aligned}
\sum_{t=1}^{T} \mathbb{E}\left[f_t(\mathbf{x}_t) - f_t(P)\right] &\leqslant \underbrace{\mathbb{E}\left[B(P, \tilde{P}_0^\infty) - B(P, \tilde{P}_T^\infty)\right]}_{S_1} + \sum_{t=1}^{T} \frac{C}{2\eta}\mathbb{E}\left[\|f_t - g_t\|_{\mathcal{F}}^2\right] \\
&\quad - \sum_{t=1}^{T} \frac{\eta}{2C}\mathbb{E}\left[\gamma_{\mathcal{F}}(P_t^\infty, \tilde{P}_{t-1}^\infty)^2\right]
\end{aligned}
$$

To finish the proof of the Theorem, we need to bound $S_1$.

**Bounding $S_1$.** From the definition of $B$, we have

$$B(P, \tilde{P}_0^\infty) - B(P, \tilde{P}_T^\infty) = R(\tilde{P}_T^\infty) - \left\langle P - \tilde{P}_T^\infty, f_{1:T} \right\rangle - R(\tilde{\mathbf{x}}_0^\infty),$$

where we used the fact that $f_{1:0} = 0$. We now rely on Lemma C.1 to convert the above equation, which is currently in terms of $R$, into a quantity which depends on $\Phi$. Using Lemma C.1, we get

$$B(P, \tilde{P}_0^\infty) - B(P, \tilde{P}_T^\infty) = \Phi(f_{1:T}) - \langle P, f_{1:T} \rangle - \Phi(0).$$

From the definition of $\Phi$ we have

$$B(P, \tilde{P}_0^\infty) - B(P, \tilde{P}_T^\infty) = \Phi(f_{1:T}) - \langle P, f_{1:T} \rangle - \Phi(0)$$

$$= \mathbb{E}_\sigma \left[ \inf_{P' \in \mathcal{P}} \left\langle P', f_{1:T} - \sigma \right\rangle \right] - \langle P, f_{1:T} \rangle - \mathbb{E}_\sigma \left[ \inf_{P' \in \mathcal{P}} \left\langle P', -\sigma \right\rangle \right]$$

$$\leqslant \mathbb{E}_\sigma \left[ \langle P, f_{1:T} - \sigma \rangle \right] - \langle P, f_{1:T} \rangle - \mathbb{E}_\sigma \left[ \inf_{P' \in \mathcal{P}} \left\langle P', -\sigma \right\rangle \right]$$

$$= \mathbb{E}_\sigma \left[ \sup_{P' \in \mathcal{P}} \left\langle P', \sigma \right\rangle \right] - \mathbb{E}_\sigma \left[ \langle P, \sigma \rangle \right]$$

$$\leqslant D \mathbb{E}_\sigma \left[ \|\sigma\|_{\mathcal{F}} \right] = \eta D,$$

where the last inequality follows from our bound on the diameter of $\mathcal{P}$. Substituting this in the above regret bound gives us the required result.

## C.2  Proof of Corollary 4.2

To prove the corollary we first show that for our choice of perturbation distribution, $\operatorname{argmin}_{\mathbf{x} \in \mathcal{X}} f(\mathbf{x}) - \sigma(\mathbf{x})$ has a unique minimizer with probability one, for any $f \in \mathcal{F}$. Next, we show that the predictions of OFTPL are stable.

### C.2.1  Intermediate Results

**Lemma C.4** (Unique Minimizer). *Suppose the perturbation function is such that $\sigma(\mathbf{x}) = \langle \bar{\sigma}, \mathbf{x} \rangle$, where $\bar{\sigma} \in \mathbb{R}^d$ is a random vector whose entries are sampled independently from $Exp(\eta)$. Then, for any $f \in \mathcal{F}$, $\operatorname{argmin}_{\mathbf{x} \in \mathcal{X}} f(\mathbf{x}) - \sigma(\mathbf{x})$ has a unique minimizer with probability one.*

*Proof.* Define $\mathbf{x}_f(\sigma)$ as

$$\mathbf{x}_f(\bar{\sigma}) \in \operatorname{argmin}_{\mathbf{x} \in \mathcal{X}} f(\mathbf{x}) - \langle \bar{\sigma}, \mathbf{x} \rangle.$$

For any $\bar{\sigma}_1, \bar{\sigma}_2$ we now show that $\mathbf{x}_f(\bar{\sigma})$ satisfies the following monotonicity property

$$\langle \mathbf{x}_f(\bar{\sigma}_1) - \mathbf{x}_f(\bar{\sigma}_2), \bar{\sigma}_1 - \bar{\sigma}_2 \rangle \geqslant 0.$$

From the optimality of $\mathbf{x}_f(\bar{\sigma}_1), \mathbf{x}_f(\bar{\sigma}_2)$ we have

$$f(\mathbf{x}_f(\bar{\sigma}_1)) - \langle \bar{\sigma}_1, \mathbf{x}_f(\bar{\sigma}_1) \rangle \leqslant f(\mathbf{x}_f(\bar{\sigma}_2)) - \langle \bar{\sigma}_1, \mathbf{x}_f(\bar{\sigma}_2) \rangle$$

$$= f(\mathbf{x}_f(\bar{\sigma}_2)) - \langle \bar{\sigma}_2, \mathbf{x}_f(\bar{\sigma}_2) \rangle + \langle \bar{\sigma}_2 - \bar{\sigma}_1, \mathbf{x}_f(\bar{\sigma}_2) \rangle$$

$$\leqslant f(\mathbf{x}_f(\bar{\sigma}_1)) - \langle \bar{\sigma}_2, \mathbf{x}_f(\bar{\sigma}_1) \rangle + \langle \bar{\sigma}_2 - \bar{\sigma}_1, \mathbf{x}_f(\bar{\sigma}_2) \rangle.$$

This shows that $\langle \bar{\sigma}_2 - \bar{\sigma}_1, \mathbf{x}_f(\bar{\sigma}_2) - \mathbf{x}_f(\bar{\sigma}_1) \rangle \geqslant 0$. To finish the proof of Lemma, we rely on Theorem 1 of Zarantonello [25], which shows that the set of points for which a monotone operator is not single-valued has Lebesgue measure zero. Since the distribution of $\bar{\sigma}$ is absolutely continuous w.r.t Lebesgue measure, this shows that $\operatorname{argmin}_{\mathbf{x} \in \mathcal{X}} f(\mathbf{x}) - \sigma(\mathbf{x})$ has a unique minimizer with probability one. $\square$

### C.2.2  Main Argument

For our choice of perturbation distribution, $\mathbb{E}_\sigma \left[ \|\sigma\|_{\mathcal{F}} \right] = \mathbb{E}_{\bar{\sigma}} \left[ \|\bar{\sigma}\|_\infty \right] = \eta \log d$. We now bound the stability of predictions of OFTPL. First note that for our choice of primal space $(\mathcal{F}, \| \cdot \|_{\mathcal{F}})$, $\gamma_{\mathcal{F}}$ is the Wasserstein-1 metric, which is defined as

$$\gamma_{\mathcal{F}}(P_1, P_2) = \sup_{f \in \mathcal{F}, \|f\|_{\mathcal{F}} \leqslant 1} \left| \mathbb{E}_{\mathbf{x} \sim P_1} \left[ f(\mathbf{x}) \right] - \mathbb{E}_{\mathbf{x} \sim P_2} \left[ f(\mathbf{x}) \right] \right| = \inf_{Q \in \Gamma(P_1, P_2)} \mathbb{E}_{(\mathbf{x}_1, \mathbf{x}_2) \sim Q} \left[ \|\mathbf{x}_1 - \mathbf{x}_2\|_1 \right],$$

where $\Gamma(P_1, P_2)$ is the set of all probability measures on $\mathcal{X} \times \mathcal{X}$ with marginals $P_1, P_2$ on the first and second factors respectively. Define $\mathbf{x}_f(\bar{\sigma})$ as

$$\mathbf{x}_f(\bar{\sigma}) \in \operatorname{argmin}_{\mathbf{x} \in \mathcal{X}} f(\mathbf{x}) - \langle \bar{\sigma}, \mathbf{x} \rangle.$$

---
**Algorithm 3** OFTPL for convex-concave games
---
1: **Input:** Perturbation Distributions $P^1_{\text{PRTB}}, P^2_{\text{PRTB}}$ of $\mathbf{x}, \mathbf{y}$ players, number of samples $m$, iterations $T$
2: **for** $t = 1 \ldots T$ **do**
3: &emsp; **if** $t = 1$ **then**
4: &emsp;&emsp; Sample $\{\sigma^1_{1,j}\}^m_{j=1}, \{\sigma^2_{1,j}\}^m_{j=1}$ from $P^1_{\text{PRTB}}, P^2_{\text{PRTB}}$
5: &emsp;&emsp; $\mathbf{x}_1 = \frac{1}{m}\sum^m_{j=1}\left[\operatorname{argmin}_{\mathbf{x}\in\mathcal{X}}\left\langle -\sigma^1_{1,j},\mathbf{x}\right\rangle\right], \mathbf{y}_1 = \frac{1}{m}\left[\sum^m_{j=1}\operatorname{argmax}_{\mathbf{y}\in\mathcal{Y}}\left\langle\sigma^2_{1,j},\mathbf{y}\right\rangle\right]$
6: &emsp;&emsp; **continue**
7: &emsp; **end if**
8: &emsp; //Compute guesses
9: &emsp; **for** $j = 1 \ldots m$ **do**
10: &emsp;&emsp; Sample $\sigma^1_{t,j} \sim P^1_{\text{PRTB}}, \sigma^2_{t,j} \sim P^2_{\text{PRTB}}$
11: &emsp;&emsp; $\tilde{\mathbf{x}}_{t-1,j} = \underset{\mathbf{x}\in\mathcal{X}}{\operatorname{argmin}}\left\langle\sum^{t-1}_{i=1}\nabla_{\mathbf{x}}f(\mathbf{x}_i,\mathbf{y}_i) - \sigma^1_{t,j},\mathbf{x}\right\rangle$
12: &emsp;&emsp; $\tilde{\mathbf{y}}_{t-1,j} = \underset{\mathbf{y}\in\mathcal{Y}}{\operatorname{argmax}}\left\langle\sum^{t-1}_{i=1}\nabla_{\mathbf{y}}f(\mathbf{x}_i,\mathbf{y}_i) + \sigma^2_{t,j},\mathbf{y}\right\rangle$
13: &emsp; **end for**
14: &emsp; $\tilde{\mathbf{x}}_{t-1} = \frac{1}{m}\sum^m_{j=1}\tilde{\mathbf{x}}_{t-1,j}, \tilde{\mathbf{y}}_{t-1} = \frac{1}{m}\sum^m_{j=1}\tilde{\mathbf{y}}_{t-1,j}$
15: &emsp; //Use the guesses to compute the next action
16: &emsp; **for** $j = 1 \ldots m$ **do**
17: &emsp;&emsp; Sample $\sigma^1_{t,j} \sim P^1_{\text{PRTB}}, \sigma^2_{t,j} \sim P^2_{\text{PRTB}}$
18: &emsp;&emsp; $\mathbf{x}_{t,j} = \underset{\mathbf{x}\in\mathcal{X}}{\operatorname{argmin}}\left\langle\sum^{t-1}_{i=1}\nabla_{\mathbf{x}}f(\mathbf{x}_i,\mathbf{y}_i) + \nabla_{\mathbf{x}}f(\tilde{\mathbf{x}}_{t-1},\tilde{\mathbf{y}}_{t-1}) - \sigma^1_{t,j},\mathbf{x}\right\rangle$
19: &emsp;&emsp; $\mathbf{y}_{t,j} = \underset{\mathbf{y}\in\mathcal{Y}}{\operatorname{argmax}}\left\langle\sum^{t-1}_{i=1}\nabla_{\mathbf{y}}f(\mathbf{x}_i,\mathbf{y}_i) + \nabla_{\mathbf{y}}f(\tilde{\mathbf{x}}_{t-1},\tilde{\mathbf{y}}_{t-1}) + \sigma^2_{t,j},\mathbf{y}\right\rangle$
20: &emsp; **end for**
21: &emsp; $\mathbf{x}_t = \frac{1}{m}\sum^m_{j=1}\mathbf{x}_{t,j}, \mathbf{y}_t = \frac{1}{m}\sum^m_{j=1}\mathbf{y}_{t,j}$
22: **end for**
23: **return** $\{(\mathbf{x}_t,\mathbf{y}_t)\}^T_{t=1}$
---

Note that $\nabla\Phi(f)$ is the distribution of random variable $\mathbf{x}_f(\bar{\sigma})$. Suggala and Netrapalli [11] show that for any $f, g \in \mathcal{F}$

$$\mathbb{E}_{\bar{\sigma}}\left[\|\mathbf{x}_f(\bar{\sigma}) - \mathbf{x}_g(\bar{\sigma})\|_1\right] \leqslant \frac{125d^2D}{\eta}\|f - g\|_{\mathcal{F}}.$$

Since $\gamma_{\mathcal{F}}(\nabla\Phi(f), \nabla\Phi(g)) \leqslant \mathbb{E}_{\bar{\sigma}}\left[\|\mathbf{x}_f(\bar{\sigma}) - \mathbf{x}_g(\bar{\sigma})\|_1\right]$, this shows that OFTPL is $O\left(d^2D\eta^{-1}\right)$ stable w.r.t $\|\cdot\|_{\mathcal{F}}$. Substituting the stability bound in the regret bound of Theorem 4.2 shows that

$$\sup_{P\in\mathcal{P}}\mathbb{E}\left[\sum^T_{t=1}f_t(\mathbf{x}_t) - f_t(P)\right] = \eta D\log d$$
$$+ O\left(\sum^T_{t=1}\frac{d^2D}{\eta}\mathbb{E}\left[\|f_t - g_t\|^2_{\mathcal{F}}\right] - \sum^T_{t=1}\frac{\eta}{d^2D}\mathbb{E}\left[\gamma_{\mathcal{F}}(P^\infty_t, \tilde{P}^\infty_{t-1})^2\right]\right).$$

# D   Convex-Concave Games

Our algorithm for convex-concave games is presented in Algorithm 3. Before presenting the proof of Theorem 5.1, we first present a more general result in Section D.1. Theorem 5.1 immediately follows from our general result by instantiating it for the uniform noise distribution.

## D.1   General Result

**Theorem D.1.** *Consider the minimax game in Equation (1). Suppose $f$ is convex in $\mathbf{x}$, concave in $\mathbf{y}$ and is Holder smooth w.r.t some norm $\|\cdot\|$*

$$\|\nabla_{\mathbf{x}}f(\mathbf{x},\mathbf{y}) - \nabla_{\mathbf{x}}f(\mathbf{x}',\mathbf{y}')\|_* \leqslant L_1\|\mathbf{x} - \mathbf{x}'\|^\alpha + L_2\|\mathbf{y} - \mathbf{y}'\|^\alpha,$$
$$\|\nabla_{\mathbf{y}}f(\mathbf{x},\mathbf{y}) - \nabla_{\mathbf{y}}f(\mathbf{x}',\mathbf{y}')\|_* \leqslant L_2\|\mathbf{x} - \mathbf{x}'\|^\alpha + L_1\|\mathbf{y} - \mathbf{y}'\|^\alpha.$$

*Define diameter of sets $\mathcal{X}, \mathcal{Y}$ as $D = \max\{\sup_{\mathbf{x}_1,\mathbf{x}_2\in\mathcal{X}}\|\mathbf{x}_1 - \mathbf{x}_2\|, \sup_{\mathbf{y}_1,\mathbf{y}_2\in\mathcal{Y}}\|\mathbf{y}_1 - \mathbf{y}_2\|\}$. Let $L = \{L_1, L_2\}$. Suppose both $\mathbf{x}$ and $\mathbf{y}$ players use Algorithm 1 to solve the minimax game. Suppose the perturbation distributions $P^1_{PRTB}, P^2_{PRTB}$, used by $\mathbf{x}, \mathbf{y}$ players are absolutely continuous*

*and satisfy* $\mathbb{E}_{\sigma \sim P^1_{PRTB}}[\|\sigma\|_*] = \mathbb{E}_{\sigma \sim P^2_{PRTB}}[\|\sigma\|_*] = \eta$. *Suppose the predictions of both the players are $C\eta^{-1}$-stable w.r.t $\|\cdot\|_*$. Suppose the guesses used by $\mathbf{x}, \mathbf{y}$ players in the $t^{th}$ iteration are $\nabla_{\mathbf{x}} f(\tilde{\mathbf{x}}_{t-1}, \tilde{\mathbf{y}}_{t-1}), \nabla_{\mathbf{y}} f(\tilde{\mathbf{x}}_{t-1}, \tilde{\mathbf{y}}_{t-1})$, where $\tilde{\mathbf{x}}_{t-1}, \tilde{\mathbf{y}}_{t-1}$ denote the predictions of $\mathbf{x}, \mathbf{y}$ players in the $t^{th}$ iteration, if guess $g_t = 0$ was used in that iteration. Then the iterates $\{(\mathbf{x}_t, \mathbf{y}_t)\}_{t=1}^T$ generated by the OFTPL based algorithm satisfy*

$$\sup_{\mathbf{x} \in \mathcal{X}, \mathbf{y} \in \mathcal{Y}} \mathbb{E}\left[ f\left( \frac{1}{T} \sum_{t=1}^T \mathbf{x}_t, \mathbf{y} \right) - f\left( \mathbf{x}, \frac{1}{T} \sum_{t=1}^T \mathbf{y}_t \right) \right] \leqslant 2L_1 \left( \frac{\Psi_1 \Psi_2 D}{\sqrt{m}} \right)^{1+\alpha} + \frac{2\eta D}{T}$$

$$+ \frac{20CL^2}{\eta} \left( \frac{\Psi_1 \Psi_2 D}{\sqrt{m}} \right)^{2\alpha} + 10L \left( \frac{5CL}{\eta} \right)^{\frac{1+\alpha}{1-\alpha}}$$

*Proof.* Since both the players are responding to each others actions using OFTPL, using Theorem 4.1, we get the following regret bounds for the players

$$\sup_{\mathbf{x} \in \mathcal{X}} \mathbb{E}\left[ \sum_{t=1}^T f(\mathbf{x}_t, \mathbf{y}_t) - f(\mathbf{x}, \mathbf{y}_t) \right] \leqslant L_1 T \left( \frac{\Psi_1 \Psi_2 D}{\sqrt{m}} \right)^{1+\alpha} + \eta D$$

$$+ \frac{C}{2\eta} \sum_{t=1}^T \mathbb{E}\left[ \|\nabla_{\mathbf{x}} f(\mathbf{x}_t, \mathbf{y}_t) - \nabla_{\mathbf{x}} f(\tilde{\mathbf{x}}_{t-1}, \tilde{\mathbf{y}}_{t-1})\|_*^2 \right]$$

$$- \frac{\eta}{2C} \sum_{t=1}^T \mathbb{E}\left[ \|\mathbf{x}_t^\infty - \tilde{\mathbf{x}}_{t-1}^\infty\|^2 \right].$$

$$\sup_{\mathbf{y} \in \mathcal{Y}} \mathbb{E}\left[ \sum_{t=1}^T f(\mathbf{x}_t, \mathbf{y}) - f(\mathbf{x}_t, \mathbf{y}_t) \right] \leqslant L_1 T \left( \frac{\Psi_1 \Psi_2 D}{\sqrt{m}} \right)^{1+\alpha} + \eta D$$

$$+ \frac{C}{2\eta} \sum_{t=1}^T \mathbb{E}\left[ \|\nabla_{\mathbf{y}} f(\mathbf{x}_t, \mathbf{y}_t) - \nabla_{\mathbf{y}} f(\tilde{\mathbf{x}}_{t-1}, \tilde{\mathbf{y}}_{t-1})\|_*^2 \right]$$

$$- \frac{\eta}{2C} \sum_{t=1}^T \mathbb{E}\left[ \|\mathbf{y}_t^\infty - \tilde{\mathbf{y}}_{t-1}^\infty\|^2 \right].$$

First, consider the regret of the $\mathbf{x}$ player. Since $\|a_1 + \cdots + a_5\|^2 \leqslant 5(\|a_1\|^2 \cdots + \|a_5\|^2)$, we have

$$\|\nabla_{\mathbf{x}} f(\mathbf{x}_t, \mathbf{y}_t) - \nabla_{\mathbf{x}} f(\tilde{\mathbf{x}}_{t-1}, \tilde{\mathbf{y}}_{t-1})\|_*^2 \leqslant 5\|\nabla_{\mathbf{x}} f(\mathbf{x}_t, \mathbf{y}_t) - \nabla_{\mathbf{x}} f(\mathbf{x}_t^\infty, \mathbf{y}_t)\|_*^2$$

$$+ 5\|\nabla_{\mathbf{x}} f(\mathbf{x}_t^\infty, \mathbf{y}_t) - \nabla_{\mathbf{x}} f(\mathbf{x}_t^\infty, \mathbf{y}_t^\infty)\|_*^2$$

$$+ 5\|\nabla_{\mathbf{x}} f(\mathbf{x}_t^\infty, \mathbf{y}_t^\infty) - \nabla_{\mathbf{x}} f(\tilde{\mathbf{x}}_{t-1}^\infty, \tilde{\mathbf{y}}_{t-1}^\infty)\|_*^2$$

$$+ 5\|\nabla_{\mathbf{x}} f(\tilde{\mathbf{x}}_{t-1}^\infty, \tilde{\mathbf{y}}_{t-1}^\infty) - \nabla_{\mathbf{x}} f(\tilde{\mathbf{x}}_{t-1}, \tilde{\mathbf{y}}_{t-1})\|_*^2$$

$$+ 5\|\nabla_{\mathbf{x}} f(\tilde{\mathbf{x}}_{t-1}, \tilde{\mathbf{y}}_{t-1}) - \nabla_{\mathbf{x}} f(\tilde{\mathbf{x}}_{t-1}, \tilde{\mathbf{y}}_{t-1})\|_*^2$$

$$\overset{(a)}{\leqslant} 5L_1^2 \|\mathbf{x}_t - \mathbf{x}_t^\infty\|^{2\alpha} + 5L_1^2 \|\tilde{\mathbf{x}}_{t-1} - \tilde{\mathbf{x}}_{t-1}^\infty\|^{2\alpha}$$

$$+ 5L_2^2 \|\mathbf{y}_t - \mathbf{y}_t^\infty\|^{2\alpha} + 5L_2^2 \|\tilde{\mathbf{y}}_{t-1} - \tilde{\mathbf{y}}_{t-1}^\infty\|^{2\alpha}$$

$$+ 5\|\nabla_{\mathbf{x}} f(\mathbf{x}_t^\infty, \mathbf{y}_t^\infty) - \nabla_{\mathbf{x}} f(\tilde{\mathbf{x}}_{t-1}^\infty, \tilde{\mathbf{y}}_{t-1}^\infty)\|_*^2.$$

where $(a)$ follows from the Holder's smoothness of $f$. Using a similar technique as in the proof of Theorem 4.1, relying on Holders inequality, we get

$$\mathbb{E}\left[ \|\mathbf{x}_t - \mathbf{x}_t^\infty\|^{2\alpha} | \tilde{\mathbf{x}}_{t-1}, \tilde{\mathbf{y}}_{t-1}, \mathbf{x}_{1:t-1}, \mathbf{y}_{1:t-1} \right] \leqslant \mathbb{E}\left[ \|\mathbf{x}_t - \mathbf{x}_t^\infty\|^2 | \tilde{\mathbf{x}}_{t-1}, \tilde{\mathbf{y}}_{t-1}, \mathbf{x}_{1:t-1}, \mathbf{y}_{1:t-1} \right]^\alpha$$

$$\leqslant \Psi_1^{2\alpha} \mathbb{E}\left[ \|\mathbf{x}_t - \mathbf{x}_t^\infty\|_2^2 | \tilde{\mathbf{x}}_{t-1}, \tilde{\mathbf{y}}_{t-1}, \mathbf{x}_{1:t-1}, \mathbf{y}_{1:t-1} \right]^\alpha$$

$$\overset{(a)}{\leqslant} \left( \frac{\Psi_1 \Psi_2 D}{\sqrt{m}} \right)^{2\alpha},$$

where $(a)$ follows from the fact that conditioned on past randomness, $\mathbf{x}_t - \mathbf{x}_t^\infty$ is the average of $m$ i.i.d bounded mean 0 random variables, the variance of which scales as $O(D^2/m)$. A similar bound

holds for the expectation of other quantities appearing in the RHS of the above equation. Using this, the regret of $\mathbf{x}$ player can be upper bounded as

$$\sup_{\mathbf{x}\in\mathcal{X}} \mathbb{E}\left[\sum_{t=1}^{T} f(\mathbf{x}_t, \mathbf{y}_t) - f(\mathbf{x}, \mathbf{y}_t)\right] \leqslant L_1 T \left(\frac{\Psi_1\Psi_2 D}{\sqrt{m}}\right)^{1+\alpha} + \eta D + \frac{10CL^2 T}{\eta}\left(\frac{\Psi_1\Psi_2 D}{\sqrt{m}}\right)^{2\alpha}$$

$$+ \frac{5C}{2\eta}\sum_{t=1}^{T}\mathbb{E}\left[\|\nabla_{\mathbf{x}} f(\mathbf{x}_t^\infty, \mathbf{y}_t^\infty) - \nabla_{\mathbf{x}} f(\tilde{\mathbf{x}}_{t-1}^\infty, \tilde{\mathbf{y}}_{t-1}^\infty)\|_*^2\right]$$

$$- \frac{\eta}{2C}\sum_{t=1}^{T}\mathbb{E}\left[\|\mathbf{x}_t^\infty - \tilde{\mathbf{x}}_{t-1}^\infty\|^2\right].$$

Similarly, the regret of $\mathbf{y}$ player can be bounded as

$$\sup_{\mathbf{y}\in\mathcal{Y}} \mathbb{E}\left[\sum_{t=1}^{T} f(\mathbf{x}_t, \mathbf{y}) - f(\mathbf{x}_t, \mathbf{y}_t)\right] \leqslant L_1 T \left(\frac{\Psi_1\Psi_2 D}{\sqrt{m}}\right)^{1+\alpha} + \eta D + \frac{10CL^2 T}{\eta}\left(\frac{\Psi_1\Psi_2 D}{\sqrt{m}}\right)^{2\alpha}$$

$$+ \frac{5C}{2\eta}\sum_{t=1}^{T}\mathbb{E}\left[\|\nabla_{\mathbf{y}} f(\mathbf{x}_t^\infty, \mathbf{y}_t^\infty) - \nabla_{\mathbf{y}} f(\tilde{\mathbf{x}}_{t-1}^\infty, \tilde{\mathbf{y}}_{t-1}^\infty)\|_*^2\right]$$

$$- \frac{\eta}{2C}\sum_{t=1}^{T}\mathbb{E}\left[\|\mathbf{y}_t^\infty - \tilde{\mathbf{y}}_{t-1}^\infty\|^2\right].$$

Summing the above two inequalities, we get

$$\sup_{\mathbf{x}\in\mathcal{X}\mathbf{y}\in\mathcal{Y}} \mathbb{E}\left[\sum_{t=1}^{T} f(\mathbf{x}_t, \mathbf{y}) - f(\mathbf{x}, \mathbf{y}_t)\right] \leqslant 2L_1 T \left(\frac{\Psi_1\Psi_2 D}{\sqrt{m}}\right)^{1+\alpha} + 2\eta D + \frac{20CL^2 T}{\eta}\left(\frac{\Psi_1\Psi_2 D}{\sqrt{m}}\right)^{2\alpha}$$

$$+ \frac{5C}{2\eta}\sum_{t=1}^{T}\mathbb{E}\left[\|\nabla_{\mathbf{x}} f(\mathbf{x}_t^\infty, \mathbf{y}_t^\infty) - \nabla_{\mathbf{x}} f(\tilde{\mathbf{x}}_{t-1}^\infty, \tilde{\mathbf{y}}_{t-1}^\infty)\|_*^2\right]$$

$$+ \frac{5C}{2\eta}\sum_{t=1}^{T}\mathbb{E}\left[\|\nabla_{\mathbf{y}} f(\mathbf{x}_t^\infty, \mathbf{y}_t^\infty) - \nabla_{\mathbf{y}} f(\tilde{\mathbf{x}}_{t-1}^\infty, \tilde{\mathbf{y}}_{t-1}^\infty)\|_*^2\right]$$

$$- \frac{\eta}{2C}\sum_{t=1}^{T}\left(\mathbb{E}\left[\|\mathbf{y}_t^\infty - \tilde{\mathbf{y}}_{t-1}^\infty\|^2\right] + \mathbb{E}\left[\|\mathbf{x}_t^\infty - \tilde{\mathbf{x}}_{t-1}^\infty\|^2\right]\right).$$

From Holder's smoothness assumption on $f$, we have

$$\mathbb{E}\left[\|\nabla_{\mathbf{x}} f(\mathbf{x}_t^\infty, \mathbf{y}_t^\infty) - \nabla_{\mathbf{x}} f(\tilde{\mathbf{x}}_{t-1}^\infty, \tilde{\mathbf{y}}_{t-1}^\infty)\|_*^2\right] \leqslant 2\mathbb{E}\left[\|\nabla_{\mathbf{x}} f(\mathbf{x}_t^\infty, \mathbf{y}_t^\infty) - \nabla_{\mathbf{x}} f(\mathbf{x}_t^\infty, \tilde{\mathbf{y}}_{t-1}^\infty)\|_*^2\right]$$

$$+ 2\mathbb{E}\left[\|\nabla_{\mathbf{x}} f(\mathbf{x}_t^\infty, \tilde{\mathbf{y}}_{t-1}^\infty) - \nabla_{\mathbf{x}} f(\tilde{\mathbf{x}}_{t-1}^\infty, \tilde{\mathbf{y}}_{t-1}^\infty)\|_*^2\right]$$

$$\overset{(a)}{\leqslant} 2L^2 \mathbb{E}\left[\|\mathbf{x}_t^\infty - \tilde{\mathbf{x}}_{t-1}^\infty\|^{2\alpha}\right] + 2L^2 \mathbb{E}\left[\|\mathbf{y}_t^\infty - \tilde{\mathbf{y}}_{t-1}^\infty\|^{2\alpha}\right],$$

Using a similar argument, we get

$$\mathbb{E}\left[\|\nabla_{\mathbf{y}} f(\mathbf{x}_t^\infty, \mathbf{y}_t^\infty) - \nabla_{\mathbf{y}} f(\tilde{\mathbf{x}}_{t-1}^\infty, \tilde{\mathbf{y}}_{t-1}^\infty)\|_*^2\right] \leqslant 2L^2 \mathbb{E}\left[\|\mathbf{x}_t^\infty - \tilde{\mathbf{x}}_{t-1}^\infty\|^{2\alpha}\right] + 2L^2 \mathbb{E}\left[\|\mathbf{y}_t^\infty - \tilde{\mathbf{y}}_{t-1}^\infty\|^{2\alpha}\right].$$

Plugging this in the previous bound, we get

$$\sup_{\mathbf{x}\in\mathcal{X}\mathbf{y}\in\mathcal{Y}} \mathbb{E}\left[\sum_{t=1}^{T} f(\mathbf{x}_t, \mathbf{y}) - f(\mathbf{x}, \mathbf{y}_t)\right] \leqslant 2L_1 T \left(\frac{\Psi_1\Psi_2 D}{\sqrt{m}}\right)^{1+\alpha} + 2\eta D + \frac{20CL^2 T}{\eta}\left(\frac{\Psi_1\Psi_2 D}{\sqrt{m}}\right)^{2\alpha}$$

$$+ \frac{10CL^2}{\eta}\sum_{t=1}^{T}\left(\mathbb{E}\left[\|\mathbf{x}_t^\infty - \tilde{\mathbf{x}}_{t-1}^\infty\|^{2\alpha}\right] + \mathbb{E}\left[\|\mathbf{y}_t^\infty - \tilde{\mathbf{y}}_{t-1}^\infty\|^{2\alpha}\right]\right)$$

$$- \frac{\eta}{2C}\sum_{t=1}^{T}\left(\mathbb{E}\left[\|\mathbf{y}_t^\infty - \tilde{\mathbf{y}}_{t-1}^\infty\|^2\right] + \mathbb{E}\left[\|\mathbf{x}_t^\infty - \tilde{\mathbf{x}}_{t-1}^\infty\|^2\right]\right).$$

**Case** $\alpha = 1$. We first consider the case of $\alpha = 1$. In this case, choosing $\eta > \sqrt{20}CL$, we get

$$\sup_{\mathbf{x}\in\mathcal{X}\mathbf{y}\in\mathcal{Y}} \mathbb{E}\left[\sum_{t=1}^{T} f(\mathbf{x}_t, \mathbf{y}) - f(\mathbf{x}, \mathbf{y}_t)\right] \leqslant 2L_1 T \left(\frac{\Psi_1\Psi_2 D}{\sqrt{m}}\right)^{1+\alpha} + 2\eta D + \frac{20CL^2 T}{\eta}\left(\frac{\Psi_1\Psi_2 D}{\sqrt{m}}\right)^{2\alpha}.$$

**General** $\alpha$. The more general case relies on AM-GM inequality. Consider the following

$$\frac{10CL^2}{\eta}\|\mathbf{x}_t^\infty - \tilde{\mathbf{x}}_{t-1}^\infty\|^{2\alpha} = \left((2\alpha C)^{\frac{\alpha}{1-\alpha}}\eta^{-\frac{1+\alpha}{1-\alpha}}(10CL^2)^{\frac{1}{1-\alpha}}\right)^{1-\alpha}\left(\frac{\|\mathbf{x}_t^\infty - \tilde{\mathbf{x}}_{t-1}^\infty\|^2}{2\alpha C\eta^{-1}}\right)^{\alpha}$$

$$\overset{(a)}{\leqslant} (1-\alpha)\left((2\alpha C)^{\frac{\alpha}{1-\alpha}}\eta^{-\frac{1+\alpha}{1-\alpha}}(10CL^2)^{\frac{1}{1-\alpha}}\right) + \frac{\eta}{2C}\|\mathbf{x}_t^\infty - \tilde{\mathbf{x}}_{t-1}^\infty\|^2$$

$$= \sqrt{20}L\left(\frac{\sqrt{20}CL}{\eta}\right)^{\frac{1+\alpha}{1-\alpha}} + \frac{\eta}{2C}\|\mathbf{x}_t^\infty - \tilde{\mathbf{x}}_{t-1}^\infty\|^2$$

where $(a)$ follows from AM-GM inequality. Plugging this in the previous bound, we get

$$\sup_{\mathbf{x}\in\mathcal{X}\mathbf{y}\in\mathcal{Y}} \mathbb{E}\left[\sum_{t=1}^{T} f(\mathbf{x}_t, \mathbf{y}) - f(\mathbf{x}, \mathbf{y}_t)\right] \leqslant 2L_1 T \left(\frac{\Psi_1\Psi_2 D}{\sqrt{m}}\right)^{1+\alpha} + 2\eta D$$

$$+ \frac{20CL^2 T}{\eta}\left(\frac{\Psi_1\Psi_2 D}{\sqrt{m}}\right)^{2\alpha} + 4\sqrt{5}LT\left(\frac{\sqrt{20}CL}{\eta}\right)^{\frac{1+\alpha}{1-\alpha}}.$$

The claim of the theorem then follows from the observation that

$$\mathbb{E}\left[f\left(\frac{1}{T}\sum_{t=1}^{T}\mathbf{x}_t, \mathbf{y}\right) - f\left(\mathbf{x}, \frac{1}{T}\sum_{t=1}^{T}\mathbf{y}_t\right)\right] \leqslant \frac{1}{T}\mathbb{E}\left[\sum_{t=1}^{T} f(\mathbf{x}_t, \mathbf{y}) - f(\mathbf{x}, \mathbf{y}_t)\right].$$

$\square$

## D.2 Proof of Theorem 5.1

To prove the Theorem, we instantiate Theorem D.1 for the uniform noise distribution. As shown in Corollary 4.1, the predictions of OFTPL are $dD\eta^{-1}$-stable in this case. Plugging this in the bound of Theorem D.1 and using the fact that $\Psi_1 = \Psi_2 = 1$ and $\alpha = 1$ gives us

$$\sup_{\mathbf{x}\in\mathcal{X},\mathbf{y}\in\mathcal{Y}} \mathbb{E}\left[f\left(\frac{1}{T}\sum_{t=1}^{T}\mathbf{x}_t, \mathbf{y}\right) - f\left(\mathbf{x}, \frac{1}{T}\sum_{t=1}^{T}\mathbf{y}_t\right)\right] \leqslant 2L\left(\frac{D}{\sqrt{m}}\right)^2 + \frac{2\eta D}{T}$$

$$+ \frac{20dDL^2}{\eta}\left(\frac{D}{\sqrt{m}}\right)^2 + 10L\left(\frac{5dDL}{\eta}\right)^\infty.$$

Plugging in $\eta = 6dD(L+1)$, $m = T$ in the above bound gives us

$$\sup_{\mathbf{x}\in\mathcal{X},\mathbf{y}\in\mathcal{Y}} \mathbb{E}\left[f\left(\frac{1}{T}\sum_{t=1}^{T}\mathbf{x}_t, \mathbf{y}\right) - f\left(\mathbf{x}, \frac{1}{T}\sum_{t=1}^{T}\mathbf{y}_t\right)\right] \leqslant O\left(\frac{dD^2(L+1)}{T}\right).$$

# E   Nonconvex-Nonconcave Games

Our algorithm for nonconvex-nonconcave games is presented in Algorithm 4. Note that in each iteration of this game, both the players play empirical distributions $(P_t, Q_t)$. Before presenting the proof of Theorem 5.2, we first present a more general result in Section E.2. Theorem 5.2 immediately follows from our general result by instantiating it for exponential noise distribution.

## E.1   Primal Dual Spaces

In this section, we present some integral probability metrics induced by popular choices of functions spaces $(\mathcal{F}, \|\cdot\|_\mathcal{F})$.

---

**Algorithm 4** OFTPL for nonconvex-nonconcave games

---

1: **Input:** Perturbation Distributions $P^1_{\text{PRTB}}, P^2_{\text{PRTB}}$ of $\mathbf{x}, \mathbf{y}$ players, number of samples $m$, iterations $T$
2: **for** $t = 1 \dots T$ **do**
3:     **if** $t = 1$ **then**
4:         **for** $j = 1 \dots m$ **do**
5:             Sample $\sigma^1_{t,j} \sim P^1_{\text{PRTB}}, \sigma^2_{t,j} \sim P^2_{\text{PRTB}}$
6:             $\mathbf{x}_{1,j} = \text{argmin}_{\mathbf{x} \in \mathcal{X}} -\sigma^1_{1,j}(\mathbf{x})$
7:             $\mathbf{y}_{1,j} = \text{argmax}_{\mathbf{y} \in \mathcal{Y}} \sigma^2_{1,j}(\mathbf{y})$
8:         **end for**
9:         Let $P_1, Q_1$ be the empirical distributions over $\{\mathbf{x}_{1,j}\}^m_{j=1}, \{\mathbf{y}_{1,j}\}^m_{j=1}$
10:         **continue**
11:     **end if**
12:     //Compute guesses
13:     **for** $j = 1 \dots m$ **do**
14:         Sample $\sigma^1_{t,j} \sim P^1_{\text{PRTB}}, \sigma^2_{t,j} \sim P^2_{\text{PRTB}}$
15:         $\tilde{\mathbf{x}}_{t-1,j} = \text{argmin}_{\mathbf{x} \in \mathcal{X}} \sum^{t-1}_{i=1} f(\mathbf{x}, Q_i) - \sigma^1_{t,j}(\mathbf{x})$
16:         $\tilde{\mathbf{y}}_{t-1,j} = \text{argmax}_{\mathbf{y} \in \mathcal{Y}} \sum^{t-1}_{i=1} f(P_i, \mathbf{y}) + \sigma^2_{t,j}(\mathbf{y})$
17:     **end for**
18:     Let $\tilde{P}_{t-1}, \tilde{Q}_{t-1}$ be the empirical distributions over $\{\tilde{\mathbf{x}}_{t-1,j}\}^m_{j=1}, \{\tilde{\mathbf{y}}_{t-1,j}\}^m_{j=1}$
19:     //Use the guesses to compute the next action
20:     **for** $j = 1 \dots m$ **do**
21:         Sample $\sigma^1_{t,j} \sim P^1_{\text{PRTB}}, \sigma^2_{t,j} \sim P^2_{\text{PRTB}}$
22:         $\mathbf{x}_{t,j} = \text{argmin}_{\mathbf{x} \in \mathcal{X}} \sum^{t-1}_{i=1} f(\mathbf{x}, Q_i) + f(\mathbf{x}, \tilde{Q}_{t-1}) - \sigma^1_{t,j}(\mathbf{x})$
23:         $\mathbf{y}_{t,j} = \text{argmax}_{\mathbf{y} \in \mathcal{Y}} \sum^{t-1}_{i=1} f(P_i, \mathbf{y}) + f(\tilde{P}_{t-1}, \mathbf{y}) + \sigma^2_{t,j}(\mathbf{y})$
24:     **end for**
25:     Let $P_t, Q_t$ be the empirical distributions over $\{\mathbf{x}_{t,j}\}^m_{j=1}, \{\mathbf{y}_{t,j}\}^m_{j=1}$
26: **end for**
27: **return** $\{(P_t, Q_t)\}^T_{t=1}$

---

| $\gamma_{\mathcal{F}}(P, Q)$ | $\|f\|_{\mathcal{F}}$ | $\mathcal{F}$ |
|---|---|---|
| Dudley Metric | $\text{Lip}(f) + \|f\|_\infty$ | $\{f : \text{Lip}(f) + \|f\|_\infty < \infty\}$ |
| Kantorovich Metric (or) Wasserstein-1 Metric | $\text{Lip}(f)$ | $\{f : \text{Lip}(f) < \infty\}$ |
| Total Variation (TV) Distance | $\|f\|_\infty$ | $\{f : \|f\|_\infty < \infty\}$ |
| Maximum Mean Discrepancy (MMD) for RKHS $\mathcal{H}$ | $\|f\|_{\mathcal{H}}$ | $\{f : \|f\|_{\mathcal{H}} < \infty\}$ |

Table 1: Table showing some popular Integral Probability Metrics. Here $\text{Lip}(f)$ is the Lipschitz constant of $f$ which is defined as $\sup_{\mathbf{x}, \mathbf{y} \in \mathcal{X}} |f(\mathbf{x}) - f(\mathbf{y})|/\|\mathbf{x} - \mathbf{y}\|$ and $\|f\|_\infty$ is the supremum norm of $f$.

## E.2 General Result

**Theorem E.1.** *Consider the minimax game in Equation (1). Suppose the domains $\mathcal{X}, \mathcal{Y}$ are compact subsets of $\mathbb{R}^d$. Let $\mathcal{F}, \mathcal{F}'$ be the set of Lipschitz functions over $\mathcal{X}, \mathcal{Y}$, and $\|g_1\|_{\mathcal{F}}, \|g_2\|_{\mathcal{F}'}$ be the Lipschitz constants of functions $g_1 : \mathcal{X} \to \mathbb{R}, g_2 : \mathcal{Y} \to \mathbb{R}$ w.r.t some norm $\|\cdot\|$. Suppose $f$ is such that $\max\{\sup_{\mathbf{x} \in \mathcal{X}} \|f(\cdot, \mathbf{y})\|_{\mathcal{F}}, \sup_{\mathbf{y} \in \mathcal{Y}} \|f(\mathbf{x}, \cdot)\|_{\mathcal{F}'}\} \leqslant G$ and satisfies the following smoothness property*

$$\|\nabla_{\mathbf{x}} f(\mathbf{x}, \mathbf{y}) - \nabla_{\mathbf{x}} f(\mathbf{x}', \mathbf{y}')\|_* \leqslant L\|\mathbf{x} - \mathbf{x}'\| + L\|\mathbf{y} - \mathbf{y}'\|,$$
$$\|\nabla_{\mathbf{y}} f(\mathbf{x}, \mathbf{y}) - \nabla_{\mathbf{y}} f(\mathbf{x}', \mathbf{y}')\|_* \leqslant L\|\mathbf{x} - \mathbf{x}'\| + L\|\mathbf{y} - \mathbf{y}'\|.$$

*Let $\mathcal{P}, \mathcal{Q}$ be the set of probability distributions over $\mathcal{X}, \mathcal{Y}$. Define diameter of $\mathcal{P}, \mathcal{Q}$ as $D = \max\{\sup_{P_1, P_2 \in \mathcal{P}} \gamma_{\mathcal{F}}(P_1, P_2), \sup_{Q_1, Q_2 \in \mathcal{Q}} \gamma_{\mathcal{F}'}(Q_1, Q_2)\}$. Suppose both $\mathbf{x}, \mathbf{y}$ players use Algorithm 2 to solve the game. Suppose the perturbation distributions $P^1_{PRTB}, P^2_{PRTB}$, used by $\mathbf{x}, \mathbf{y}$ players are such that $\text{argmin}_{\mathbf{x} \in \mathcal{X}} f(\mathbf{x}) - \sigma(\mathbf{x}), \text{argmax}_{\mathbf{y} \in \mathcal{Y}} f(\mathbf{y}) + \sigma(\mathbf{y})$ have unique optimizers with probability one, for any $f$ in $\mathcal{F}, \mathcal{F}'$ respectively. Moreover, suppose $\mathbb{E}_{\sigma \sim P^1_{PRTB}} [\|\sigma\|_{\mathcal{F}}] = \mathbb{E}_{\sigma \sim P^2_{PRTB}} [\|\sigma\|_{\mathcal{F}'}] = \eta$ and predictions of both the players are $C\eta^{-1}$-stable w.r.t norms $\|\cdot\|_{\mathcal{F}}, \|\cdot\|_{\mathcal{F}'}$. Suppose the guesses*

used by $\mathbf{x}, \mathbf{y}$ *players in the $t^{th}$ iteration are* $f(\cdot, \tilde{Q}_{t-1}), f(\tilde{P}_{t-1}, \cdot)$, *where* $\tilde{P}_{t-1}, \tilde{Q}_{t-1}$ *denote the predictions of* $\mathbf{x}, \mathbf{y}$ *players in the $t^{th}$ iteration, if guess $g_t = 0$ was used. Then the iterates $\{(P_t, Q_t)\}_{t=1}^{T}$ generated by the Algorithm 3 satisfy the following, for $\eta > \sqrt{3}CL$*

$$\sup_{\mathbf{x} \in \mathcal{X}, \mathbf{y} \in \mathcal{Y}} \mathbb{E}\left[ f\left( \frac{1}{T} \sum_{t=1}^{T} P_t, \mathbf{y} \right) - f\left( \mathbf{x}, \frac{1}{T} \sum_{t=1}^{T} Q_t \right) \right] = O\left( \frac{\eta D}{T} + \frac{CD^2 L^2}{\eta m} \right)$$

$$+ O\left( \min\left\{ \frac{dC\Psi_1^2 \Psi_2^2 G^2 \log(2m)}{\eta m}, \frac{CD^2 L^2}{\eta} \right\} \right).$$

*Proof.* The proof of this Theorem uses similar arguments as Theorem D.1. Since both the players are responding to each others actions using OFTPL, using Theorem 4.2, we get the following regret bounds for the players

$$\sup_{\mathbf{x} \in \mathcal{X}} \mathbb{E}\left[ \sum_{t=1}^{T} f(P_t, Q_t) - f(\mathbf{x}, Q_t) \right] \leqslant \eta D + \sum_{t=1}^{T} \frac{C}{2\eta} \mathbb{E}\left[ \|f(\cdot, Q_t) - f(\cdot, \tilde{Q}_{t-1})\|_{\mathcal{F}}^2 \right]$$

$$- \frac{\eta}{2C} \sum_{t=1}^{T} \mathbb{E}\left[ \gamma_{\mathcal{F}}(P_t^\infty, \tilde{P}_{t-1}^\infty)^2 \right],$$

$$\sup_{\mathbf{y} \in \mathcal{Y}} \mathbb{E}\left[ \sum_{t=1}^{T} f(P_t, \mathbf{y}) - f(P_t, Q_t) \right] \leqslant \eta D + \sum_{t=1}^{T} \frac{C}{2\eta} \mathbb{E}\left[ \|f(P_t, \cdot) - f(\tilde{P}_{t-1}, \cdot)\|_{\mathcal{F}'}^2 \right]$$

$$- \frac{\eta}{2C} \sum_{t=1}^{T} \mathbb{E}\left[ \gamma_{\mathcal{F}'}(Q_t^\infty, \tilde{Q}_{t-1}^\infty)^2 \right],$$

where $P_t^\infty, \tilde{P}_{t-1}^\infty, Q_t^\infty, \tilde{Q}_{t-1}^\infty$ are as defined in Theorem 4.2. First, consider the regret of the $\mathbf{x}$ player. We upper bound $\|f(\cdot, Q_t) - f(\cdot, \tilde{Q}_{t-1})\|_{\mathcal{F}}^2$ as

$$\|f(\cdot, Q_t) - f(\cdot, \tilde{Q}_{t-1})\|_{\mathcal{F}}^2 \leqslant 3\|f(\cdot, Q_t) - f(\cdot, Q_t^\infty)\|_{\mathcal{F}}^2$$

$$+ 3\|f(\cdot, Q_t^\infty) - f(\cdot, \tilde{Q}_{t-1}^\infty)\|_{\mathcal{F}}^2$$

$$+ 3\|f(\cdot, \tilde{Q}_{t-1}^\infty) - f(\cdot, \tilde{Q}_{t-1})\|_{\mathcal{F}}^2.$$

We now show that $\mathbb{E}\left[ \|f(\cdot, Q_t) - f(\cdot, Q_t^\infty)\|_{\mathcal{F}}^2 | \tilde{P}_{t-1}, \tilde{Q}_{t-1}, P_{1:t-1}, Q_{1:t-1} \right]$ is $O(1/m)$. To simplify the notation, we let $\zeta_t = \{\tilde{P}_{t-1}, \tilde{Q}_{t-1}, P_{1:t-1}, Q_{1:t-1}\}$. Let $\mathcal{N}_\epsilon$ be the $\epsilon$-net of $\mathcal{X}$ w.r.t $\|\cdot\|$. Then

$$\|f(\cdot, Q_t) - f(\cdot, Q_t^\infty)\|_{\mathcal{F}} \stackrel{(a)}{=} \sup_{\mathbf{x} \in \mathcal{X}} \|\nabla_{\mathbf{x}} f(\mathbf{x}, Q_t) - \nabla_{\mathbf{x}} f(\mathbf{x}, Q_t^\infty)\|_*$$

$$\stackrel{(b)}{\leqslant} \sup_{\mathbf{x} \in \mathcal{N}_\epsilon} \|\nabla_{\mathbf{x}} f(\mathbf{x}, Q_t) - \nabla_{\mathbf{x}} f(\mathbf{x}, Q_t^\infty)\|_* + 2L\epsilon,$$

where $(a)$ follows from the definition of Lipschitz constant and $(b)$ follows from our smoothness assumption on $f$. Using this, we get

$$\mathbb{E}\left[ \|f(\cdot, Q_t) - f(\cdot, Q_t^\infty)\|_{\mathcal{F}}^2 | \zeta_t \right] \leqslant 2\mathbb{E}\left[ \sup_{\mathbf{x} \in \mathcal{N}_\epsilon} \|\nabla_{\mathbf{x}} f(\mathbf{x}, Q_t) - \nabla_{\mathbf{x}} f(\mathbf{x}, Q_t^\infty)\|_*^2 \Big| \zeta_t \right] + 8L^2 \epsilon^2,$$

Since $f$ is Lipschitz, $\|\nabla_{\mathbf{x}} f(\mathbf{x}, \mathbf{y})\|_*$ is bounded by $G$. So $\|\nabla_{\mathbf{x}} f(\mathbf{x}, Q_t) - \nabla_{\mathbf{x}} f(\mathbf{x}, Q_t^\infty)\|_*$ is bounded by $2G$ and $\|\nabla_{\mathbf{x}} f(\mathbf{x}, Q_t) - \nabla_{\mathbf{x}} f(\mathbf{x}, Q_t^\infty)\|_2$ is bounded by $2\Psi_1 G$. Moreover, conditioned on past randomness $(\zeta_t)$, $\nabla_{\mathbf{x}} f(\mathbf{x}, Q_t) - \nabla_{\mathbf{x}} f(\mathbf{x}, Q_t^\infty)$ is a sub-Gaussian random vector and satisfies the following bound

$$\mathbb{E}\left[ \langle \mathbf{u}, \nabla_{\mathbf{x}} f(\mathbf{x}, Q_t) - \nabla_{\mathbf{x}} f(\mathbf{x}, Q_t^\infty) \rangle | \zeta_t \right] \leqslant \exp\left( 2\Psi_1^2 G^2 \|\mathbf{u}\|_2^2 / m \right).$$

From tail bounds of sub-Gaussian random vectors [26], we have

$$\mathbb{P}\left( \|\nabla_{\mathbf{x}} f(\mathbf{x}, Q_t) - \nabla_{\mathbf{x}} f(\mathbf{x}, Q_t^\infty)\|_2^2 > \frac{4\Psi_1^2 G^2}{m} (d + 2\sqrt{ds} + 2s) \Big| \zeta_t \right) \leqslant e^{-s},$$

for any $s > 0$. Using union bound, and the fact that $\log |\mathcal{N}_\epsilon|$ is upper bounded by $d \log (1 + 2D/\epsilon)$, we get

$$\mathbb{P}\left(\sup_{\mathbf{x} \in \mathcal{N}_\epsilon} \|\nabla_{\mathbf{x}} f(\mathbf{x}, Q_t) - \nabla_{\mathbf{x}} f(\mathbf{x}, Q_t^\infty)\|_2^2 > \frac{4\Psi_1^2 G^2}{m}(d + 2\sqrt{ds} + 2s)\Big| \zeta_t\right) \leqslant e^{-s + d \log(1 + 2D/\epsilon)}.$$

Let $Z = \sup_{\mathbf{x} \in \mathcal{N}_\epsilon} \|\nabla_{\mathbf{x}} f(\mathbf{x}, Q_t) - \nabla_{\mathbf{x}} f(\mathbf{x}, Q_t^\infty)\|_2^2$. The expectation of $Z$ can be bounded as follows

$$\mathbb{E}\left[Z | \zeta_t\right] = \mathbb{P}(Z \leqslant a | \zeta_t) \mathbb{E}\left[Z | \zeta_t, Z \leqslant a\right] + \mathbb{P}(Z > a | \zeta_t) \mathbb{E}\left[Z | \zeta_t, Z > a\right]$$
$$\leqslant a + 4\Psi_1^2 G^2 \mathbb{P}(Z > a | \zeta_t).$$

Choosing $\epsilon = Dm^{-1/2}$, $s = 3d \log(1 + 2m^{1/2})$, and $a = \frac{44 d \Psi_1^2 G^2 \log(1 + 2m^{1/2})}{m}$, we get

$$\mathbb{E}\left[Z | \zeta_t\right] \leqslant \frac{48 d \Psi_1^2 G^2 \log(1 + 2m^{1/2})}{m}.$$

This shows that $\mathbb{E}\left[\|f(\cdot, Q_t) - f(\cdot, Q_t^\infty)\|_{\mathcal{F}}^2 | \zeta_t\right] \leqslant \frac{96 d \Psi_1^2 \Psi_2^2 G^2 \log(1 + 2m^{1/2})}{m} + \frac{8 D^2 L^2}{m}$. Note that another trivial upper bound for $\|f(\cdot, Q_t) - f(\cdot, Q_t^\infty)\|_{\mathcal{F}}$ is $DL$, which can obtained as follows

$$\|f(\cdot, Q_t) - f(\cdot, Q_t^\infty)\|_{\mathcal{F}} = \sup_{\mathbf{x} \in \mathcal{X}} \|\nabla_{\mathbf{x}} f(\mathbf{x}, Q_t) - \nabla_{\mathbf{x}} f(\mathbf{x}, Q_t^\infty)\|_*$$
$$= \|\mathbb{E}_{\mathbf{y}_1 \sim Q_t, \mathbf{y}_2 \sim Q_t^\infty}\left[\nabla_{\mathbf{x}} f(\mathbf{x}, \mathbf{y}_1) - \nabla_{\mathbf{x}} f(\mathbf{x}, \mathbf{y}_2)\right]\|_*$$
$$\overset{(a)}{\leqslant} LD,$$

where $(a)$ follows from the smoothness assumption on $f$ and the fact that the diameter of $\mathcal{X}$ is $D$. When $L$ is close to $0$, this bound can be much better than the above bound. So we have

$$\mathbb{E}\left[\|f(\cdot, Q_t) - f(\cdot, Q_t^\infty)\|_{\mathcal{F}}^2 | \zeta_t\right] \leqslant \min\left(\frac{96 d \Psi_1^2 \Psi_2^2 G^2 \log(1 + 2m^{1/2})}{m} + \frac{8 D^2 L^2}{m}, L^2 D^2\right).$$

Using this, the regret of the $\mathbf{x}$ player can be bounded as follows

$$\sup_{\mathbf{x} \in \mathcal{X}} \mathbb{E}\left[\sum_{t=1}^T f(P_t, Q_t) - f(\mathbf{x}, Q_t)\right] \leqslant \eta D + \frac{24 C D^2 L^2 T}{\eta m}$$
$$+ \min\left(\frac{288 d C \Psi_1^2 \Psi_2^2 G^2 T \log(1 + 2m^{1/2})}{\eta m}, \frac{3 C D^2 L^2 T}{\eta}\right)$$
$$+ \sum_{t=1}^T \frac{3C}{2\eta} \mathbb{E}\left[\|f(\cdot, Q_t^\infty) - f(\cdot, \tilde{Q}_{t-1}^\infty)\|_{\mathcal{F}}^2\right]$$
$$- \frac{\eta}{2C} \sum_{t=1}^T \mathbb{E}\left[\gamma_{\mathcal{F}}(P_t^\infty, \tilde{P}_{t-1}^\infty)^2\right].$$

A similar analysis shows that the regret of $\mathbf{y}$ player can be bounded as

$$\sup_{\mathbf{y} \in \mathcal{Y}} \mathbb{E}\left[\sum_{t=1}^T f(P_t, \mathbf{y}) - f(P_t, Q_t)\right] \leqslant \eta D + \frac{24 C D^2 L^2 T}{\eta m}$$
$$+ \min\left(\frac{288 d C \Psi_1^2 \Psi_2^2 G^2 T \log(1 + 2m^{1/2})}{\eta m}, \frac{3 C D^2 L^2 T}{\eta}\right)$$
$$+ \sum_{t=1}^T \frac{3C}{2\eta} \mathbb{E}\left[\|f(P_t^\infty, \cdot) - f(\tilde{P}_{t-1}^\infty, \cdot)\|_{\mathcal{F}'}^2\right]$$
$$- \frac{\eta}{2C} \sum_{t=1}^T \mathbb{E}\left[\gamma_{\mathcal{F}'}(Q_t^\infty, \tilde{Q}_{t-1}^\infty)^2\right],$$

Summing the above two inequalities, we get

$$\sup_{\mathbf{x}\in\mathcal{X},\mathbf{y}\in\mathcal{Y}}\mathbb{E}\left[\sum_{t=1}^{T}f(P_t,\mathbf{y})-f(P,Q_t)\right]\leqslant 2\eta D+\frac{48CD^2L^2T}{\eta m}$$

$$+\min\left(\frac{576dC\Psi_1^2\Psi_2^2G^2T\log(1+2m^{1/2})}{\eta m},\frac{6CD^2L^2T}{\eta}\right)$$

$$+\sum_{t=1}^{T}\frac{3C}{2\eta}\mathbb{E}\left[\|f(\cdot,Q_t^\infty)-f(\cdot,\tilde{Q}_{t-1}^\infty)\|_{\mathcal{F}}^2\right]$$

$$+\sum_{t=1}^{T}\frac{3C}{2\eta}\mathbb{E}\left[\|f(P_t^\infty,\cdot)-f(\tilde{P}_{t-1}^\infty,\cdot)\|_{\mathcal{F}'}^2\right]$$

$$-\frac{\eta}{2C}\sum_{t=1}^{T}\left(\mathbb{E}\left[\gamma_{\mathcal{F}}(P_t^\infty,\tilde{P}_{t-1}^\infty)^2\right]+\mathbb{E}\left[\gamma_{\mathcal{F}'}(Q_t^\infty,\tilde{Q}_{t-1}^\infty)^2\right]\right).$$

From our assumption on smoothness of $f$, we have

$$\|f(\cdot,Q_t^\infty)-f(\cdot,\tilde{Q}_{t-1}^\infty)\|_{\mathcal{F}}\leqslant L\gamma_{\mathcal{F}'}(Q_t^\infty,\tilde{Q}_{t-1}^\infty),\quad \|f(P_t^\infty,\cdot)-f(\tilde{P}_{t-1}^\infty,\cdot)\|_{\mathcal{F}'}\leqslant L\gamma_{\mathcal{F}}(P_t^\infty,\tilde{P}_{t-1}^\infty).$$

To see this, consider the following

$$\|f(\cdot,Q_t^\infty)-f(\cdot,\tilde{Q}_{t-1}^\infty)\|_{\mathcal{F}}=\sup_{\mathbf{x}\in\mathcal{X}}\|\nabla_{\mathbf{x}}f(\mathbf{x},Q_t^\infty)-\nabla_{\mathbf{x}}f(\mathbf{x},\tilde{Q}_{t-1}^\infty)\|_*$$

$$=\sup_{\mathbf{x}\in\mathcal{X},\|\mathbf{u}\|\leqslant 1}\left\langle\mathbf{u},\nabla_{\mathbf{x}}f(\mathbf{x},Q_t^\infty)-\nabla_{\mathbf{x}}f(\mathbf{x},\tilde{Q}_{t-1}^\infty)\right\rangle$$

$$=\sup_{\mathbf{x}\in\mathcal{X},\|\mathbf{u}\|\leqslant 1}\mathbb{E}_{\mathbf{y}\sim Q_t^\infty}\left[\langle\mathbf{u},\nabla_{\mathbf{x}}f(\mathbf{x},\mathbf{y})\rangle\right]-\mathbb{E}_{\mathbf{y}\sim\tilde{Q}_{t-1}^\infty}\left[\langle\mathbf{u},\nabla_{\mathbf{x}}f(\mathbf{x},\mathbf{y})\rangle\right]$$

$$\leqslant\gamma_{\mathcal{F}'}(Q_t^\infty,\tilde{Q}_{t-1}^\infty)\sup_{\mathbf{x}\in\mathcal{X},\|\mathbf{u}\|\leqslant 1}\|\langle\mathbf{u},\nabla_{\mathbf{x}}f(\mathbf{x},\cdot)\rangle\|_{\mathcal{F}'}$$

$$=\gamma_{\mathcal{F}'}(Q_t^\infty,\tilde{Q}_{t-1}^\infty)\sup_{\mathbf{x}\in\mathcal{X},\|\mathbf{u}\|\leqslant 1}\left(\sup_{\mathbf{y}_1\neq\mathbf{y}_2\in\mathcal{Y}}\frac{|\langle\mathbf{u},\nabla_{\mathbf{x}}f(\mathbf{x},\mathbf{y}_1)\rangle-\langle\mathbf{u},\nabla_{\mathbf{x}}f(\mathbf{x},\mathbf{y}_2)\rangle|}{\|\mathbf{y}_1-\mathbf{y}_2\|}\right)$$

$$\leqslant\gamma_{\mathcal{F}'}(Q_t^\infty,\tilde{Q}_{t-1}^\infty)\sup_{\mathbf{x}\in\mathcal{X}}\left(\sup_{\mathbf{y}_1\neq\mathbf{y}_2\in\mathcal{Y}}\frac{\|\nabla_{\mathbf{x}}f(\mathbf{x},\mathbf{y}_1)-\nabla_{\mathbf{x}}f(\mathbf{x},\mathbf{y}_2)\|_*}{\|\mathbf{y}_1-\mathbf{y}_2\|}\right)$$

$$\overset{(a)}{\leqslant}L\gamma_{\mathcal{F}'}(Q_t^\infty,\tilde{Q}_{t-1}^\infty),$$

where $(a)$ follows from smoothness of $f$. Substituting this in the previous equation, and choosing $\eta>\sqrt{3}CL$, we get

$$\sup_{\mathbf{x}\in\mathcal{X},\mathbf{y}\in\mathcal{Y}}\mathbb{E}\left[\sum_{t=1}^{T}f(P_t,\mathbf{y})-f(P,Q_t)\right]\leqslant 2\eta D+\frac{48CD^2L^2T}{\eta m}$$

$$+\min\left(\frac{576dC\Psi_1^2\Psi_2^2G^2T\log(1+2m^{1/2})}{\eta m},\frac{6CD^2L^2T}{\eta}\right)$$

This finishes the proof of the Theorem. $\qquad\square$

**Remark E.1.** *We note that a similar result can be obtained for other choice of function classes such as the set of all bounded and Lipschitz functions. The only difference between proving such a result vs. proving Theorem E.1 is in bounding $\|f(\cdot,Q_t)-f(\cdot,Q_t^\infty)\|_{\mathcal{F}}$.*

### E.3 Proof of Theorem 5.2

To prove the Theorem, we instantiate Theorem E.1 for exponential noise distribution. Recall, in Corollary 4.2, we showed that $\mathbb{E}_\sigma\left[\|\sigma\|_{\mathcal{F}}\right]=\eta\log d$ and OFTPL is $O\left(d^2D\eta^{-1}\right)$ stable w.r.t $\|\cdot\|_{\mathcal{F}}$,

for this choice of perturbation distribution (similar results hold for $(\mathcal{F}', \|\cdot\|_{\mathcal{F}'})$). Substituting this in the bounds of Theorem E.1 and using the fact that $\Psi_1 = \sqrt{d}, \Psi_2 = 1$, we get

$$\sup_{\mathbf{x}\in\mathcal{X},\mathbf{y}\in\mathcal{Y}} \mathbb{E}\left[f\left(\frac{1}{T}\sum_{t=1}^{T} P_t, \mathbf{y}\right) - f\left(\mathbf{x}, \frac{1}{T}\sum_{t=1}^{T} Q_t\right)\right] = O\left(\frac{\eta D \log d}{T} + \frac{d^2 D^3 L^2}{\eta m}\right)$$
$$+ O\left(\min\left\{\frac{d^4 D G^2 \log(2m)}{\eta m}, \frac{d^2 D^3 L^2}{\eta}\right\}\right).$$

Choosing $\eta = 10d^2 D(L+1), m = T$, we get

$$\sup_{\mathbf{x}\in\mathcal{X},\mathbf{y}\in\mathcal{Y}} \mathbb{E}\left[f\left(\frac{1}{T}\sum_{t=1}^{T} P_t, \mathbf{y}\right) - f\left(\mathbf{x}, \frac{1}{T}\sum_{t=1}^{T} Q_t\right)\right] = O\left(\frac{d^2 D^2 (L+1) \log d}{T}\right)$$
$$+ O\left(\min\left\{\frac{d^2 G^2 \log(T)}{LT}, D^2 L\right\}\right).$$

### E.4  Regret Bounds of OFTPL in [11]

In this section, we rely on the OFTPL regret bounds of [11] to derive the rate of convergence of Algorithm 4 to a NE of nonconvex-nonconcave games. Our goal is to show that even if we set $m = \infty$, the OFTPL regret bounds of [11] only give us $O\left(T^{-3/4}\right)$ convergence rates. In contrast, using the tighter regret bounds in Theorem 4.2, we obtain $O\left(T^{-1}\right)$ convergence rates.

**Theorem E.2.** *Consider the setting of Theorem 5.2. Suppose OFTPL is run for $T$ iterations with $\eta = (d^3 DGL)^{1/2} T^{1/4}, m = \infty$. then the iterates $\{(P_t, Q_t)\}_{t=1}^{T}$ satisfy*

$$\sup_{\mathbf{x}\in\mathcal{X},\mathbf{y}\in\mathcal{Y}} f\left(\frac{1}{T}\sum_{t=1}^{T} P_t, \mathbf{y}\right) - f\left(\mathbf{x}, \frac{1}{T}\sum_{t=1}^{T} Q_t\right) = O\left(\frac{(d^3 D^3 GL)^{1/2} \log d}{T^{3/4}}\right).$$

*Proof.* From the regret bounds of OFTPL derived by [11] we know that the regret of $\mathbf{x}$ player can be bounded as

$$\sup_{\mathbf{x}\in\mathcal{X}} \sum_{t=1}^{T} f(P_t, Q_t) - f(\mathbf{x}, Q_t) \leqslant O\left(\eta D \log d + \sum_{t=1}^{T} \frac{d^2 D}{\eta} \|f(\cdot, Q_t) - f(\cdot, \tilde{Q}_{t-1})\|_{\mathcal{F}}^2\right),$$

where $\|f(\cdot, Q_t) - f(\cdot, \tilde{Q}_{t-1})\|_{\mathcal{F}}$ is the Lipscthiz constant of $f(\cdot, Q_t) - f(\cdot, \tilde{Q}_{t-1})$, which can be upper bounded as follows

$$\|f(\cdot, Q_t) - f(\cdot, \tilde{Q}_{t-1})\|_{\mathcal{F}} \leqslant L\mathbb{E}_\sigma\left[\|\mathbf{y}_t(\sigma) - \tilde{\mathbf{y}}_{t-1}(\sigma)\|_1\right] \quad \text{(smoothness of } F)$$
$$\leqslant O\left(\frac{LGd^2 D}{\eta}\right) \quad \text{(stability)},$$

where the last inequality follows from the stability of FTPL proved in Theorem 1 of Suggala and Netrapalli [11] . Plugging this in the regret bound of the $\mathbf{x}$ player, we get

$$\sup_{\mathbf{x}\in\mathcal{X}} \left(\sum_{t=1}^{T} f(P_t, Q_t) - f(\mathbf{x}, Q_t)\right) \leqslant O\left(\frac{d^6 D^3 G^2 L^2 T}{\eta^3} + \eta D \log d\right).$$

A similar argument shows that the regret of $\mathbf{y}$ player can be bounded as

$$\sup_{\mathbf{y}\in\mathcal{Y}} \left(\sum_{t=1}^{T} f(P_t, \mathbf{y}) - f(P_t, Q_t)\right) \leqslant O\left(\frac{d^6 D^3 G^2 L^2 T}{\eta^3} + \eta D \log d\right).$$

Summing the above two regret bounds and substituting $\eta = (d^3 DGL)^{1/2} T^{1/4}$ gives us the required bound. $\qquad\square$

# F Choice of Perturbation Distributions

**Regularization of some Perturbation Distributions.** We first study the regularization effect of various perturbation distributions. Table 2 presents the regularizer $R$ corresponding to some commonly used perturbation distributions, when the action space $\mathcal{X}$ is $\ell_\infty$ ball of radius 1 centered at origin.

| Perturbation Distribution $P_{\text{PRTB}}$ | Regularizer |
|---|---|
| Uniform over $[0, \eta]^d$ | $\eta\|\mathbf{x} - 1\|_2^2$ |
| Exponential $P(\sigma > t) = \exp(-t/\eta)$ | $\sum_i \eta(\mathbf{x}_i + 1)\left[\log(\mathbf{x}_i + 1) - (1 + \log 2)\right]$ |
| Gaussian $P(\sigma = t) \propto e^{-t^2/2\eta^2}$ | $\sum_i \sup_{u \in \mathbb{R}} u\left[\mathbf{x}_i - 1 + 2F(-u/\eta)\right]$ |

Table 2: Regularizers corresponding to various perturbation distributions used in FTPL when the action space $\mathcal{X}$ is $\ell_\infty$ ball of radius 1 centered at origin. Here, $F$ is the CDF of a standard normal random variable.

**Dimension independent rates.** Recall, the OFTPL algorithm described in Algorithm 3 converges at $O\left(d/T\right)$ rate to a Nash equilibrium of smooth convex-concave games (see Theorem 5.1). We now show that for certain constraint sets $\mathcal{X}, \mathcal{Y}$, by choosing the perturbation distributions appropriately, the dimension dependence in the rates can *potentially* be removed.

Suppose the action set is $\mathcal{X} = \{\mathbf{x} : \|\mathbf{x}\|_2 \leqslant 1\}$. Suppose the perturbation distribution $P_{\text{PRTB}}$ is the multivariate Gaussian distribution with mean $0$ and covariance $\eta^2 I_{d \times d}$, where $I_{d \times d}$ is the identity matrix. We now try to explicitly compute the regularizer corresponding to this perturbation distribution and action set. Define function $\Psi$ as

$$\Psi(f) = \mathbb{E}_\sigma\left[\max_{\mathbf{x} \in \mathcal{X}} \langle f + \sigma, \mathbf{x}\rangle\right] = \mathbb{E}_\sigma\left[\|f + \sigma\|_2\right].$$

As shown in Proposition 3.1, the regularizer $R$ corresponding to any perturbation distribution is given by the Fenchel conjugate of $\Psi$

$$R(\mathbf{x}) = \sup_f \langle f, \mathbf{x}\rangle - \Psi(f).$$

Since getting an exact expression for $R$ is a non-trivial task, we only compute an *approximate expression* for $R$. Consider the high dimensional setting (*i.e.,* very large $d$). In this setting, $\|f + \sigma\|_2$, for $\sigma$ drawn from $\mathcal{N}(0, \eta^2 I_{d \times d})$, can be approximated as follows

$$\|f + \sigma\|_2 = \sqrt{\|f\|_2^2 + \|\sigma\|_2^2 + 2\langle f, \sigma\rangle}$$
$$\overset{(a)}{\approx} \sqrt{\|f\|_2^2 + \eta^2 d + 2\langle f, \sigma\rangle}$$
$$\overset{(b)}{\approx} \sqrt{\|f\|_2^2 + \eta^2 d}$$

where $(a)$ follows from the fact that $\|\sigma\|_2^2$ is highly concentrated around $\eta^2 d$ [26]. To be precise

$$\mathbb{P}(\|\sigma\|_2^2 \geqslant \eta^2(d + 2\sqrt{dt} + 2t)) \leqslant e^{-t}.$$

A similar bound holds for the lower tail. Approximation $(b)$ follows from the fact that $\langle f, \sigma\rangle$ is a Gaussian random variable with mean $0$ and variance $\eta^2\|f\|_2^2$, and with high probability its magnitude is upper bounded by $\tilde{O}(\eta\|f\|_2)$. Since $\eta\|f\|_2 \ll \sqrt{d}\eta\|f\|_2 \leqslant \|f\|_2^2 + \eta^2 d$, approximation $(b)$ holds. This shows that $\Psi(f)$ can be approximated as

$$\Psi(f) \approx \sqrt{\|f\|_2^2 + \eta^2 d}.$$

Using this approximation, we now compute the regularizer corresponding to the perturbation distribution

$$R(\mathbf{x}) = \sup_f \langle f, \mathbf{x}\rangle - \Psi(f) \approx \sup_f \langle f, \mathbf{x}\rangle - \sqrt{\|f\|_2^2 + \eta^2 d} = -\eta\sqrt{d}\sqrt{1 - \|\mathbf{x}\|_2^2}.$$

This shows that $R$ is $\eta\sqrt{d}$-strongly convex w.r.t $\|\cdot\|_2$ norm. Following duality between strong convexity and strong smoothness, $\Psi(f)$ is $(\eta^2 d)^{-1/2}$ strongly smooth w.r.t $\|\cdot\|_2$ norm and satisfies

$$\|\nabla\Psi(f_1) - \nabla\Psi(f_2)\|_2 \leqslant (\eta^2 d)^{-1/2}\|f_1 - f_2\|_2.$$

This shows that the predictions of OFTPL are $(\eta^2 d)^{-1/2}$ stable w.r.t $\|\cdot\|_2$ norm. We now instantiate Theorem D.1 for this perturbation distribution and for constraint sets which are unit balls centered at origin, and use the above stability bound, together with the fact that $\mathbb{E}_\sigma\left[\|\sigma\|_2\right] \approx \eta\sqrt{d}$. Suppose $f$ is smooth w.r.t $\|\cdot\|_2$ norm and satisfies

$$\|\nabla_{\mathbf{x}}f(\mathbf{x},\mathbf{y}) - \nabla_{\mathbf{x}}f(\mathbf{x}',\mathbf{y}')\|_2 + \|\nabla_{\mathbf{y}}f(\mathbf{x},\mathbf{y}) - \nabla_{\mathbf{y}}f(\mathbf{x}',\mathbf{y}')\|_2 \leqslant L\|\mathbf{x} - \mathbf{x}'\|_2 + L\|\mathbf{y} - \mathbf{y}'\|_2.$$

Then Theorem D.1 gives us the following rates of convergence to a NE

$$\sup_{\mathbf{x}\in\mathcal{X},\mathbf{y}\in\mathcal{Y}} \mathbb{E}\left[f\left(\frac{1}{T}\sum_{t=1}^{T}\mathbf{x}_t,\mathbf{y}\right) - f\left(\mathbf{x},\frac{1}{T}\sum_{t=1}^{T}\mathbf{y}_t\right)\right] \leqslant \frac{2L_1}{m} + \frac{2\eta\sqrt{d}}{T}$$
$$+ \frac{20L^2}{\eta\sqrt{d}}\left(\frac{1}{m}\right) + 10L\left(\frac{5L}{\eta\sqrt{d}}\right)^\infty$$

Choosing $\eta = 6L/\sqrt{d}, m = T$, we get $O\left(\frac{L}{T}\right)$ rate of convergence. Although, these rates are dimension independent, we note that our stability bound is only approximate. More accurate analysis is needed to actually claim that Algorithm 3 achieves dimension independent rates in this setting. That being said, for general constraints sets, we believe one can get dimension independent rates by choosing the perturbation distribution appropriately.

# G High Probability Bounds

In this section, we provide high probability bounds for Theorems 4.1, 5.1. Our results rely on the following concentration inequalities.

**Proposition G.1** (Jin et al. [27]). *Let $X_1,\ldots X_K$ be $K$ independent mean $0$ vector-valued random variables such that $\|X_i\|_2 \leqslant B_i$. Then*

$$\mathbb{P}\left(\|\sum_{i=1}^{K} X_i\|_2 \geqslant t\right) \leqslant 2\exp\left(-c\frac{t^2}{\sum_{i=1}^{K} B_i^2}\right),$$

*where $c > 0$ is a universal constant.*

We also need the following concentration inequality for martingales.

**Proposition G.2** (Wainwright [28]). *Let $X_1,\ldots X_K \in \mathbb{R}$ be a martingale difference sequence, where $\mathbb{E}\left[X_i|\mathcal{F}_{i-1}\right] = 0$. Assume that $X_i$ satisfy the following tail condition, for some scalar $B_i > 0$*

$$\mathbb{P}\left(\left|\frac{X_i}{B_i}\right| \geqslant z\Big|\mathcal{F}_{i-1}\right) \leqslant 2\exp(-z^2).$$

*Then*

$$\mathbb{P}\left(\left|\sum_{i=1}^{K} X_i\right| \geqslant z\right) \leqslant 2\exp\left(-c\frac{z^2}{\sum_{i=1}^{K} B_i^2}\right),$$

*where $c > 0$ is a universal constant.*

## G.1 Online Convex Learning

In this section, we present a high probability version of Theorem 4.1.

**Theorem G.1.** *Suppose the perturbation distribution $P_{PRTB}$ is absolutely continuous w.r.t Lebesgue measure. Let $D$ be the diameter of $\mathcal{X}$ w.r.t $\|\cdot\|$, which is defined as $D = \sup_{\mathbf{x}_1,\mathbf{x}_2\in\mathcal{X}}\|\mathbf{x}_1 - \mathbf{x}_2\|$. Let $\eta = \mathbb{E}_\sigma\left[\|\sigma\|_*\right]$, and suppose the predictions of OFTPL are $C\eta^{-1}$-stable w.r.t $\|\cdot\|_*$, where $C$ is a constant that depends on the set $\mathcal{X}$. Suppose, the sequence of loss functions $\{f_t\}_{t=1}^{T}$ are $G$-Lipschitz*

*w.r.t* $\|\cdot\|$ *and satisfy* $\sup_{\mathbf{x}\in\mathcal{X}}\|\nabla f_t(\mathbf{x})\|_* \leqslant G$. *Moreover, suppose* $\{f_t\}_{t=1}^T$ *are Holder smooth and satisfy*

$$\forall \mathbf{x}_1, \mathbf{x}_2 \in \mathcal{X} \quad \|\nabla f_t(\mathbf{x}_1) - \nabla f_t(\mathbf{x}_2)\|_* \leqslant L\|\mathbf{x}_1 - \mathbf{x}_2\|^\alpha,$$

*for some constant* $\alpha \in [0,1]$. *Then the regret of Algorithm 1 satisfies the following with probability at least* $1 - \delta$

$$\sup_{\mathbf{x}\in\mathcal{X}} \sum_{t=1}^T f_t(\mathbf{x}_t) - f_t(\mathbf{x}) \leqslant \eta D + \sum_{t=1}^T \frac{C}{2\eta}\|\nabla_t - g_t\|_*^2 - \sum_{t=1}^T \frac{\eta}{2C}\|\mathbf{x}_t^\infty - \tilde{\mathbf{x}}_{t-1}^\infty\|^2$$

$$+ cGD\sqrt{\frac{T\log 2/\delta}{m}} + cLT\left(\frac{\Psi_1^2\Psi_2^2 D^2 \log 4T/\delta}{m}\right)^{\frac{1+\alpha}{2}},$$

*where* $c$ *is a universal constant,* $\mathbf{x}_t^\infty = \mathbb{E}\left[\mathbf{x}_t | g_t, f_{1:t-1}, \mathbf{x}_{1:t-1}\right]$ *and* $\tilde{\mathbf{x}}_{t-1}^\infty = \mathbb{E}\left[\tilde{\mathbf{x}}_{t-1} | f_{1:t-1}, \mathbf{x}_{1:t-1}\right]$ *and* $\tilde{\mathbf{x}}_{t-1}$ *denotes the prediction in the* $t^{th}$ *iteration of Algorithm 1, if guess* $g_t = 0$ *was used. Here,* $\Psi_1, \Psi_2$ *denote the norm compatibility constants of* $\|\cdot\|$.

*Proof.* Our proof uses the same notation and similar arguments as in the proof Theorem 4.1. Recall, in Theorem 4.1 we showed that the regret of OFTPL is upper bounded by

$$\sum_{t=1}^T f_t(\mathbf{x}_t) - f_t(\mathbf{x}) \leqslant \sum_{t=1}^T \langle \mathbf{x}_t - \mathbf{x}_t^\infty, \nabla_t \rangle + \eta D + \sum_{t=1}^T \|\mathbf{x}_t^\infty - \tilde{\mathbf{x}}_t^\infty\|\|\nabla_t - g_t\|_*$$

$$- \frac{\eta}{2C}\sum_{t=1}^T \left(\|\tilde{\mathbf{x}}_t^\infty - \mathbf{x}_t^\infty\|^2 + \|\mathbf{x}_t^\infty - \tilde{\mathbf{x}}_{t-1}^\infty\|^2\right)$$

$$\leqslant \sum_{t=1}^T \langle \mathbf{x}_t - \mathbf{x}_t^\infty, \nabla_t \rangle + \eta D + \sum_{t=1}^T \frac{C}{2\eta}\|\nabla_t - g_t\|_*^2 - \sum_{t=1}^T \frac{\eta}{2C}\|\mathbf{x}_t^\infty - \tilde{\mathbf{x}}_{t-1}^\infty\|^2.$$

From Holder's smoothness assumption, we have

$$\langle \mathbf{x}_t - \mathbf{x}_t^\infty, \nabla_t - \nabla f_t(\mathbf{x}_t^\infty) \rangle \leqslant L\|\mathbf{x}_t - \mathbf{x}_t^\infty\|^{1+\alpha}.$$

Substituting this in the previous bound gives us

$$\sum_{t=1}^T f_t(\mathbf{x}_t) - f_t(\mathbf{x}) \leqslant \underbrace{\sum_{t=1}^T \langle \mathbf{x}_t - \mathbf{x}_t^\infty, \nabla f_t(\mathbf{x}_t^\infty) \rangle}_{S_1} + \sum_{t=1}^T L\underbrace{\|\mathbf{x}_t - \mathbf{x}_t^\infty\|^{1+\alpha}}_{S_2} + \eta D$$

$$+ \sum_{t=1}^T \frac{C}{2\eta}\|\nabla_t - g_t\|_*^2 - \sum_{t=1}^T \frac{\eta}{2C}\|\mathbf{x}_t^\infty - \tilde{\mathbf{x}}_{t-1}^\infty\|^2.$$

We now provide high probability bounds for $S_1$ and $S_2$.

**Bounding** $S_1$. Let $\xi_i = \{g_{i+1}, f_{i+1}, \mathbf{x}_i\}$ and let $\xi_{0:t}$ denote the union of sets $\xi_0, \xi_1, \dots, \xi_t$. Let $\zeta_t = \langle \mathbf{x}_t - \mathbf{x}_t^\infty, \nabla f_t(\mathbf{x}_t^\infty) \rangle$ with $\zeta_0 = 0$. Note that $\{\zeta_t\}_{t=0}^T$ is a martingale difference sequence w.r.t $\xi_{0:T}$. This is because $\mathbb{E}\left[\mathbf{x}_t | \xi_{0:t-1}\right] = \mathbf{x}_t^\infty$ and $\nabla f_t(\mathbf{x}_t^\infty)$ is a deterministic quantity conditioned on $\xi_{0:t-1}$. As a result $\mathbb{E}\left[\zeta_t | \xi_{0:t-1}\right] = 0$. Moreover, conditioned on $\xi_{0:t-1}$, $\zeta_t$ is the average of $m$ independent mean 0 random variables, each of which is bounded by $GD$. Using Proposition G.1, we get

$$\mathbb{P}\left(|\zeta_t| \geqslant s \Big| \xi_{0:t-1}\right) \leqslant 2\exp\left(-\frac{ms^2}{G^2 D^2}\right).$$

Using Proposition G.2 on the martingale difference sequence $\{\zeta_t\}_{t=0}^T$, we get

$$\mathbb{P}\left(\left|\sum_{t=1}^T \zeta_t\right| \geqslant s\right) \leqslant 2\exp\left(-c\frac{ms^2}{G^2 D^2 T}\right),$$

where $c > 0$ is a universal constant. This shows that with probability at least $1 - \delta/2$, $S_1$ is upper bounded by $O\left(\sqrt{\frac{G^2 D^2 T \log \frac{2}{\delta}}{m}}\right)$.

**Bounding $S_2$.** Conditioned on $\{g_t, f_{1:t-1}, \mathbf{x}_{1:t-1}\}$, $\mathbf{x}_t - \mathbf{x}_t^\infty$ is the average of $m$ independent mean $0$ random variables which are bounded by $D$ in $\|\cdot\|$ norm. From our definition of norm compatibility constant $\Psi_2$, this implies the random variables are bounded by $\Psi_2 D$ in $\|\cdot\|_2$. Using Proposition G.1, we get

$$\mathbb{P}\left(\|\mathbf{x}_t - \mathbf{x}_t^\infty\|_2 \geqslant \Psi_2 D\sqrt{\frac{c\log 4T/\delta}{m}}\,\Big|\, g_t, f_{1:t-1}, \mathbf{x}_{1:t-1}\right) \leqslant \frac{\delta}{2T}.$$

Since the above bound holds for any set of $\{g_t, f_{1:t}, \mathbf{x}_{1:t-1}\}$, the same tail bound also holds without the conditioning. This shows that

$$\mathbb{P}\left(\|\mathbf{x}_t - \mathbf{x}_t^\infty\|^{1+\alpha} \geqslant \left(\frac{c\Psi_1^2\Psi_2^2 D^2 \log 4T/\delta}{m}\right)^{\frac{1+\alpha}{2}}\right) \leqslant \frac{\delta}{2T},$$

where we converted back to $\|\cdot\|$ by introducing the norm compatibility constant $\Psi_1$.

**Bounding the regret.** Plugging the above high probability bounds for $S_1, S_2$ in the previous regret bound and using union bound, we get the following regret bound which holds with probability at least $1 - \delta$

$$\sum_{t=1}^T f_t(\mathbf{x}_t) - f_t(\mathbf{x}) \leqslant cGD\sqrt{\frac{T\log 2/\delta}{m}} + cLT\left(\frac{\Psi_1^2\Psi_2^2 D^2 \log 4T/\delta}{m}\right)^{\frac{1+\alpha}{2}} + \eta D$$

$$+ \sum_{t=1}^T \frac{C}{2\eta}\|\nabla_t - g_t\|_*^2 - \sum_{t=1}^T \frac{\eta}{2C}\|\mathbf{x}_t^\infty - \tilde{\mathbf{x}}_{t-1}^\infty\|^2,$$

where $c > 0$ is a universal constant. $\qquad\square$

## G.2  Convex-Concave Games

In this section, we present a high probability version of Theorem 5.1.

**Theorem G.2.** *Consider the minimax game in Equation (1). Suppose both the domains $\mathcal{X}, \mathcal{Y}$ are compact subsets of $\mathbb{R}^d$, with diameter $D = \max\{\sup_{\mathbf{x}_1, \mathbf{x}_2 \in \mathcal{X}} \|\mathbf{x}_1 - \mathbf{x}_2\|_2, \sup_{\mathbf{y}_1, \mathbf{y}_2 \in \mathcal{Y}} \|\mathbf{y}_1 - \mathbf{y}_2\|_2\}$. Suppose $f$ is convex in $\mathbf{x}$, concave in $\mathbf{y}$ and is Lipschitz w.r.t $\|\cdot\|_2$ and satisfies*

$$\max\left\{\sup_{\mathbf{x}\in\mathcal{X},\mathbf{y}\in\mathcal{Y}}\|\nabla_\mathbf{x} f(\mathbf{x},\mathbf{y})\|_2,\ \sup_{\mathbf{x}\in\mathcal{X},\mathbf{y}\in\mathcal{Y}}\|\nabla_\mathbf{y} f(\mathbf{x},\mathbf{y})\|_2\right\} \leqslant G.$$

*Moreover, suppose $f$ is smooth w.r.t $\|\cdot\|_2$*

$$\|\nabla_\mathbf{x} f(\mathbf{x},\mathbf{y}) - \nabla_\mathbf{x} f(\mathbf{x}',\mathbf{y}')\|_2 + \|\nabla_\mathbf{y} f(\mathbf{x},\mathbf{y}) - \nabla_\mathbf{y} f(\mathbf{x}',\mathbf{y}')\|_2 \leqslant L\|\mathbf{x}-\mathbf{x}'\|_2 + L\|\mathbf{y}-\mathbf{y}'\|_2.$$

*Suppose Algorithm 3 is used to solve the minimax game. Suppose the perturbation distributions used by both the players are the same and equal to the uniform distribution over $\{\mathbf{x} : \|\mathbf{x}\|_2 \leqslant (1+d^{-1})\eta\}$. Suppose the guesses used by $\mathbf{x}, \mathbf{y}$ players in the $t^{th}$ iteration are $\nabla_\mathbf{x} f(\tilde{\mathbf{x}}_{t-1}, \tilde{\mathbf{y}}_{t-1}), \nabla_\mathbf{y} f(\tilde{\mathbf{x}}_{t-1}, \tilde{\mathbf{y}}_{t-1})$, where $\tilde{\mathbf{x}}_{t-1}, \tilde{\mathbf{y}}_{t-1}$ denote the predictions of $\mathbf{x}, \mathbf{y}$ players in the $t^{th}$ iteration, if guess $g_t = 0$ was used. If Algorithm 3 is run with $\eta = 6dD(L+1), m = T$, then the iterates $\{(\mathbf{x}_t, \mathbf{y}_t)\}_{t=1}^T$ satisfy the following bound with probability at least $1 - \delta$*

$$\sup_{\mathbf{x}\in\mathcal{X},\mathbf{y}\in\mathcal{Y}}\left[f\left(\frac{1}{T}\sum_{t=1}^T \mathbf{x}_t, \mathbf{y}\right) - f\left(\mathbf{x}, \frac{1}{T}\sum_{t=1}^T \mathbf{y}_t\right)\right] = O\left(\frac{GD\sqrt{\log\frac{8}{\delta}}}{T} + \frac{D^2(L+1)\left(d + \log\frac{16T}{\delta}\right)}{T}\right).$$

*Proof.* We use the same notation and proof technique as Theorems D.1, 5.1. From Theorem 4.1 we know that the predictions of OFTPL are $dD\eta^{-1}$ stable w.r.t $\|\cdot\|_2$, for the particular perturbation distribution we consider here. We use this stability bound in our proof. From Theorem G.1, we have

the following regret bound for both the players, which holds with probability at least $1 - \delta/2$

$$\sup_{\mathbf{x} \in \mathcal{X}} \left[ \sum_{t=1}^{T} f(\mathbf{x}_t, \mathbf{y}_t) - f(\mathbf{x}, \mathbf{y}_t) \right] \leqslant cGD\sqrt{\frac{T \log 8/\delta}{m}} + cLT \left( \frac{D^2 \log 16T/\delta}{m} \right) + \eta D$$

$$+ \frac{dD}{2\eta} \sum_{t=1}^{T} \left[ \|\nabla_{\mathbf{x}} f(\mathbf{x}_t, \mathbf{y}_t) - \nabla_{\mathbf{x}} f(\tilde{\mathbf{x}}_{t-1}, \tilde{\mathbf{y}}_{t-1})\|_2^2 \right]$$

$$- \frac{\eta}{2dD} \sum_{t=1}^{T} \left[ \|\mathbf{x}_t^{\infty} - \tilde{\mathbf{x}}_{t-1}^{\infty}\|_2^2 \right].$$

$$\sup_{\mathbf{y} \in \mathcal{Y}} \left[ \sum_{t=1}^{T} f(\mathbf{x}_t, \mathbf{y}) - f(\mathbf{x}_t, \mathbf{y}_t) \right] \leqslant cGD\sqrt{\frac{T \log 8/\delta}{m}} + cLT \left( \frac{D^2 \log 16T/\delta}{m} \right) + \eta D$$

$$+ \frac{dD}{2\eta} \sum_{t=1}^{T} \left[ \|\nabla_{\mathbf{y}} f(\mathbf{x}_t, \mathbf{y}_t) - \nabla_{\mathbf{y}} f(\tilde{\mathbf{x}}_{t-1}, \tilde{\mathbf{y}}_{t-1})\|_2^2 \right]$$

$$- \frac{\eta}{2dD} \sum_{t=1}^{T} \left[ \|\mathbf{y}_t^{\infty} - \tilde{\mathbf{y}}_{t-1}^{\infty}\|_2^2 \right].$$

First, consider the regret of the $\mathbf{x}$ player. From the proof of Theorem D.1, we have

$$\|\nabla_{\mathbf{x}} f(\mathbf{x}_t, \mathbf{y}_t) - \nabla_{\mathbf{x}} f(\tilde{\mathbf{x}}_{t-1}, \tilde{\mathbf{y}}_{t-1})\|_2^2 \leqslant 5L^2 \|\mathbf{x}_t - \mathbf{x}_t^{\infty}\|_2^2 + 5L^2 \|\tilde{\mathbf{x}}_{t-1} - \tilde{\mathbf{x}}_{t-1}^{\infty}\|_2^2$$

$$+ 5L^2 \|\mathbf{y}_t - \mathbf{y}_t^{\infty}\|_2^2 + 5L^2 \|\tilde{\mathbf{y}}_{t-1} - \tilde{\mathbf{y}}_{t-1}^{\infty}\|_2^2$$

$$+ 5\|\nabla_{\mathbf{x}} f(\mathbf{x}_t^{\infty}, \mathbf{y}_t^{\infty}) - \nabla_{\mathbf{x}} f(\tilde{\mathbf{x}}_{t-1}^{\infty}, \tilde{\mathbf{y}}_{t-1}^{\infty})\|_2^2.$$

Moreover, from the proof of Theorem G.1, we know that $\|\mathbf{x}_t - \mathbf{x}_t^{\infty}\|_2^2$ satisfies the following tail bound

$$\mathbb{P}\left( \|\mathbf{x}_t - \mathbf{x}_t^{\infty}\|_2^2 \geqslant \frac{cD^2 \log 16T/\delta}{m} \right) \leqslant \frac{\delta}{8T}.$$

Similar bounds hold for the quantities appearing in the regret bound of $\mathbf{y}$ player. Plugging this in the previous regret bounds, we get the following which hold with probability at least $1 - \delta$

$$\sup_{\mathbf{x} \in \mathcal{X}} \left[ \sum_{t=1}^{T} f(\mathbf{x}_t, \mathbf{y}_t) - f(\mathbf{x}, \mathbf{y}_t) \right] \leqslant cGD\sqrt{\frac{T \log 8/\delta}{m}} + \left( L + \frac{10dDL^2}{\eta} \right) \left( \frac{cD^2 \log 16T/\delta}{m} \right) T$$

$$+ \eta D + \frac{5dD}{2\eta} \sum_{t=1}^{T} \left[ \|\nabla_{\mathbf{x}} f(\mathbf{x}_t^{\infty}, \mathbf{y}_t^{\infty}) - \nabla_{\mathbf{x}} f(\tilde{\mathbf{x}}_{t-1}^{\infty}, \tilde{\mathbf{y}}_{t-1}^{\infty})\|_2^2 \right]$$

$$- \frac{\eta}{2dD} \sum_{t=1}^{T} \left[ \|\mathbf{x}_t^{\infty} - \tilde{\mathbf{x}}_{t-1}^{\infty}\|_2^2 \right].$$

$$\sup_{\mathbf{y} \in \mathcal{Y}} \left[ \sum_{t=1}^{T} f(\mathbf{x}_t, \mathbf{y}) - f(\mathbf{x}_t, \mathbf{y}_t) \right] \leqslant cGD\sqrt{\frac{T \log 8/\delta}{m}} + \left( L + \frac{10dDL^2}{\eta} \right) \left( \frac{cD^2 \log 16T/\delta}{m} \right) T$$

$$+ \eta D + \frac{5dD}{2\eta} \sum_{t=1}^{T} \left[ \|\nabla_{\mathbf{y}} f(\mathbf{x}_t^{\infty}, \mathbf{y}_t^{\infty}) - \nabla_{\mathbf{y}} f(\tilde{\mathbf{x}}_{t-1}^{\infty}, \tilde{\mathbf{y}}_{t-1}^{\infty})\|_2^2 \right]$$

$$- \frac{\eta}{2dD} \sum_{t=1}^{T} \left[ \|\mathbf{y}_t^{\infty} - \tilde{\mathbf{y}}_{t-1}^{\infty}\|_2^2 \right].$$

Summing these two regret bounds, we get

$$\sup_{\mathbf{x}\in\mathcal{X},\mathbf{y}\in\mathcal{Y}}\left[\sum_{t=1}^{T}f(\mathbf{x}_t,\mathbf{y})-f(\mathbf{x},\mathbf{y}_t)\right]\leqslant 2cGD\sqrt{\frac{T\log 8/\delta}{m}}+\left(L+\frac{10dDL^2}{\eta}\right)\left(\frac{2cD^2\log 16T/\delta}{m}\right)T+2\eta D$$

$$+\frac{10dD}{2\eta}\sum_{t=1}^{T}\left[\|\nabla_{\mathbf{x}}f(\mathbf{x}_t^\infty,\mathbf{y}_t^\infty)-\nabla_{\mathbf{x}}f(\tilde{\mathbf{x}}_{t-1}^\infty,\tilde{\mathbf{y}}_{t-1}^\infty)\|_2^2\right]$$

$$+\frac{10dD}{2\eta}\sum_{t=1}^{T}\left[\|\nabla_{\mathbf{y}}f(\mathbf{x}_t^\infty,\mathbf{y}_t^\infty)-\nabla_{\mathbf{y}}f(\tilde{\mathbf{x}}_{t-1}^\infty,\tilde{\mathbf{y}}_{t-1}^\infty)\|_2^2\right]$$

$$-\frac{\eta}{2dD}\sum_{t=1}^{T}\left[\|\mathbf{x}_t^\infty-\tilde{\mathbf{x}}_{t-1}^\infty\|_2^2+\|\mathbf{y}_t^\infty-\tilde{\mathbf{y}}_{t-1}^\infty\|_2^2\right].$$

From Holder's smoothness assumption on $f$, we have

$$\|\nabla_{\mathbf{x}}f(\mathbf{x}_t^\infty,\mathbf{y}_t^\infty)-\nabla_{\mathbf{x}}f(\tilde{\mathbf{x}}_{t-1}^\infty,\tilde{\mathbf{y}}_{t-1}^\infty)\|_2^2\leqslant 2\|\nabla_{\mathbf{x}}f(\mathbf{x}_t^\infty,\mathbf{y}_t^\infty)-\nabla_{\mathbf{x}}f(\mathbf{x}_t^\infty,\tilde{\mathbf{y}}_{t-1}^\infty)\|_2^2$$
$$+2\|\nabla_{\mathbf{x}}f(\mathbf{x}_t^\infty,\tilde{\mathbf{y}}_{t-1}^\infty)-\nabla_{\mathbf{x}}f(\tilde{\mathbf{x}}_{t-1}^\infty,\tilde{\mathbf{y}}_{t-1}^\infty)\|_2^2$$
$$\leqslant 2L^2\|\mathbf{x}_t^\infty-\tilde{\mathbf{x}}_{t-1}^\infty\|_2^2+2L^2\|\mathbf{y}_t^\infty-\tilde{\mathbf{y}}_{t-1}^\infty\|_2^2,$$

Using a similar argument, we get

$$\|\nabla_{\mathbf{y}}f(\mathbf{x}_t^\infty,\mathbf{y}_t^\infty)-\nabla_{\mathbf{y}}f(\tilde{\mathbf{x}}_{t-1}^\infty,\tilde{\mathbf{y}}_{t-1}^\infty)\|_2^2\leqslant 2L^2\|\mathbf{x}_t^\infty-\tilde{\mathbf{x}}_{t-1}^\infty\|_2^2+2L^2\|\mathbf{y}_t^\infty-\tilde{\mathbf{y}}_{t-1}^\infty\|_2^2.$$

Plugging this in the previous bound, and setting $\eta=6dD(L+1), m=T$, we get the following bound which holds with probability at least $1-\delta$

$$\sup_{\mathbf{x}\in\mathcal{X},\mathbf{y}\in\mathcal{Y}}\left[\sum_{t=1}^{T}f(\mathbf{x}_t,\mathbf{y})-f(\mathbf{x},\mathbf{y}_t)\right]\leqslant O\left(GD\sqrt{\log\frac{8}{\delta}}+D^2(L+1)\left(d+\log\frac{16T}{\delta}\right)\right).$$

$\square$

### G.3 Nonconvex-Nonconcave Games

In this section, we present a high probability version of Theorem 5.2.

**Theorem G.3.** *Consider the minimax game in Equation (1). Suppose the domains $\mathcal{X},\mathcal{Y}$ are compact subsets of $\mathbb{R}^d$ with diameter $D=\max\{\sup_{\mathbf{x}_1,\mathbf{x}_2\in\mathcal{X}}\|\mathbf{x}_1-\mathbf{x}_2\|_1,\sup_{\mathbf{y}_1,\mathbf{y}_2\in\mathcal{Y}}\|\mathbf{y}_1-\mathbf{y}_2\|_1\}$. Suppose $f$ is Lipschitz w.r.t $\|\cdot\|_1$ and satisfies*

$$\max\left\{\sup_{\mathbf{x}\in\mathcal{X},\mathbf{y}\in\mathcal{Y}}\|\nabla_{\mathbf{x}}f(\mathbf{x},\mathbf{y})\|_\infty,\sup_{\mathbf{x}\in\mathcal{X},\mathbf{y}\in\mathcal{Y}}\|\nabla_{\mathbf{y}}f(\mathbf{x},\mathbf{y})\|_\infty\right\}\leqslant G.$$

*Moreover, suppose $f$ satisfies the following smoothness property*

$$\|\nabla_{\mathbf{x}}f(\mathbf{x},\mathbf{y})-\nabla_{\mathbf{x}}f(\mathbf{x}',\mathbf{y}')\|_\infty+\|\nabla_{\mathbf{y}}f(\mathbf{x},\mathbf{y})-\nabla_{\mathbf{y}}f(\mathbf{x}',\mathbf{y}')\|_\infty\leqslant L\|\mathbf{x}-\mathbf{x}'\|_1+L\|\mathbf{y}-\mathbf{y}'\|_1.$$

*Suppose both $\mathbf{x}$ and $\mathbf{y}$ players use Algorithm 4 to solve the game with linear perturbation functions $\sigma(\mathbf{z})=\langle\bar{\sigma},\mathbf{z}\rangle$, where $\bar{\sigma}\in\mathbb{R}^d$ is such that each of its entries is sampled independently from $Exp(\eta)$. Suppose the guesses used by $\mathbf{x}$ and $\mathbf{y}$ players in the $t^{th}$ iteration are $f(\cdot,\tilde{Q}_{t-1}),f(\tilde{P}_{t-1},\cdot)$, where $\tilde{P}_{t-1},\tilde{Q}_{t-1}$ denote the predictions of $\mathbf{x},\mathbf{y}$ players in the $t^{th}$ iteration, if guess $g_t=0$ was used. If Algorithm 4 is run with $\eta=10d^2D(L+1), m=T$, then the iterates $\{(P_t,Q_t)\}_{t=1}^T$ satisfy the following with probability at least $1-\delta$*

$$\sup_{\mathbf{x}\in\mathcal{X},\mathbf{y}\in\mathcal{Y}}\sum_{t=1}^{T}f(P_t,\mathbf{y})-f(\mathbf{x},Q_t)=O\left(\frac{d^2D^2(L+1)\log d}{T}+\frac{GD}{T}\sqrt{\log\frac{8}{\delta}}\right)$$

$$+O\left(\min\left\{D^2L,\frac{d^2G^2\log T+dG^2\log\frac{8}{\delta}}{LT}\right\}\right).$$

*Proof.* We use the same notation used in the proofs of Theorems 4.2, E.1. Let $\mathcal{F}, \mathcal{F}'$ be the set of Lipschitz functions over $\mathcal{X}, \mathcal{Y}$, and $\|g_1\|_{\mathcal{F}}, \|g_2\|_{\mathcal{F}'}$ be the Lipschitz constants of functions $g_1 : \mathcal{X} \to \mathbb{R}$, $g_2 : \mathcal{Y} \to \mathbb{R}$ w.r.t $\|\cdot\|_1$. Recall, in Corollary 4.2 we showed that for our choice of perturbation distribution, $\mathbb{E}_\sigma [\|\sigma\|_{\mathcal{F}}] = \eta \log d$ and OFTPL is $O\left(d^2 D \eta^{-1}\right)$ stable. We use this in our proof.

From Theorem 4.2, we know that the regret of $\mathbf{x}, \mathbf{y}$ players satisfy

$$\sum_{t=1}^{T} f(P_t, Q_t) - f(\mathbf{x}, Q_t) \leqslant \eta D \log d + \underbrace{\sum_{t=1}^{T} \langle P_t - P_t^\infty, f(\cdot, Q_t) \rangle}_{S_1}$$

$$+ \sum_{t=1}^{T} \frac{cd^2 D}{2\eta} \underbrace{\|f(\cdot, Q_t) - f(\cdot, \tilde{Q}_{t-1})\|_{\mathcal{F}}^2}_{S_2}$$

$$- \sum_{t=1}^{T} \frac{\eta}{2cd^2 D} \gamma_{\mathcal{F}}(P_t^\infty, \tilde{P}_{t-1}^\infty)^2$$

$$\sum_{t=1}^{T} f(P_t, \mathbf{y}) - f(P_t, Q_t) \leqslant \eta D \log d + \sum_{t=1}^{T} \langle Q_t - Q_t^\infty, f(P_t, \cdot) \rangle$$

$$+ \sum_{t=1}^{T} \frac{cd^2 D}{2\eta} \|f(P_t, \cdot) - f(\tilde{P}_{t-1}, \cdot)\|_{\mathcal{F}'}^2$$

$$- \sum_{t=1}^{T} \frac{\eta}{2cd^2 D} \gamma_{\mathcal{F}'}(Q_t^\infty, \tilde{Q}_{t-1}^\infty)^2,$$

where $c > 0$ is a positive constant. We now provide high probability bounds for $S_1, S_2$.

**Bounding $S_1$.** Let $\xi_i = \{\tilde{P}_i, \tilde{Q}_i, P_i, Q_{i+1}\}$ with $\xi_0 = \{Q_1\}$ and let $\xi_{0:t}$ denote the union of sets $\xi_0, \dots, \xi_t$. Let $\zeta_t = \langle P_t - P_t^\infty, f(\cdot, Q_t) \rangle$ with $\zeta_0 = 0$. Note that $\{\zeta_t\}_{t=0}^{T}$ is a martingale difference sequence w.r.t $\xi_{0:T}$. This is because $\mathbb{E}[P_t | \xi_{0:t-1}] = P_t^\infty$ and $f(\cdot, Q_t)$ is a deterministic quantity conditioned on $\xi_{0:t-1}$. As a result $\mathbb{E}[\zeta_t | \xi_{0:t-1}] = 0$. Moreover, conditioned on $\xi_{0:t-1}$, $\zeta_t$ is the average of $m$ independent mean 0 random variables, each of which is bounded by $2GD$. Using Proposition G.1, we get

$$\mathbb{P}\left(|\zeta_t| \geqslant s \Big| \xi_{0:t-1}\right) \leqslant 2\exp\left(-\frac{ms^2}{4G^2 D^2}\right).$$

Using Proposition G.2 on the martingale difference sequence $\{\zeta_t\}_{t=0}^{T}$, we get

$$\mathbb{P}\left(\Big|\sum_{t=1}^{T} \zeta_t\Big| \geqslant s\right) \leqslant 2\exp\left(-c\frac{ms^2}{G^2 D^2 T}\right),$$

where $c > 0$ is a universal constant. This shows that with probability at least $1 - \delta/8$, $S_1$ is upper bounded by $O\left(\sqrt{\frac{G^2 D^2 T \log \frac{8}{\delta}}{m}}\right)$.

**Bounding $S_2$.** We upper bound $S_2$ as

$$\|f(\cdot, Q_t) - f(\cdot, \tilde{Q}_{t-1})\|_{\mathcal{F}}^2 \leqslant 3\|f(\cdot, Q_t) - f(\cdot, Q_t^\infty)\|_{\mathcal{F}}^2$$
$$+ 3\|f(\cdot, Q_t^\infty) - f(\cdot, \tilde{Q}_{t-1}^\infty)\|_{\mathcal{F}}^2$$
$$+ 3\|f(\cdot, \tilde{Q}_{t-1}^\infty) - f(\cdot, \tilde{Q}_{t-1})\|_{\mathcal{F}}^2.$$

We first provide a high probability bound for $\|f(\cdot, Q_t) - f(\cdot, Q_t^\infty)\|_{\mathcal{F}}^2$. A trivial bound for this quantity is $L^2 D^2$, which can be obtained as follows

$$\|f(\cdot, Q_t) - f(\cdot, Q_t^\infty)\|_{\mathcal{F}} = \sup_{\mathbf{x} \in \mathcal{X}} \|\nabla_\mathbf{x} f(\mathbf{x}, Q_t) - \nabla_\mathbf{x} f(\mathbf{x}, Q_t^\infty)\|_\infty$$

$$= \|\mathbb{E}_{\mathbf{y}_1 \sim Q_t, \mathbf{y}_2 \sim Q_t^\infty} [\nabla_\mathbf{x} f(\mathbf{x}, \mathbf{y}_1) - \nabla_\mathbf{x} f(\mathbf{x}, \mathbf{y}_2)]\|_\infty$$

$$\overset{(a)}{\leqslant} LD,$$

where $(a)$ follows from the smoothness assumption on $f$ and the fact that the diameter of $\mathcal{X}$ is $D$. A better bound for this quantity can be obtained as follows. From proof of Theorem E.1, we have

$$\|f(\cdot, Q_t) - f(\cdot, Q_t^\infty)\|_{\mathcal{F}}^2 \leqslant 2 \sup_{\mathbf{x} \in \mathcal{N}_\epsilon} \|\nabla_{\mathbf{x}} f(\mathbf{x}, Q_t) - \nabla_{\mathbf{x}} f(\mathbf{x}, Q_t^\infty)\|_\infty^2 + 8L^2\epsilon^2.$$

where $\mathcal{N}_\epsilon$ be the $\epsilon$-net of $\mathcal{X}$ w.r.t $\|\cdot\|$. Recall, in the proof of Theorem E.1, we showed the following high probability bound for the RHS quantity

$$\mathbb{P}\left(\sup_{\mathbf{x} \in \mathcal{N}_\epsilon} \|\nabla_{\mathbf{x}} f(\mathbf{x}, Q_t) - \nabla_{\mathbf{x}} f(\mathbf{x}, Q_t^\infty)\|_2^2 > \frac{4dG^2}{m}(d + 2\sqrt{ds} + 2s)\right) \leqslant e^{-s + d\log(1 + 2D/\epsilon)}.$$

Choosing $\epsilon = Dm^{-1/2}, s = \log\frac{8}{\delta} + d\log(1 + 2m^{1/2})$, we get the following bound for $\sup_{\mathbf{x} \in \mathcal{N}_\epsilon} \|\nabla_{\mathbf{x}} f(\mathbf{x}, Q_t) - \nabla_{\mathbf{x}} f(\mathbf{x}, Q_t^\infty)\|_2^2$ which holds with probability at least $1 - \delta/8$

$$\sup_{\mathbf{x} \in \mathcal{N}_\epsilon} \|\nabla_{\mathbf{x}} f(\mathbf{x}, Q_t) - \nabla_{\mathbf{x}} f(\mathbf{x}, Q_t^\infty)\|_2^2 \leqslant \frac{20dG^2}{m}\left(\log\frac{8}{\delta} + d\log(1 + 2m^{1/2})\right).$$

Together with our trivial bound of $D^2L^2$, this gives us the following bound for $\|f(\cdot, Q_t) - f(\cdot, Q_t^\infty)\|_{\mathcal{F}}^2$, which holds with probability at least $1 - \delta/8$

$$\|f(\cdot, Q_t) - f(\cdot, Q_t^\infty)\|_{\mathcal{F}}^2 \leqslant \min\left(\frac{20dG^2}{m}\left(\log\frac{8}{\delta} + d\log(1 + 2m^{1/2})\right), D^2L^2\right) + \frac{8D^2L^2}{m}.$$

Next, we bound $\|f(\cdot, Q_t^\infty) - f(\cdot, \tilde{Q}_{t-1}^\infty)\|_{\mathcal{F}}^2$. From our smoothness assumption on $f$, we have

$$\|f(\cdot, Q_t^\infty) - f(\cdot, \tilde{Q}_{t-1}^\infty)\|_{\mathcal{F}} \leqslant L\gamma_{\mathcal{F}'}(Q_t^\infty, \tilde{Q}_{t-1}^\infty).$$

Combining the previous two results, we get the following upper bound for $S_2$ which holds with probability at least $1 - \delta/8$

$$\|f(\cdot, Q_t) - f(\cdot, \tilde{Q}_{t-1})\|_{\mathcal{F}}^2 \leqslant 3L^2\gamma_{\mathcal{F}'}(Q_t^\infty, \tilde{Q}_{t-1}^\infty)^2 + \frac{48D^2L^2}{m}$$

$$+ \min\left(\frac{120dG^2}{m}\left(\log\frac{8}{\delta} + d\log(1 + 2m^{1/2})\right), 6D^2L^2\right).$$

**Regret bound.** Substituting the above bounds for $S_1, S_2$ in the regret bound for $\mathbf{x}$ player gives us the following bound, which holds with probability at least $1 - \delta/2$

$$\sum_{t=1}^T f(P_t, Q_t) - f(\mathbf{x}, Q_t) \leqslant \eta D \log d + O\left(GD\sqrt{\frac{T\log\frac{8}{\delta}}{m}} + \frac{d^2D^3L^2T}{\eta m}\right)$$

$$+ O\left(\min\left(\frac{d^3DG^2T}{\eta m}\left(\log\frac{8}{\delta} + d\log(2m)\right), \frac{d^2D^3L^2T}{\eta}\right)\right)$$

$$+ \sum_{t=1}^T \frac{3cd^2DL^2}{2\eta}\gamma_{\mathcal{F}'}(Q_t^\infty, \tilde{Q}_{t-1}^\infty)^2 - \sum_{t=1}^T \frac{\eta}{2cd^2D}\gamma_{\mathcal{F}}(P_t^\infty, \tilde{P}_{t-1}^\infty)^2$$

Using a similar analysis, we get the following regret bound for the $\mathbf{y}$ player

$$\sum_{t=1}^T f(P_t, Q_t) - f(\mathbf{x}, Q_t) \leqslant \eta D \log d + O\left(GD\sqrt{\frac{T\log\frac{8}{\delta}}{m}} + \frac{d^2D^3L^2T}{\eta m}\right)$$

$$+ O\left(\min\left(\frac{d^3DG^2T}{\eta m}\left(\log\frac{8}{\delta} + d\log(2m)\right), \frac{d^2D^3L^2T}{\eta}\right)\right)$$

$$+ \sum_{t=1}^T \frac{3cd^2DL^2}{2\eta}\gamma_{\mathcal{F}}(P_t^\infty, \tilde{P}_{t-1}^\infty)^2 - \sum_{t=1}^T \frac{\eta}{2cd^2D}\gamma_{\mathcal{F}'}(Q_t^\infty, \tilde{Q}_{t-1}^\infty)^2$$

Choosing, $\eta = 10d^2D(L+1), m = T$, and adding the above two regret bounds, we get

$$\sup_{\mathbf{x} \in \mathcal{X}, \mathbf{y} \in \mathcal{Y}} \sum_{t=1}^T f(P_t, \mathbf{y}) - f(\mathbf{x}, Q_t) = O\left(d^2D^2(L+1)\log d + GD\sqrt{\log\frac{8}{\delta}}\right)$$

$$+ O\left(\min\left\{D^2LT, \frac{d^2G^2\log T}{L} + \frac{dG^2\log\frac{8}{\delta}}{L}\right\}\right).$$

$\square$

# H Background on Convex Analysis

**Fenchel Conjugate.** The Fenchel conjugate of a function $f$ is defined as

$$f^*(x^*) = \sup_x \langle x, x^* \rangle - f(x).$$

We now state some useful properties of Fenchel conjugates. These properties can be found in Rockafellar [22].

**Theorem H.1.** *Let $f$ be a proper convex function. The conjugate function $f^*$ is then a closed and proper convex function. Moreover, if $f$ is lower semi-continuous then $f^{**} = f$.*

**Theorem H.2.** *For any proper convex function $f$ and any vector $x$, the following conditions on a vector $x^*$ are equivalent to each other*

- $x^* \in \partial f(x)$

- $\langle z, x^* \rangle - f(z)$ *achieves its supremum in $z$ at $z = x$*

- $f(x) + f^*(x^*) = \langle x, x^* \rangle$

*If $(clf)(x) = f(x)$, the following condition can be added to the list*

- $x \in \partial f^*(x^*)$

**Theorem H.3.** *If $f$ is a closed proper convex function, $\partial f^*$ is the inverse of $\partial f$ in the sense of multivalued mappings, i.e., $x \in \partial f^*(x^*)$ iff $x^* \in \partial f(x)$.*

**Theorem H.4.** *Let $f$ be a closed proper convex function. Let $\partial f$ be the subdifferential mapping. The effective domain of $\partial f$, which is the set $dom(\partial f) = \{x | \partial f \neq 0\}$, satisfies*

$$ri(dom(f)) \subseteq dom(\partial f) \subseteq dom(f).$$

*The range of $\partial f$ is defined as $range \partial f = \cup \{\partial f(x) | x \in \mathbb{R}^d\}$. The range of $\partial f$ is the effective domain of $\partial f^*$, so*

$$ri(dom(f^*)) \subseteq range \partial f \subseteq dom(f^*).$$

**Strong Convexity and Smoothness.** We now define strong convexity and strong smoothness and show that these two properties are duals of each other.

**Definition H.1** (Strong Convexity). A function $f : \mathcal{X} \to \mathbb{R} \cup \{\infty\}$ is $\beta$-strongly convex w.r.t a norm $\| \cdot \|$ if for all $x, y \in ri(dom(f))$ and $\alpha \in (0, 1)$ we have

$$f(\alpha x + (1-\alpha)y) \leqslant \alpha f(x) + (1-\alpha)f(y) - \frac{1}{2}\beta\alpha(1-\alpha)\|x - y\|^2.$$

This definition of strong convexity is equivalent to the following condition on $f$ [see Lemma 13 of 24]

$$f(y) \geqslant f(x) + \langle g, y - x \rangle + \frac{1}{2}\beta\|y - x\|^2, \quad \text{for any } x, y \in ri(dom(f)), g \in \partial f(x)$$

**Definition H.2** (Strong Smoothness). A function $f : \mathcal{X} \to \mathbb{R} \cup \{\infty\}$ is $\beta$-strongly smooth w.r.t a norm $\| \cdot \|$ if $f$ is everywhere differentiable and if for all $x, y$ we have

$$f(y) \leqslant f(x) + \langle \nabla f(x), y - x \rangle + \frac{1}{2}\beta\|y - x\|^2.$$

**Theorem H.5** (Kakade et al. [29]). *Assume that $f$ is a proper closed and convex function. Suppose $f$ is $\beta$-strongly smooth w.r.t a norm $\| \cdot \|$. Then its conjugate $f^*$ satisfies the following for all $a, x$ with $u = \nabla f(x)$*

$$f^*(a + u) \geqslant f^*(u) + \langle x, a \rangle + \frac{1}{2\beta}\|a\|_*^2.$$

**Theorem H.6** (Kakade et al. [29]). *Assume that $f$ is a closed and convex function. Then $f$ is $\beta$-strongly convex w.r.t a norm $\| \cdot \|$ iff $f^*$ is $\frac{1}{\beta}$-strongly smooth w.r.t the dual norm $\| \cdot \|_*$.*