[Reviews · NeurIPS 2020]

Review 1

Summary and Contributions: - It is known in the literature that optimistic variants of FTRL algorithm can yield better bounds when the sequence of loss functions are predictable. Such results are relatively rare for FTPL. This paper proposes the optimistic variant of the FTPL algorithm, which in the worst case known optimal bounds, but has the potential to achieve better regret for predictable sequence of loss functions. Specifically, the bounds depend on the || g_t - \nabla_t ||_* where g_t is the estimate of the gradient for the next loss function and \nabla_t is the observed gradient. - The key idea behind achieving these regret guarantees lie in result that the perturbation in the FTPL algorithm can be viewed as a regularizer and then some known piggyback transformations of the regret guarantees in the FTRL analysis helps to achieve the desired regrets. - The authors provide regret guarantees for the OFTPL case for both the convex and nonconvex losses. They instantiate this generic result for the worst case analysis via treating the future estimate g_t=0 and achieve the optimal O(T^{\frac{1}{2}}) regret. - Finally, the authors use the above analysis to propose optimistic variant of the FTPL algorithm to solve the minimax problem for convex-concave and nonconvex-nonconcave settings. The proposed algorithm also enjoys parallelization which makes it efficient for usage in the large scale ML tasks such as GAN training, adversarial learning, etc. ------------------- Update post rebuttal I have read the author feedback and other reviews. I am keeping my original score.

Strengths: - OFTPL achieves optimal worst case bounds for both convex and nonconvex losses. For the sequence of losses which have some notion of predictability it can provide better regret bounds. - Proposition 3.1, which shows that for the online convex learning, for any general perturbation distributions, FTPL algorithm can be treated as an FTRL algorithm, i.e. perturbation in the FTPL can be viewed as some form of regularization. This view results in the bounding the regret for the FTPL algorithm in the convex losses easily. For the non-convex losses, the authors follow similar arguments except try to substitute fenchel duality with carefully analyzed arguments including monotonicity of the FTPL predictions. This results in the theorem 4.1 and 4.2. - The authors instantiate these results for the worst case with estimate g_t = 0, this results in known optimal bound for the convex case and tighter bound for the non-convex case. - The authors also show how OFTPL can result in better convergence bounds for the minimax cases. Parallelizable OFTPL for convex-concave and nonconvex-nonconcave minimax games. This achieves the an accuracy upto O(T^{- \frac{1}{2}} ) with upto T calls to the optimization oracle.

Weaknesses: - One thing missing in the analysis is instantiating the results for Theorem 4.1 and 4.2 for some simple to understand basic cases like i.i.d. losses. This would have given better insights into the OFTPL algorithm. - No discussion on the obtained minimax bounds for nonconvex-nonconcave losses (for example comparison with known results such as [18]) - The proof techniques for OFTPL regret in convex and nonconvex losses, follow a similar strategy with few modifications. This should have been explained in the main text rather than pushing everything to the supplementary.

Correctness: - There's some confusion on the parallelization front. The paper state that for minimax games, OFTPL can achieve accuracy upto O(T^{- \frac{1}{2}} ) with upto T calls to the optimization oracle. But it requires O(T^{\frac{1}{2}}) iterations and each iteration shoots up O(T^{\frac{1}{2}}) parallel calls to the optimization oracle. In order to achieve the said accuracy you have to run the algorithm for T iterations and so you'll be calling the oracle more than O(T) times. Did I miss something? - Besides in the Algorithms 3 and 4, there are m argmin/argmax calls. Does the big-O notation hide this constant?

Clarity: - The paper is well written when read along with the supplementary material. The high level idea for proof techniques should be moved to the main text ( since in both convex and nonconvex cases, the proofs end up creating some pseudo Bregman type quantity and end up proving characteristics of the regularizer corresponding to the dual view of the perturbations. While nonconvex losses, end up using some of the properties of the FTPL like monotonicity, the general structure for both cases remain same). Such a move would have enhanced the clarity of the paper. - In the proof, line 451, the quantity B looks awfully similar to Bregmann divergence except that the regularizer need not be strictly convex. Is there a reason why the regularizer cannot be a strictly convex function? - There are some crucial elements of the proof or common techniques which could be bought into the main text and some of the results on minimax games can be pushed to the supplementary.

Relation to Prior Work: - The authors cite previous related works on the worst case optimal regret for both convex and non-convex analysis. - They also cite relevant literature for convex-concave and nonconvex-nonconcave minimax problems. - Its interesting that there's no comparison/discussion on the nonconvex minimax games for the reference [18].

Reproducibility: Yes

Additional Feedback:


Review 2

Summary and Contributions: This paper gives an algorithm "Optimistic Follow the Perturbed Leader" (OFTPL) for when one has, at the start of every trial, an estimate of the loss function corresponding to that trial (which is revealed at the end of the trial). OFTPL is in fact two algorithms: one for when the loss functions are convex and one for when they are non-convex. These algorithms require access to a linear optimisation oracle (for the convex case) and a more general optimisation oracle (for the non-convex case). Bounds are given for both algorithms. Finally, the authors then show how OFTPL can be applied to the computation of Nash equilibria in Minimax games.

Strengths: The authors show how FTPL can be modified to take into account predictability of loss functions, which appears novel. Their solution of Minimax games, although having the same rate of convergence as FTPL when run on a single processor, is highly parallelisable (whereas FTPL isn't). The authors also give a proof of the duality of FTPL and "Follow the regularised leader" (FTRL) for very general perturbation distributions.

Weaknesses: The given bounds of OFTPL depend on x_t (or P_t in the non-convex case) which are quantities which have been computed by the algorithm - an intuitive/clear bound should not depend on quantities computed by the algorithm. The only application of OFTPL given is that of finding Nash equilibria of minmax games and their result only improves on FTPL when we have parallel processing (and only a quadratic improvement (in T) on the single processor case which isn't that much of an improvement for when parallelising an algorithm). I do not think that this result is good enough for a Neurips paper. The main weakness is that the general bounds depend on quantities computed in the algorithm (with no simple and natural interpretation in terms of the problem). In the minimax games example this isn't the case but I don't think this example is a strong enough result on its own.

Correctness: As far as I can see the claims and method are correct.

Clarity: The paper is well written.

Relation to Prior Work: The difference between the results OFTPL and OFTRL are not discussed.

Reproducibility: Yes

Additional Feedback: After Rebuttal: I have incorporated the author feedback into the review - my score remains the same.


Review 3

Summary and Contributions: The paper analyzes the regret minimization strategy known as optimistic follow the perturbed leader (OFTPL). The setting is that the loss functions are "benign and predictable" and hence the strategy has access to reasonable predictions of the gradients at each step (this is the meaning of "optimistic"). They consider both convex loss functions and non-convex loss functions, and in both cases they give a tighter analysis of the regret bounds. The implications of these new bounds is demonstrated on the application of solving minimax games, both in the convex-concave setting and the general smooth setting. The paper proves bounds on the ergodic convergence to equilibrium. They match the bounds that are known for FTPL, but the algorithm can be parallelized and therefore can use fewer iterations.

Strengths: Understanding the regret bounds of OFTPL is clearly an important question, given the practical appeal: FTPL and OFTPL are computationally superior to their follow the regularized leader counterparts, and the optimistic setting is often realizable in practice. In the convex case, the regret bounds proven here are superior to those of FPTL, and they match them of the gradient predictions are meaningless. Parallelization seems useful.

Weaknesses: The non-convex bounds are less impressive. In the application to non-convex minimax games they only recover known bounds (if we ignore parallelization). The method converges to equilibrium only ergodically, even in the case of convex minimax games, where one could hope for point-wise convergence.

Correctness: The claims appear to be correct.

Clarity: The paper is clear.

Relation to Prior Work: Previous work is well discussed.

Reproducibility: Yes

Additional Feedback:


Review 4

Summary and Contributions: This paper looks at the problem of FTPL in the context of predictable loss sequences. They extend FTPL to Optimistic FTPL (similar to what was done with FTRL in a different paper). They provide better regret bounds for the OFTPL case, both when the loss function is convex as well as when it is non-convex. They extend their formulation to smooth minimax games and provide an efficient algorithm to converge to Nash Equilibrium, where the algorithm is parallelizable with fewer iterations O(T) compared to FTPL which takes O(T^2) time.

Strengths: Main advantages: 1. New formulation for OFTPL. 2. Stability defined to bound regret. 3. Provides better bounds than previous known algorithms both for convex and non-convex cost function. 4. Parallelizable efficient application provided for minimax games.

Weaknesses: The paper builds on top of the previous work in the FTRL space. One part that was not completely explored was how this connects to other projection free methods like Frank-Wolfe and whether those methods can also be extended to give similar rates for FTPL. Further it would be good to see a perturbation-regularization analog for the optimistic versions of FTPL and FTRL.

Correctness: The claims largely look correct. I didn't go through the mathematical details very thoroughly.

Clarity: The paper is largely well written and for anyone familiar with the basic work in online regret minimization it is easy to follow the paper.

Relation to Prior Work: Yes. The relationships as well as novelty is clearly called out.

Reproducibility: Yes

Additional Feedback: In Corollary 4.2, the tightness on the bound is in term of the last term. It would be good to get an idea what is the average/max bound of these terms to get an idea how much better the bound provided in this case is. I have read the authors' feedback and stick by my rating.

[Author Response · NeurIPS 2020]

We thank the reviewers for their feedback. Below, we address the main questions/concerns raised.

**Reviewer 1.** *Confusion on parallelization:* Theorems $5.1, 5.2$, show that OFTPL converges at $O(T^{-1})$ rate after
running for $T$ iterations and making $2T$ parallel calls to the optimization oracle in each iteration. This is equivalent to
the claim made in lines 16-18, with $T$ replaced by $T^{1/2}$.

*Comparison with [18]:* we note that the result of [18] was improved by [11], which we compare against (see lines
316-317). [18] show that FTPL converges to a NE at $O(T^{-1/3})$ rate using $T$ calls to the optimization oracle. [11]
improve this result and show that FTPL converges at $O(T^{-1/2})$ rate. The OFTPL algorithm achieves these rates, but
unlike FTPL which runs for $T$ iterations, OFTPL only requires $T^{1/2}$ iterations and makes $O(T^{1/2})$ parallel calls to the
optimization oracle in each iteration.

*Line 451:* If the predictions of OFTPL are stable, we can actually show that regularizer $R$ is strongly convex (we rely on
this property of $R$ in line 454). However, it need not be differentiable. As a result, the traditional Bregmann divergence
is not well defined. So we need to do a more careful analysis. As rightly pointed out, our analysis relies on Bregmann
divergence like quantities defined in line 451.

**Reviewer 2.** *OFTPL vs OFTRL.* Although the idea of optimism has been applied to FTRL quite some time ago, its
extension to FTPL has not been done so far. The main challenge has been that the proof methodology of FTPL is quite
different from that of FTRL, and it was unknown how to incorporate optimism in this proof framework. Our work
addresses this challenge and incorporates optimism in the proof methodology of FTPL, by using the duality between
perturbation and regularization. Computationally, OFTPL has significant benefits over OFTRL. *In the convex case*,
unlike OFTRL, OFTPL only requires access to a linear optimization oracle, which is much easier to implement for a
number of problems of interest [8,9,10] (the counterpart of this in offline optimization is projected gradient descent vs.
Frank-Wolfe algorithms, which has been widely studied [Jaggi 2013]). Moreover, OFTPL achieves the same $O(T^{-1})$
rates as OFTRL by making $T$ parallel calls to the linear optimization oracle in each iteration. The computational
advantages of OFTPL over OFTRL become even more stark in the *nonconvex case*. The well known OFTRL algorithm
for nonconvex case is entropic mirror descent, which works in the space of probability distributions over $\mathcal{X}$ (see line
102). In this case, OFTRL recommends playing an entire distribution over $\mathcal{X}$ in each iteration, which is not feasible in
practice.

*Bounds.* By letting the bounds depend on $\mathbf{x}_t, P_t$, we get very tight regret bounds. When instantiated for specific
problems such as minimax games, such bounds help us derive fast convergence rates. To be precise, the terms depending
on $\mathbf{x}_t, P_t$ are very crucial for deriving the results in Theorems 5.1, 5.2. Without these exact terms, we believe one can't
show fast convergence rates. Besides, such terms also appear in the regret bounds of OFTRL [14].

*Significance of results.* We believe the quadratic improvement is very significant, especially for large scale problems
such as training of GANs, adversarial training on ImageNet. Our results show that the performance of FTPL run with
$T = 1000$ can be matched by OFTPL run with $T = 50$ iterations with multiple parallel calls to the optimization oracle
in each iteration. Given that each iteration of OFTPL/OFTRL can take hours, this is a significant improvement.

**Reviewer 3.** *Nonconvex Bounds.* We believe the nonconvex bounds are very useful, especially given the prevalence
of parallel compute resources. For example, many popular nonconvex games such as adversarial training and GAN
training arise in the context of deep learning, where one has access to GPUs which support parallel computation.

*Last iterate convergence.* We agree that having last-iterate convergence is useful in the convex case. We will pursue this
direction in the future.

**Reviewer 4.** *FTPL vs. Frank-Wolfe.* The updates of FTPL and vanilla online Frank-Wolfe (FW) algorithms look very
similar to each other, except for the perturbation term in the former. This suggests there might be a deeper connection
between the two. However, it is not immediately clear if FW is as parallelizable as OFTPL. We believe this is an
interesting direction to pursue in the future.

*Perturbation-regularization analog.* The duality between perturbation and regularization also holds for optimistic
versions of FTPL and FTRL. This follows from Proposition 3.1 by replacing $\nabla_{1:t-1}$ with $\nabla_{1:t-1} + g_t$. We infact rely
on this duality between OFTPL and OFTRL in our proof of Theorem 4.1 (see line 449).

*Corollary 4.2.* One way to understand the usefulness of the last term in the bound is to study the rates of convergence
one would be able to derive for smooth minimax games without this term. For the setting considered in Theorem 5.1,
one can only show $O(T^{-3/4})$ rate of convergence without the last term in the bound of Corollary 4.2.

[1] Jaggi, Martin. "Revisiting Frank-Wolfe: Projection-free sparse convex optimization." Proceedings of the 30th international
conference on machine learning. No. CONF. 2013.


[Meta-Review · NeurIPS 2020]

The paper provides a follow the perturbed leader algorithm and analysis that can obtain better regret bounds when loss/gradient sequence is predictable. The proofs relies on using the equivalent regularization view of FTPL. The authors also provide an application of this result to providing a parallelizable algorithms for solving smooth convex concave saddlepoint games Most of the reviewers found the result interesting. Please address the concerns of the reviewer. Personally, I find the predictable sequences result interesting. However the application to minimax saddle point solving I find less interesting given the multiple calls to the linear optimization oracle. Its not clear how this competes with other FTRL methods that also obtain 1/T rates but without multiple steps in one iteration.